# Sublinear Algorithms for Hierarchical Clustering

**Arpit Agarwal**[1]   **Sanjeev Khanna**[2]   **Huan Li**[2]   **Prathamesh Patil**[2]

[1]Data Science Institute, Columbia University
[2]Department of Computer and Information Science, University of Pennsylvania
`{arpit.agarwal}@columbia.edu`
`{sanjeev,huanli,pprath}@cis.upenn.edu`

## Abstract

Hierarchical clustering over graphs is a fundamental task in data mining and machine learning with applications in many domains including phylogenetics, social network analysis, and information retrieval. Specifically, we consider the recently popularized objective function for hierarchical clustering due to Dasgupta [20], namely, minimum cost hierarchical partitioning. Previous algorithms for (approximately) minimizing this objective function require linear time/space complexity. In many applications the underlying graph can be massive in size making it computationally challenging to process the graph even using a linear time/space algorithm. As a result, there is a strong interest in designing algorithms that can perform global computation using only sublinear resources (space, time, and communication). The focus of this work is to study hierarchical clustering for massive graphs under three well-studied models of sublinear computation which focus on space, time, and communication, respectively, as the primary resources to optimize: (1) (dynamic) streaming model where edges are presented as a stream, (2) query model where the graph is queried using neighbor and degree queries, (3) massively parallel computation (MPC) model where the edges of the graph are partitioned over several machines connected via a communication channel.

We design sublinear algorithms for hierarchical clustering in all three models above. At the heart of our algorithmic results is a view of the objective in terms of cuts in the graph, which allows us to use a relaxed notion of cut sparsifiers to do hierarchical clustering while introducing only a small distortion in the objective function. Our main algorithmic contributions are then to show how cut sparsifiers of the desired form can be efficiently constructed in the query model and the MPC model. We believe this relaxed notion of cut sparsifiers may be of broader interest. We complement our algorithmic results by establishing nearly matching lower bounds that rule out the possibility of designing algorithms with better performance guarantees in each of these models.

## 1   Introduction

Hierarchical clustering (HC) is a popular unsupervised learning method for organizing data into a dendrogram (rooted tree). It can be viewed as clustering datapoints at multiple levels of granularity simultaneously, with each leaf of the tree corresponding to a datapoint and each internal node of the tree corresponding to a cluster consisting of its descendent leaves. Much of the technical development of HC originated in the field of *phylogenetics*, where the motivation was to organize the different species into an evolutionary tree based on genomic similarities [22]. Since then, this tool has seen widespread use in data analysis for a variety of domains ranging from *social networks*, *information retrieval*, *financial markets* [28, 8, 43] amongst many others.

36th Conference on Neural Information Processing Systems (NeurIPS 2022).

Due to its popularity, HC has been extensively studied and several algorithms have been proposed. The most prominent amongst these are *bottom-up agglomerative* algorithms such as average linkage, single linkage, complete linkage etc. (see Chapter 14 in [33]). However, despite these advances on the algorithmic front, very few formal guarantees were known for their performance, primarily owing to a historic lack of a well defined objective function. Therefore, the study of HC was largely empirical in nature for a long time.

A part of this issue was recently resolved, when [20] proposed an objective function for similarity-based HC. This has since sparked interest in both the theoretical computer science as well as machine learning communities, for designing algorithms with provable guarantees for this objective [9, 12, 10, 19, 2]. The formal description is as follows: given as input a weighted undirected graph $G = (V, E, w)$ with $n$ vertices (datapoints) and $m$ edges with positive edge weights corresponding to pairwise similarities between its endpoints, the objective is to find a hierarchy $\mathcal{T}$ over leaf nodes corresponding to the vertices $V$ that minimizes the cost function

$$\mathsf{cost}_G(\mathcal{T}) := \sum_{\{i,j\} \in E} w_{ij} \cdot |\mathcal{T}_{ij}|, \tag{1}$$

where $\mathcal{T}_{ij}$ is the subtree rooted at the least-common ancestor of $i, j$ in $\mathcal{T}$ and $|\mathcal{T}_{ij}|$ is the number of descendent leaves in $\mathcal{T}_{ij}$. Intuitively, $\mathsf{cost}_G(\mathcal{T})$ incentivizes cutting *heavy* edges at lower levels in $\mathcal{T}$, thereby placing more similar datapoints closer together. This objective has been shown to have several desirable properties, including one that guarantees an optimal tree which is binary [20].

This minimization objective however, turns out to be NP-hard. Consequently, [20] and other subsequent work explored this objective from an approximation algorithms perspective [42, 9, 12, 10, 19, 2]. The best known polynomial time approximation is $O(\sqrt{\log n})$ which is achieved by the recursive sparsest cut (RSC) algorithm [20, 9, 19]. It is also known that no constant factor polynomial time approximation is possible for this objective under the *small-set expansion* (SSE) hypothesis [9].

In this paper, we study the above minimization objective for HC in the context of *massive* graphs. While the currently known best algorithm can be considered "efficient" in the classical sense, i.e. requires polynomial space and time[1], this complexity can be prohibitive in many modern applications of HC that deal with staggering volumes of data. For example, current social networks contain billions of edges which imposes serious limits on their storage and processing. Therefore, alternative models of computation need to be considered in the context of such massive graphs. In this work, we consider three widely-studied models, each aimed at optimizing a different fundamental resource: $(i)$ the (dynamic) *streaming model* [24] for space efficiency, where the edges are presented in a stream, $(ii)$ the *general graph (query) model* [31] for time efficiency, where the edges can be accessed via degree and neighbour queries, and $(iii)$ the *massively parallel computation model* (MPC) for communication efficiency, where the edges are partitioned across multiple machines connected together through a communication channel. The focus of our work is the following fundamental question:

*Can we design sublinear (in the number of edges) algorithms for hierarchical clustering in each of these massive-graph computation models?*

We provide an *almost complete resolution* to this question by providing *matching upper and lower bounds* for sublinear algorithms in all three canonical models of computation discussed above.

**Remark 1.** *When studying graph problems in the sublinear setting, one can consider an even more constrained setting where the available resource is $o(n)$, i.e. sublinear in the number of vertices. However, we are interested in actually finding a hierarchical clustering of the data, the writing of which takes $\Omega(n)$ time and space (and in MPC, $\Omega(n)$ machine memory). Since in most practical settings, the bottleneck is often the edges in the graph rather than the vertices, we believe it makes more sense for us to consider sublinearity only in the edge parameter, i.e. $m$, in all three models.*

Due to the space needed to present full technical details of our main results, we focus here on giving a overview; the complete paper is deferred to the Appendix. Our results are summarized in Table 1.

**Notation.** In the following section, we use $n$ and $m$ to refer to the number of vertices and edges in the input graph, respectively. We use $\widetilde{O}(\cdot)$ to suppress multiplicative $O(\log^c n)$ factors for constant $c$,

---

[1]We also note that a near-linear (in the number of edges) time implementation of RSC can be obtained by plugging a recent breakthrough result for fast max-flow computation [14] into the balanced separator approximation algorithm in [44].

Table 1: **Summary of Results.** Each row gives an upper and lower bound on the resource (space/time/communication) required for $\widetilde{O}(1)$ approximation in the corresponding model.

| | Setting/Parameters | Upper Bound | Lower Bound |
|---|---|---|---|
| **Streaming Model** (Sublinear Space) | 1-pass | $\widetilde{O}(n)$, Thm 2 | $\Omega(n)$, trivial |
| **Query Model** (Sublinear Time) Edges $m = \Theta(n^\zeta)$ in $G$ | $1 < \zeta \leq 4/3$ | $\widetilde{O}(n^\zeta)$, Thm 4 | $\Omega(n^{\zeta - o(1)})$, Thm 8 |
| | $4/3 < \zeta \leq 3/2$ | $\widetilde{O}(n^{4-2\zeta})$, Thm 4 | $\Omega(n^{4-2\zeta - o(1)})$, Thm 8 |
| | $3/2 \leq \zeta < 2$ | $\widetilde{O}(n)$, Thm 4 | $\Omega(n)$, Thm 8 |
| **MPC Model** (Sublinear Communication) | 1-round | $\widetilde{O}(n^{4/3})$, Thm 6 | $\Omega(n^{4/3 - o(1)})$, Thm 9 |
| | 2-round | $\widetilde{O}(n)$, Thm 5 | $\Omega(n)$, trivial |

and the term "w.h.p." implies with probability $1 - 1/\text{poly}(n)$. Lastly, $\phi$ will denote the approximation ratio of any desired offline algorithm for hierarchical clustering. For example, if allowed unbounded computation time, we have $\phi = 1$; given polynomial time, the current best algorithm [9] gives $\phi = O(\sqrt{\log n})$. We assume this abstraction as any improvement in the approximation ratio here automatically implies an identical improvement in our upper bounds.

## 2 Overview of Algorithmic Results

We begin by presenting our algorithmic results for the three models of computation, which at their core, are all based on the same meta-algorithm which follows from a *new structural view* of the minimization objective defined in Eq. (1) in terms of global cuts in the input graph.

### 2.1 A Meta-Algorithm for Sublinear-Resource Hierarchical Clustering

In their paper, [20] showed that $\mathsf{cost}_G(\mathcal{T})$ can be viewed in two equivalent ways, the first being the one defined earlier in Eq. (1), and the other in terms of the *splits* induced by the internal nodes in the hierarchy: given a hierarchy $\mathcal{T}$ with each internal node corresponding to a binary split[2] where some subset of vertices $S \subseteq V$ of the input graph is partitioned into two pieces $(S_\ell, S_r)$, then

$$\mathsf{cost}_G(\mathcal{T}) := \sum_{\text{splits } S \to (S_\ell, S_r) \text{ in } \mathcal{T}} |S| \cdot w_G(S_\ell, S_r),$$

where for any disjoint subsets $S, T \subset V$, $w_G(S, T)$ is the total weight of the edges in $G$ going between $S$ and $T$. At this point, one might be tempted to think that if we could somehow construct a sparse representation of $G$ such that the weights $w_G(S, T)$ are approximately preserved for any disjoint $S, T \subset V$, then the cost of every hierarchy would also be approximately preserved. Following this, we could run any desired offline algorithm on this representation with improved efficiency due to its sparsity without much loss in the quality of its solution. Unfortunately, this is just wishful thinking as such a representation can easily be shown to require $\Omega(m)$ time and space[3]. Our first contribution is to show there is in fact a *third equivalent view* of this same objective function in terms of global cuts in $G$, and the above alternate formulation serves as our starting point.

This result follows from two critical observations, the first of which is given any two disjoint $S, T \subset V$, we can compute $w_G(S, T)$ exactly as $w_G(S, T) = (1/2) \cdot (w_G(S, \overline{S}) + w_G(T, \overline{T}) - w_G(S \cup T, \overline{S \cup T}))$. We could stop here as the quantities on the right are all graph cuts, and it is well known [7] that one can construct a $\widetilde{O}(n)$ sized sparsifier that approximately preserves all graph cuts. Unfortunately, the distortion in $w_G(S, T)$ can be very large depending on the quantities on the right, and the cumulative error in $\mathsf{cost}_G(T)$ blows up with the depth of the tree which is even worse. Here

---

[2]This is without loss of generality since there always exists an optimal hierarchy that is binary.

[3]Given such a sparsifier, by setting $S = \{u\}$ and $T = \{v\}$, one can recover whether or not edge $(u, v)$ is present in $G$ for any $u, v \in V$.

is the second observation: the negative term $w_G(S \cup T, \overline{S \cup T})$ that internal node $S$ contributes to the cost also appears as a positive term in its parent's contribution to the cost. We can pass this term as a *discount* in its parent's contribution to the cost, which after cascading gives a third view of Eq. (1).

$$\mathsf{cost}_G(\mathcal{T}) := \frac{1}{2} \cdot \left( \sum_{\text{splits } S \to (S_\ell, S_r) \text{ in } \mathcal{T}} \left( |S_r| \cdot w_G(S_\ell, \overline{S_\ell}) + |S_\ell| \cdot w_G(S_r, \overline{S_r}) \right) + \sum_{v \in V} w_G(\{v\}, \overline{\{v\}}) \right),$$

a linear combination of graph cuts. This gives a *strong blackbox reduction* to cut-sparsifiers; preserving graph cuts to a $(1 \pm \epsilon)$ factor also preserves the cost of all hierarchies to a $(1 \pm \epsilon)$ factor.

However, we are not done yet, as cut-sparsifiers cannot be computed efficiently in certain models of computation; for instance, they necessarily require $\Omega(m)$ queries to the underlying graph. We therefore introduce a weaker notion of sparsification that, for any cut $(S, \overline{S})$, allows for an additive error of $\delta \min\{|S|, |\overline{S}|\}$ in addition to the usual multiplicative error of $(1 \pm \epsilon)$ (Appendix B Defn. 1). We term this generalization an $(\epsilon, \delta)$-cut sparsifier. A similar notion was also proposed in an earlier work by [36], which unfortunately does not work here (see Appendix D.1 for details). Our next result then shows that the distortion in the cost of any hierarchy under this weaker sparsifier is also bounded.

**Theorem 1** (informal). *Given any weighted graph $G$, an $(\epsilon, \delta)$-cut sparsifier of $G$ preserves the cost of any tree $\mathcal{T}$ up to a multiplicative $(1 \pm \epsilon)$ factor, and an additive $O(\delta n^2)$ factor.*

Therefore, if we could lower bound the cost of the optimal hierarchical clustering by some quantity $C$, we could set $\delta = \epsilon C / n^2$. The above result would then imply morally the same result as that achieved by traditional cut-sparsifiers: preserving graph cuts in this $\epsilon, \delta$ sense for a sufficiently small $\delta$ also preserves the cost of all hierarchies upto a $(1 \pm \epsilon)$ factor. The last key result exactly establishes such a general purpose lower bound on the cost of any hierarchical clustering in a graph, which can be efficiently estimated in all models of computation we consider.

This chain of ideas results in the following meta-algorithm for sublinear HC given any parameter $\epsilon > 0$ and model of computation: Compute the lower bound on the cost of an optimal HC which establishes the tolerable additive error in our $(\epsilon, \delta)$ sparsifier, following which we efficiently (in the resources to be optimized) compute the said sparsifier. We finally run any $\phi$-approximate HC algorithm, which is guaranteed to find a $(1 + \epsilon)\phi$-approximate HC tree. Our subsequent results give sublinear constructions of these $(\epsilon, \delta)$-cut sparsifiers in each of the three models of computation.

## 2.2 Sublinear Space Algorithms in the (Dynamic) Streaming Model

We first consider the dynamic streaming model for sublinear space algorithms, where the edges in the input graph are presented in an arbitrarily ordered stream of edge insertions and deletions. Our upper bound here is a direct consequence of Theorem 1 used in conjunction with [1], a seminal result that showed an $(\epsilon, 0)$-cut sparsifier can be constructed in $\widetilde{O}(\epsilon^{-2} n)$ space and a single pass in this setting.

**Theorem 2** (informal). *There exists a single-pass, $\widetilde{O}(n)$ space, streaming algorithm that given any weighted graph $G$ presented in a dynamic stream, w.h.p. finds a $(1 + o(1)) \cdot \phi$-approximate HC of $G$.*

As outlined in our meta-algorithm, instantiating the offline algorithm with RSC with the input graph being the sparsifier gives us a polynomial time, $\widetilde{O}(n)$ space, single-pass dynamic streaming algorithm with approximation ratio $O(\sqrt{\log n})$ as a corollary. See Appendix C for more details.

We note that coupled with a recent result [15], our Theorem 1 also implies an $\widetilde{O}(n)$-space algorithm for finding a $(1 + o(1)) \cdot \phi$-approximate HC in the more general *turnstile streams*, where arbitrary edge weight updates can appear.

## 2.3 Sublinear Time Algorithms in the Query Model

We next consider the general graph model [31] for sublinear time algorithms, where the input graph can be accessed via two[4] types of queries: $(i)$ degree queries: given $u \in V$, returns degree $d_u$, and $(ii)$ neighbour queries: given $u \in V$, $i \leq d_u$, returns the $i$-th neighbour of $u$. Note that this model

---

[4]This query model also allows a third type of queries: pair queries which answer whether an edge $(u, v)$ exists or not. However, we do not need these queries in our algorithm.

can be easily implemented using an adjacency array representation of the graph (See Appendix D for a more detailed discussion). We first present the result for unweighted graphs, where it is easier to see the key intuition. Our main result in this model is a sublinear time construction of an $(\epsilon, \delta)$-sparsifier.

**Theorem 3** (informal). *There exists an algorithm that given query access to any unweighted graph $G$, and any parameters $\epsilon, \delta \in (0, 1]$, can find an $(\epsilon, \delta)$-cut sparsifier of $G$ w.h.p. in $\widetilde{O}(n/(\epsilon^2 \delta))$ time.*

Our algorithm is based on a simple yet elegant idea (which builds upon a slightly different idea proposed in [36]; see Appendix D.1 for a detailed discussion): if we embed a constant-degree expander with edge weights $\delta$ in an unweighted graph (with unit edge weights), then the effective resistance of every edge in the resulting composite graph is tightly bound in terms of the *effective degrees* of its incident vertices; the effective degree of a vertex is a weighted sum of its degree in the input graph and its degree in the expander. We can then leverage the effective resistance sampling scheme of [45] to construct an $(\epsilon, 0)$-cut sparsifier of this composite graph, which then is *deterministically* an $(\epsilon, \delta)$-cut sparsifier of the input graph with the sources of error being the usual multiplicative $\epsilon$ term due to sparsification itself, and the (small) additive $\delta$ term due to the (few) extra edges introduced by the expander. We can construct constant degree graphs that are expanders with high probability in sublinear time, and we show that there is an efficient rejection sampling scheme for sampling edges according to their effective resistances, giving the above result: an $(\epsilon, \delta)$-cut sparsifier with $\widetilde{O}(n/(\epsilon^2 \delta))$ edges in the same amount of time and queries. Moreover, the queries are completely *non-adaptive* assuming prior knowledge of vertex degrees. This construction of $(\epsilon, \delta)$-cut sparsifiers in conjunction with Theorem 1 then gives our sublinear time upper bound in this model.

**Theorem 4** (informal). *There exists an algorithm that given query access to any unweighted graph $G$ with $m = \Theta(n^\zeta)$ for $\zeta \in [0, 2]$, can find a $(1 + o(1)) \cdot \phi$-approximate HC of $G$ w.h.p. using $\widetilde{O}(g(n, \zeta))$ queries, where $g(n, \zeta) \leq n^{4/3}$ is given by $g(n, \zeta) = \max\{n, n^\zeta\}$ when $\zeta \in [0, 4/3]$, $g(n, \zeta) = \max\{n, n^{4-2\zeta}\}$ when $\zeta \in (4/3, 2]$. Moreover, given any (arbitrarily small) constant $\tau > 0$, the algorithm can find an $O(\sqrt{\log n})$-approximate HC of $G$ w.h.p. in $\widetilde{O}(g(n, \zeta) + n^{1+\tau})$ time.*

It is interesting to observe that the query complexity $g(n, \zeta)$ reduces as the graph becomes denser. This is because the cost of the optimal HC increases with the density of the input graph, which allows us to tolerate a larger additive error in our cut sparsifier, thereby making it sparser. In Section 3 we also discuss lower bounds showing that this query complexity is also the best possible for any $\widetilde{O}(1)$-approximate algorithm. The sublinear time claim in Theorem 4 is implied by results from [44] and [14] . Specifically, [44] showed that, for any constant $\tau \in (0, 1/2)$, sparsest cuts and balanced separators can be approximated to within $O(\sqrt{\log n})$ in $\widetilde{O}(m)$ time plus $\widetilde{O}(n^\tau)$ maximum flow computations on graphs with $\widetilde{O}(n)$ edges, and [14] showed that a maximum flow on a graph of $m$ edges can be computed in $O(m^{1+o(1)})$ time. These two combined with the fact that our sparsifier contains just $\widetilde{O}(g(n, \zeta))$ edges give us the desired running time bound.

We generalize the above result in Appendix D.2 to weighted graphs by grouping edges according to geometrically increasing weights and constructing $(\epsilon, \delta)$-cut sparsifiers for each weight class, and get an algorithm with essentially the same worst case performance and $O(\log n)$ rounds of adaptivity.

## 2.4  Sublinear Communication Algorithms in the MPC model

Lastly, we consider the MPC model [6, 3] for sublinear communication, which is a common abstraction of many MapReduce-style computational frameworks. Here, the edges in the input graph are partitioned across several machines that communicate with each other in synchronous rounds. Each machine has memory sublinear in $m$, with its total communication bounded by its memory. A more detailed description of this model is given in Appendix E.

Our first result is a 2-round MPC algorithm that uses $\widetilde{O}(n)$ memory per machine. In our algorithm, we leverage a construction of $(\epsilon, 0)$-cut sparsifiers due to [1] using $\widetilde{O}(1)$ random *linear sketches* per vertex in the graph. We show that 2 rounds are sufficient to construct these linear sketches for each vertex– the first round is used to construct *partial* sketches using local edges and the second round is used to aggregate these partial sketches into complete sketches for each vertex. This construction of $(\epsilon, 0)$-cut sparsifiers in conjunction with Theorem 1 gives us the following result.

**Theorem 5** (informal)**.** *There exists an MPC algorithm that given any weighted graph $G$ with edges partitioned across machines with $\widetilde{O}(n)$ memory and access to public randomness, can find a $(1 + o(1)) \cdot \phi$-approximate HC of $G$ w.h.p. on a designated machine in 2 rounds of MPC.*

Our second result is a 1-round MPC algorithm that solves this problem for unweighted graphs using machines with $\widetilde{\Theta}(n^{4/3})$ memory. The execution of our algorithm depends on the density of the underlying graph. If $m \leq n^{5/3}$, then we can again use the result from [1] by constructing local linear sketches on each machine and sending them to a coordinator who can aggregate them. Note that we only require 1 round in this case as the number of machines is $\leq n^{1/3}$, and hence, we only need to communicate $\widetilde{O}(n^{1/3})$ sketches per vertex which is within our memory budget. If $m > n^{5/3}$, we show that the cost of the optimal hierarchy is sufficiently large such that a *coarse* $(\epsilon, \delta)$-cut sparsifier of size $\widetilde{O}(n^{4/3})$ obtained by randomly subsampling edges will suffice. As a consequence, each machine can subsample its edges independently and send these $\widetilde{O}(n^{4/3})$ edges to the coordinator. We now summarize this 1-round result below.

**Theorem 6.** *(informal) There exists an MPC algorithm that given any unweighted graph $G$ with edges partitioned across machines with $\widetilde{O}(n^{4/3})$ memory and access to public randomness, can find a $(1 + o(1)) \cdot \phi$-approximate HC of $G$ w.h.p. on a designated machine in 1 round of MPC.*

In the next section, we discuss a 1-round lower bound for MPC which shows that $n^{4/3 - o(1)}$ memory per machine is needed by any $\widetilde{O}(1)$-approximate algorithm on unweighted graphs.

## 3 Overview of Lower Bounds

First note that in the (dynamic) streaming model, since our goal is for the algorithm to output a hierarchical clustering tree, we necessarily need $\Omega(n)$ space. Thus our $\widetilde{O}(n)$ space dynamic streaming algorithm that obtains a $(1 + o(1))$-approximation is nearly optimal. We then show that in the other two models of computation, our algorithms are also essentially the best possible.

### 3.1 Lower bounds in the Query Model

Note that the query complexity bound of our sublinear time algorithm is always at most $\widetilde{O}(n^{4/3})$, where the worst-case input is an unweighted graph with about $m \approx n^{4/3}$ edges. We note that our algorithm obtains an $O(\sqrt{\log n})$-approximation and (i) is completely *non-adaptive* on unweighted graphs assuming prior knowledge of vertex degrees, and has $O(\log n)$ rounds of adaptivity on weighted graphs; (ii) only uses degree queries and neighbor queries (no pair queries needed, see Footnote 4). We then show that $n^{4/3 - o(1)}$ queries are indeed necessary for obtaining *any* $\widetilde{O}(1)$-*approximation* even in *unweighted* graphs and given *unlimited* adaptivity and access to *pair queries*.

**Theorem 7** (informal)**.** *Let $\mathcal{A}$ be a randomized algorithm that, on any input unweighted graph with $\Theta(n^{4/3})$ edges, outputs with high probability a $\mathrm{polylog}(n)$-approximate hierarchical clustering tree. Then $\mathcal{A}$ necessarily uses at least $n^{4/3 - o(1)}$ queries.*

We briefly describe the family of hard graph instances that we use to prove this result. Roughly, a graph from such a family is generated by first taking a union of $n^{2/3}$ vertex-disjoint cliques of size $n^{1/3}$ each, and then connecting them by a random "perfect matching". More specifically, we treat each clique as a *supernode*, and generate a perfect matching between these $n^{2/3}$ supernodes uniformly at random. Then if the $i^{\text{th}}$ clique is matched to the $j^{\text{th}}$ clique in the perfect matching, we will add about $n^{o(1)}$ edges between these two cliques, which are also chosen in a random manner. We show that, in order to output a good hierarchical clustering solution, it is necessary to discover a non-trivial portion of the edges that we add between the cliques, even though their number is relatively tiny compared to those within, and the latter task provably requires $n^{4/3 - o(1)}$ queries.

While this plan looks intuitive, one has to be careful about not leaking information about the "perfect matching" between the cliques from the vertex degrees, which an algorithm knows a priori (or can otherwise acquire using $O(n)$ non-adaptive degree queries). In particular, once the inter-clique edges are added, one could tell that the vertices with degree higher than $n^{1/3} - 1$ are those participating in the perfect matching. Note that there are only $n^{2/3 + o(1)}$ such vertices in total, and each of them has

degree at most $O(n^{1/3})$. As a result, by probing all neighbors of these vertices, one can easily find all the inter-clique edges using $n^{1+o(1)}$ neighbor queries.

Our way around this issue is to also *delete* certain edges within the cliques based on what inter-clique edges we have added, so as to ensure that each vertex has the exact same degree of $n^{1/3} - 1$. This of course further complicates things as it increases the correlation between the edge slots — for instance, whenever an edge between a matched pair of cliques is revealed to the algorithm, missing edges within each clique are no longer independent. Consequently, our proof for this lower bound turns out to be considerably involved; we refer the reader to Appendix F.2 for more details.

Theorem 7 shows that the worst-case query complexity of our algorithm is nearly optimal. Note that, however, for unweighted graphs, our algorithm also obtains improved query/time complexity when $m$ is far from $n^{4/3}$. It is then natural to ask — are these improvements also the best possible? We answer this question in the affirmative. In particular, we show that one can push further the ideas we discussed above to get *a tight query lower bound for every graph density*. We summarize these lower bounds below. Note in particular that, for $m = \Theta(n^2)$, as we will show later, *any* hierarchical clustering achieves an $O(1)$-approximation, thus trivially 0 queries are sufficient.

**Theorem 8** (informal). *Let $\zeta \in [0, 2]$ be any constant. Let $\mathcal{A}$ be a randomized algorithm that, on any input unweighted graph with $\Theta(n^\zeta)$ edges, outputs with high probability a $\mathrm{polylog}(n)$-approximate HC. Then $\mathcal{A}$ necessarily uses at least $\Omega(g(n, \zeta))$ queries, where $g(n, \zeta) = \max\{n, n^{\zeta - o(1)}\}$ when $\zeta \in [0, 4/3]$, $g(n, \zeta) = \max\{n, n^{4 - 2\zeta - o(1)}\}$ when $\zeta \in (4/3, 2)$, and $g(n, \zeta) = 0$ when $\zeta = 2$.*

### 3.2 Lower bounds in the MPC Model

As our goal is for some machine to output a good hierarchical clustering tree, $\Omega(n)$ memory per machine is necessary. Indeed, our 2-round MPC algorithm obtains a $(1 + o(1))$-approximation for *weighted* graphs using a nearly optimal memory of $\widetilde{O}(n)$ per machine (Theorem 5).

To show that the number of rounds of our algorithm is also optimal, we prove that a superlinear (in particular, $n^{4/3 - o(1)}$) memory per machine is necessary for any 1-round MPC algorithm in which some machine outputs with high probability a $\mathrm{polylog}(n)$-approximate hierarchical clustering tree. Moreover, in our lower bound instance, the total memory of all machines is $\approx m$, which means that the input is split across fewest possible machines. We specifically prove the following result:

**Theorem 9** (informal). *Let $P$ be any 1-round protocol in the MPC model where each machine has memory $O(n^{4/3 - \varepsilon})$ for any constant $\varepsilon > 0$. Then at the end of the protocol $P$, no machine can output a $\mathrm{polylog}(n)$-hierarchical clustering tree with probability better than $o(1)$.*

Note that this lower bound matches our upper bound result in Theorem 6. Our family of hard instances is roughly defined as follows. Let $\alpha \approx 2/3, \beta \approx 1/3$ be certain constants. A graph $G$ of $2n$ vertices from such a family consists of two vertex-disjoint parts, each supported on $n$ vertices. The first part is supported on vertices $V_1$ and is a union of vertex-disjoint bi-cliques of size $n^\alpha$; the second part is supported on vertices $V_2$ and is in turn a union of vertex-disjoint bi-cliques of size $n^\beta$, where we have $|V_1| = |V_2| = n$. We will also permute the vertex labels of $G$ uniformly at random. See Figure 1 for an illustrative example of such a graph $G$.

We first show that in order to output a good hierarchical clustering solution, it is necessary to discover (almost) the exact clique structures of the vertex-induced subgraphs $G[V_1], G[V_2]$, for otherwise a balanced cut has non-trivial probability of cutting too many edges within the cliques. Then as an adversary, our strategy is to hide $G[V_2]$, which has significantly fewer edges than $G[V_1]$, by splitting $G$ across multiple machines. To this end, we observe that $G[V_1]$ can be tiled using edge-disjoint subgraphs $G_1, \ldots, G_t$ that are each *isomorphic* to $G[V_2]$ (see Figure 1). This suggests that we could give $G[V_2]$ to a uniformly random machine, and then give each $G_i$ to one of the other machines.

Note that, crucially, each machine's input follows the *exact same* distribution, namely a union of bi-cliques of size $n^\beta$ with vertex labels permuted uniformly at random, although the input distributions of different machines are correlated. As a result, each machine individually has no information whether its input graph is $G[V_2]$ or just a subgraph $G_i$ of $G[V_1]$. Therefore, each machine has to send a message to the coordinator such that, if its input graph happens to be $G[V_2]$, the coordinator will be able to recover the clique structures with high probability.

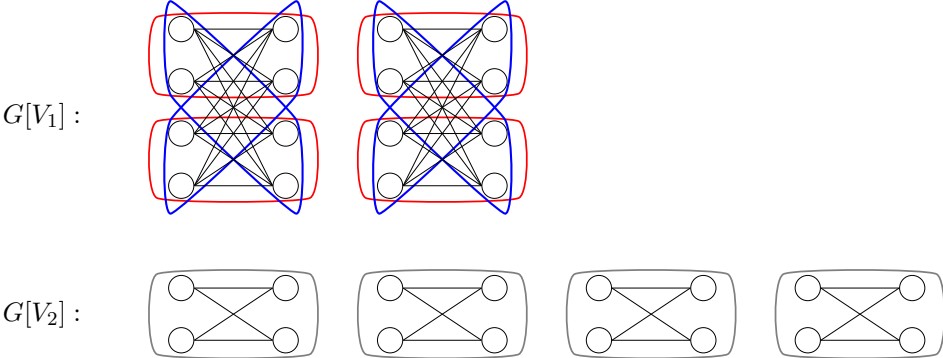

Figure 1: An illustrative example of an input graph $G$, where $G[V_1]$ is a union of two bi-cliques of size 8 each, and $G[V_2]$ is a union of four bi-cliques of size 4 each. Here $G[V_1]$ can be tiled using two edge-disjoint subgraphs each isomorphic to $G[V_2]$, which are the subgraph induced by edges within the four red frames, and the subgraph induced by edges within the four blue frames. So no machine can tell locally whether it was given the red subgraph, the blue subgraph, or $G[V_2]$.

Since the coordinator can only receive a total message size bounded by its machine memory, by choosing suitable values of $\alpha, \beta$, each machine on average can only send a message of size $o(n)$. Now the problem effectively becomes a one-way, two party communication problem, where Alice is given $G[V_2]$ and needs to send Bob a single message so that Bob can recover the clique structures with high probability. We then conclude the proof by showing that this two-party problem requires $\Omega(n)$ communication. We refer the reader to Appendix G for more details.

## 4   Related Work

The work of Dasgupta [20] is the starting point of our work. [20] defined the objective function for hierarchical clustering, namely minimum cost hierarchical partitioning, that we study in this paper. They showed that the resulting problem is NP-hard and the recursive sparsest-cut algorithm achieves an $O(\phi \log n)$ approximation, where $\phi = O(\sqrt{\log n})$ is the current best poly-time approximation for sparsest-cut. [42] improved this approximation factor to $O(\log n)$ using an LP-based algorithm. [19, 9] showed that the recursive sparsest cut algorithm of [20] in-fact achieves an $O(\phi)$ approximation. [42] and [9] also showed that no polynomial time algorithm can achieve constant factor approximation under the small set expansion (SSE) hypothesis. [19, 37] showed that by imposing certain probabilistic or structural assumptions on the graph, one can circumvent this constant factor inapproximability.

There has also been work on maximization objectives for hierarchical clustering. [39] considered a "dual" version of the Dasgupta objective: where the goal is to maximize the revenue $n \sum_{e \in E} w_e - \text{cost}_G(\mathcal{T})$. While the optimal values for both objectives are achieved by the same solution, this objective behaves very differently from an approximation perspective. [19] considered a setting where the edge weights correspond to dissimilarities rather than similarities and the goal is to maximize the dissimilarity-based objective $\text{cost}_G(T)$. [39] and [19] both study the average-linkage algorithm and show that it achieves approximation factors of $1/3$ and $2/3$, respectively. [10] provided algorithms with slightly better approximation factors of $1/3 + \delta$ and $2/3 + \delta$, respectively. [13] improved the approximation factor to $0.4246$ for the dual objective in [39], which was later improved to $0.585$ by [2]. Very recently, [40] improved the approximation to $0.71604$ for the dissimilarity objective of [19].

Several other variations of this basic setup have been considered. For example, [12] have considered this problem in the presence of structural constraints. [11, 38, 41] considered a setting where vertices are embedded in a metric space and the similarity/dissimilarity between two vertices is given by their distances. The most relevant to our work amongst these is [41] which considered this metric embedded hierarchical clustering problem in a streaming setting. However, the stream in their setting is composed of vertices while edge weights can be directly inferred using distances between vertices; whereas the stream in our streaming setting is composed of edges while vertices are already known. Moreover, their study is only limited to the streaming setting. There has also been work on designing faster/parallel agglomerative algorithms such as single-linkage, average-linkage etc. [48, 21, 46]. While these works share the same motivation as ours, namely, scaling HC algorithms to massive

datasets, these results are largely orthogonal to ours. The primary philosophical difference is that these aforementioned works are aimed at speeding up/parallelizing very specific kinds of linkage based algorithms, while recovering the same or similar cluster trees (under very different notions of similarity) that would have been computed by the slower/sequential algorithm. Moreover, the specific algorithms considered in these works have no known approximation guarantees for Dasgupta's objective. Our work on the other hand approaches this problem from an optimization perspective. Through data sparsification, we aim to recover a cluster tree with marginal loss in objective function value as compared to one computed over the entire (dense) input data by *any given HC algorithm* as a blackbox, achieving a speedup in runtime or reducing its memory requirement due to sparsity. [32] studied the hierarchical clustering problem in an MPC setting. However, their work only considered the maximization objectives [39, 19], while our work is primarily focussed on the minimization objective of [20].

**Recent Independent work:** Very recently and independent of our work, [5] considered the problem of hierarchical clustering under Dasgupta's objective in the (dynamic) streaming setting. The primary focus of their work is on estimating the cost of an optimal hierarchy in $o(n)$ space whereas the focus of our work is to actually output a near-optimal hierarchical clustering. [5] also gives a single-pass, $\widetilde{O}(n)$ memory streaming algorithm for finding a clustering that also uses cut-sparsifiers. However, their algorithm needs to restrict the solution space to only balanced trees, and hence, is only able to achieve an $O(\phi)$ approximation guarantee instead of the stronger $(1 + o(1)) \cdot \phi$ approximation guarantee that we achieve for the streaming setting. They also obtain a 2-round algorithm for MPC that achieves an $O(\phi)$ approximation guarantee using $\widetilde{O}(n)$ memory per machine, which is again slightly weaker than the $(1 + o(1))\phi$ approximation guarantee that we achieve. But [5] does not show any lower bound results for the MPC model. Moreover, their work does not consider sublinear time setting and their algorithms cannot be easily adapted to this setting.

**More Related work:** The techniques we use in our paper can also be used to compute an expander decomposition of a graph presented in a dynamic stream. In particular, [25] showed that computing an expander decomposition reduces to computing the so-called *power cut sparsifier* of a graph, which approximates every cut to within a small multiplicative error plus some additive error that is proportional to the *volume* of the smaller side of the cut. In this work, we show how to obtain cut sparsifiers with similar guarantees except that (i) our additive error is proportional to the *number of vertices* of the smaller side of the cut, and (ii) our approach does *not* work for graphs with self-loops, which is required for the reduction given in [25]. Here (i) is actually a stronger guarantee, since in a graph without isolated vertices, the volume of a subset of vertices is at least its size, whereas (ii) is a drawback for this application. However, one can in fact trade off these two by slightly changing the analysis of our algorithm, while keeping the algorithm exactly the same, on which we elaborate below.

Specifically, our analysis is done by doing the following thought experiment - add a (weighted) constant degree expander to the original graph, and then show that the effective resistance of each edge $(u, v)$ in the original graph has become $\approx 1/d_u + 1/d_v$, with $d_u, d_v$ being the degree of $u, v$, and therefore sampling each edge with probability $\approx 1/d_u + 1/d_v$ gives a sparsifier of the composed graph, which is in turn a sparsifier with some additive error of the original graph. One can also adapt this analysis so that the same algorithm works for obtaining power cut sparsifiers. In particular, in our thought experiment, we will now add an expander where each vertex has degree (roughly) proportional to its original degree, and can again show that the effective resistance can be bounded by $\approx 1/d_u + 1/d_v$ even if there are self-loops in the original graph. Therefore the same algorithm works, with the additive error now being proportional to the volume of the smaller side, as desired. Finally, as observed in [25], sampling edges with probability $\approx 1/d_u + 1/d_v$ can be implemented in a straightforward manner in dynamic streams.

## 5 Implications for Other Hierarchical Clustering Objectives

As noted in the previous section, two *maximization* objectives for hierarchical clustering were proposed subsequent to the work of [20]: (1) the *revenue* objective [39] for similarity-based HC which is a "dual" of $\mathsf{cost}_G(\mathcal{T})$, (2) the *dissimilarity* objective [19] where the edges correspond to pairwise *dissimilarities* and the objective is the same as $\mathsf{cost}_G(\mathcal{T})$. In this section, we discuss the implications of our work for both objectives in the sublinear-resource regime.

We begin by noting a sharp contrast in the difficulty of achieving a "good" solution for the minimization objective [20], and the two maximization objectives described above. In fact, it is possible to achieve a $O(1)$ approximation to both maximization objectives *non-adaptively*; a *random* binary hierarchy, in expectation, is a $1/3$ approximation of the optimal revenue [39], and is a $2/3$ approximation of the optimal dissimilarity objective [19], constructing which requires no knowledge of the input graph. On the other hand, it is not hard to see that one would achieve an arbitrarily bad approximation of the minimization objective unless something non-trivial was learned about the input graph.

That said, one can further question whether it is possible to match the solution quality of any given $\psi$-approximate offline algorithm for the maximization objectives in the models of computation we consider. We answer this in the affirmative; we can in fact achieve even stronger performance guarantees for both objectives in the sublinear resource regime by exploiting the fact that their corresponding optimal hierarchies have large objective function values[5], allowing us to tolerate even larger additive errors in our cut-sparsifiers. A straightforward application of our structural decomposition of the cost function along with its downstream implications in each of the three models of computation directly gives us $(\psi - o(1))$-approximate algorithms for both HC maximization objectives in weighted graphs that use $(i)$ a single-pass and $\widetilde{O}(n)$ space in the dynamic streaming model, $(ii)$ $\widetilde{O}(n)$ queries[6] in the general graph (query) model, $(iii)$ 2-rounds and $\widetilde{O}(n)$ communication in the MPC model, which can be further improved to use just $(iv)$ 1-round, and $\widetilde{O}(n)$ communication for unweighted graphs.

## 6   Conclusions and Future Directions

In this paper, we studied hierarchical clustering problem under Dasgupta's objective [20] in the regime of sublinear computational resources. We gave sublinear space, query, and communication algorithms for finding a $(1 + o(1))\phi$-approximate hierarchical clustering, where $\phi$ is the approximation ratio of any offline algorithm for this problem, and a sublinear time algorithm for finding an $O(\sqrt{\log n})$-approximate hierarchical clustering. At the core of our sublinear algorithms is a novel meta-algorithm which first obtains an $(\varepsilon, \delta)$-cut sparsifier of the graph and then runs hierarchical clustering algorithm on the sparsifier. We also proved sharp information-theoretic lower bounds showing that the performance of *all* our sublinear algorithms is essentially optimal for *any* polylog($n$)-approximation.

Note that all our algorithms and lower bounds are aimed at finding an explicit hierarchical clustering tree. Therefore a natural direction for future work is to understand whether we can get even more efficient sublinear algorithms if we only want to estimate the *cost* of the optimal hierarchical clustering to within some small error. We note that two recent and independent works have already considered this question. In the streaming model, [5] have proved extensive lower bounds for optimal cost estimation under various regimes of space and number of passes over the input stream. In the query model, [35] have proved both upper and lower bounds for sublinear time cost estimation under certain "well-clusterability" assumptions and given auxiliary information about the underlying input graph instance. However, this question remains open in the query model (sublinear time) for more general input graph instances, and completely unexplored in the MPC model (sublinear communication).

## Acknowledgments and Disclosure of Funding

This work was supported in part by NSF awards CCF-1763514, CCF-1934876, and CCF-2008305.

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
