# A    Notation and Preliminaries

We use the notation $G = (V, E)$ to represent unweighted graphs, and $G = (V, E, w)$ for weighted graphs. We use lowercase letters $u, v$ to refer to vertices in $V$, and given a vertex $v$, we use $d_G(v)$ to refer to its degree in graph $G$. We use capital letters $S, T$ to represent subsets of vertices, and given a vertex set $S \subset V$, we use $|S|$ to refer to its cardinality, $\overline{S} := V \setminus S$ to refer to its complement, and $G[S]$ to refer to the subgraph of $G$ induced by vertex set $S$. Furthermore, given two disjoint vertex sets $S, T$, we use $w_G(S, T) := \sum_{(u,v) \in E : u \in S, v \in T} w(u, v)$ to represent the total weight of the edges in graph $G$ with one endpoint in $S$ and the other in $T$. In the case of an unweighted graph, this is equivalent to the number of edges going from $S$ to $T$. For ease of notation, we use $w_G(S) := w_G(S, \overline{S})$, and when the implied graph is clear from context, $w_e$ to refer to the weight of an edge $e \in E$ in that graph.

Given a graph $G = (V, E)$, we use $\mathcal{T}$ to refer to a hierarchical clustering (tree) of the vertex set $V$, and $\mathsf{cost}_G(\mathcal{T})$ to refer to the cost of this clustering in graph $G$. Without loss of generality, we restrict our attention to just full binary hierarchical clustering trees, since the optimal tree is binary [20]. Any internal node $S$ of a hierarchical clustering tree corresponds to a binary split $(S_\ell, S_r)$ (the left and right children of $S$ in $\mathcal{T}$) of the set of leaves in the subtree rooted at $S$. With some overload of notation, we let $S$ represent both, the internal node of the clustering tree as well as the set of leaves $S \subseteq V$ in the subtree rooted at internal node $S$. Furthermore, since (the leaves in the subtree rooted at) an internal node $S$ can correspond to an arbitrary subset of $V$, we use the term *split* to refer to a partition $(S_\ell, S_r)$ of $S$ to disambiguate it from *cuts*, which are a partition of the entire vertex set $V$.

We conclude the preliminaries by presenting two useful facts from [20]; the first is an equivalent reformulation of the similarity based hierarchical clustering cost function defined earlier in the introduction, and the second is the cost of any hierarchical clustering in an unweighted clique.

**Fact 1.** *The hierarchical clustering cost of any tree $\mathcal{T}$ with each internal node $S$ corresponding to a binary split $(S_\ell, S_r)$ of the subset $S \subseteq V$ of vertices is equivalent to the sum*

$$\mathsf{cost}_G(\mathcal{T}) = \sum_{\text{splits } S \to (S_\ell, S_r) \text{ in } \mathcal{T}} |S| \cdot w_G(S_\ell, S_r) \,.$$

**Fact 2.** *The cost of any hierarchical clustering in an unweighted $n$-vertex clique is $(n^3 - n)/3$.*

# B    Hierarchical Clustering using $(\epsilon, \delta)$-Cut Sparsification

In this section, we shall present the key insight behind all our results: *the hierarchical clustering cost function can equivalently be viewed as a linear combination of global cuts in the graph*. As a consequence, approximately preserving cuts in the graph also approximately preserves the cost of hierarchies in the graph, effectively reducing the sublinear-resource hierarchical clustering problem to a cut-sparsification problem. However, there are some hard lower bounds that refute an efficient (sublinear) computation of traditional cut-sparsifiers in certain models of interest. Therefore, we begin by introducing a weaker notion of cut sparsification, which we call $(\epsilon, \delta)$-cut sparsification.

**Definition 1** ($(\epsilon, \delta)$-cut sparsifier)**.** *Given a weighted graph $G = (V, E, w)$ and parameters $\epsilon, \delta \geq 0$, we say that a weighted graph $\widetilde{G} = (V, \widetilde{E}, \widetilde{w})$ is an $(\epsilon, \delta)$-cut sparsifier of $G$ if for all cuts $S \subset V$,*

$$(1 - \epsilon)w_G(S) \leq w_{\widetilde{G}}(S) \leq (1 + \epsilon)w_G(S) + \delta \min\{|S|, |\overline{S}|\}$$

The above is a generalization of the usual notion of cut-sparsifiers (which are $(\epsilon, 0)$-cut sparsifiers as per the above definition) that allows for an additive error in addition to the usual multiplicative error in any cut of the graph. A variant of this idea has been proposed before under the term *probabilistic $(\epsilon, \delta)$-spectral sparsifiers* in [36] which was similarly motivated by designing sublinear time algorithms for (single) cut problems on unweighted graphs. However, as the name might suggest, the key difference between the prior work and ours is that the above bounds on the cut-values hold only in expectation (or any given constant probability) for any fixed cut in the former. Due to this limitation, we cannot use this previous work in a blackbox, and new ideas are needed.

We now show that for any two graphs that are close in this $\epsilon, \delta$ sense, the cost of any hierarchy in these two graphs is also close as a function of these parameters, effectively allowing the use of $(\epsilon, \delta)$-cut sparsifiers in a *blackbox*.

**Lemma 1.** *Given any input weighted graph $G = (V, E, w)$ on $n$ vertices, and an $(\epsilon, \delta)$-cut sparsifier $\widetilde{G}$ of $G$, then for any hierarchy $\mathcal{T}$ over the vertex set $V$, we have*

$$(1 - \epsilon)\mathsf{cost}_G(\mathcal{T}) \leq \mathsf{cost}_{\widetilde{G}}(\mathcal{T}) \leq (1 + \epsilon)\mathsf{cost}_G(\mathcal{T}) + \frac{n(n + 1)\delta}{2} \,.$$

*Therefore, running a $\phi$-approximate hierarchical clustering oracle $\mathcal{A}$ with input as the sparsifier $\widetilde{G}$ with $\epsilon \leq 1/2$ produces a hierarchical clustering $\mathcal{T}_{\mathcal{A}}$ whose cost in $G$ is at most*

$$\mathsf{cost}_G(\mathcal{T}_{\mathcal{A}}) \leq (1 + 4\epsilon)\phi \cdot \mathsf{cost}_G(\mathcal{T}^*) + n(n+1)\delta\phi,$$

*where $\mathcal{T}^*$ is an optimal hierarchical clustering of $G$.*

*Proof.* Consider any graph $H$ (not necessarily $G$ or $\widetilde{G}$) over vertex set $V$. Given any hierarchy $\mathcal{T}$ over the vertex set $V$, let $S^0$ be the root node with left and right children $S_\ell^0, S_r^0$, respectively. Then we have the the cost of this hierarchy in $H$ is given by

$$\mathsf{cost}_H(\mathcal{T}) = \sum_{S \to (S_\ell, S_r) \in \mathcal{T}} |S| \cdot w_H(S_\ell, S_r)$$

$$= |S^0| \cdot w_H(S_\ell^0, S_r^0) + \sum_{S \to (S_\ell, S_r) \in \mathcal{T}, S \neq S^0} |S| \cdot w_H(S_\ell, S_r).$$

Now observe that since the split at the root $S^0$ is a partition of the entire vertex set $V$ into $S_\ell^0, S_r^0$, we have $w_H(S_\ell^0, S_r^0) = w_H(S_\ell^0) = w_H(S_r^0)$. Furthermore, observe that for any split of $S$, $S_\ell \cup S_r = S$, and therefore, we can represent the total weight of the edges crossing the split $w_H(S_\ell, S_r) = (1/2) \cdot (w_H(S_\ell) + w_H(S_r) - w_H(S))$. Therefore,

$$\mathsf{cost}_H(\mathcal{T}) = \frac{|S^0|}{2}(w_H(S_\ell^0) + w_H(S_r^0)) + \sum_{S \to (S_\ell, S_r) \in \mathcal{T}, S \neq S^0} \frac{|S|}{2}(w_H(S_\ell) + w_H(S_r) - w_H(S))$$

$$= \sum_{S \to (S_\ell, S_r) \in \mathcal{T}} \left( \frac{|S| - |S_\ell|}{2} w_H(S_\ell) + \frac{|S| - |S_r|}{2} w_H(S_r) \right) + \sum_{v \in V} \frac{1}{2} w_H(v)$$

$$= \frac{1}{2} \cdot \left( \sum_{S \to (S_\ell, S_r) \in \mathcal{T}} (|S_r| \cdot w_H(S_\ell) + |S_\ell| \cdot w_H(S_r)) + \sum_{v \in V} w_H(v) \right),$$

Therefore, the hierarchical clustering cost function can equivalently be represented as a non-negative weighted sum of cuts in a graph. We shall now use this reformulation of the clustering cost function to bound the error in the cost of any hierarchy $\mathcal{T}$ over a graph $G$ and its $(\epsilon, \delta)$-sparsifier $\widetilde{G}$ as a function of the error in the cuts in these two graphs, which is parameterized by $\epsilon, \delta$. Our claimed lower bound is easy to see since for every cut $(S, \overline{S})$, $w_{\widetilde{G}}(S) \geq (1 - \epsilon)w_G(S)$, and therefore,

$$\mathsf{cost}_{\widetilde{G}}(\mathcal{T}) \geq \frac{(1-\epsilon)}{2} \cdot \left( \sum_{S \to (S_\ell, S_r) \in \mathcal{T}} (|S_r| \cdot w_G(S_\ell) + |S_\ell| \cdot w_G(S_r)) + \sum_{v \in V} w_G(v) \right) = (1-\epsilon)\mathsf{cost}_G(\mathcal{T}).$$

To show the upper bound, we have

$$\mathsf{cost}_{\widetilde{G}}(\mathcal{T}) \leq \frac{(1+\epsilon)}{2} \cdot \left( \sum_{S \to (S_\ell, S_r) \in \mathcal{T}} (|S_r| \cdot w_G(S_\ell) + |S_\ell| \cdot w_G(S_r)) + \sum_{v \in V} w_G(v) \right)$$

$$+ \frac{\delta}{2} \cdot \left( \sum_{S \to (S_\ell, S_r) \in \mathcal{T}} (|S_r| \cdot \min\{|S_\ell|, |\overline{S_\ell}|\} + |S_\ell| \cdot \min\{|S_r|, |\overline{S_r}|\}) + n \right)$$

$$\leq (1+\epsilon)\mathsf{cost}_G(\mathcal{T}) + \delta \cdot \left( \frac{n}{2} + \sum_{S \to (S_\ell, S_r) \in \mathcal{T}} |S_\ell| \cdot |S_r| \right).$$

Finally, we claim that for any binary hierarchical clustering tree $\mathcal{T}$ over $n$ vertices (leaves),

$$\sum_{S \to (S_\ell, S_r) \in \mathcal{T}} |S_\ell| \cdot |S_r| \leq \frac{n^2}{2}$$

We shall prove this claim by induction on the number of leaves of $\mathcal{T}$. The base case is easy to see, which is a binary tree on 2 leaves. Assuming this claim holds for all binary trees on $n' < n$ leaves, consider any binary tree $\mathcal{T}$ with $n$ leaves. Suppose the split at the root partitions the set of $n$ leaves $S^0$ into sets $S_\ell^0$ and $S_r^0$. Let $\mathcal{T}_\ell, \mathcal{T}_r$ be the subtrees of $\mathcal{T}$ rooted at $S_\ell^0, S_r^0$, respectively. Then we have

$$\sum_{S \to (S_\ell, S_r) \in \mathcal{T}} |S_\ell| \cdot |S_r| = |S_\ell^0| \cdot |S_r^0| + \sum_{S \to (S_\ell, S_r) \in \mathcal{T}_\ell} |S_\ell| \cdot |S_r| + \sum_{S \to (S_\ell, S_r) \in \mathcal{T}_r} |S_\ell| \cdot |S_r|.$$

Since both $|S_\ell^0|, |S_r^0| < n$, applying our induction hypothesis on the subtrees $\mathcal{T}_\ell, \mathcal{T}_r$ gives us that

$$\sum_{S \to (S_\ell, S_r) \in \mathcal{T}_\ell} |S_\ell| \cdot |S_r| \leq \frac{|S_\ell^0|^2}{2}, \quad \text{and} \quad \sum_{S \to (S_\ell, S_r) \in \mathcal{T}_r} |S_\ell| \cdot |S_r| \leq \frac{|S_r^0|^2}{2}.$$

Substituting these bounds on the above sum proves our claim as

$$\sum_{S \to (S_\ell, S_r) \in \mathcal{T}} |S_\ell| \cdot |S_r| \leq |S_\ell^0| \cdot |S_r^0| + \frac{|S_\ell^0|^2 + |S_r^0|^2}{2} = \frac{|S_\ell^0 + S_r^0|^2}{2} = \frac{|S^0|^2}{2} = \frac{n^2}{2}.$$

Finally, observe that the $\phi$-approximate hierarchical clustering oracle on input $\widetilde{G}$ finds a tree $\mathcal{T}_\mathcal{A}$ such that

$$\mathsf{cost}_{\widetilde{G}}(\mathcal{T}_\mathcal{A}) \leq \phi \cdot \mathsf{cost}_{\widetilde{G}}(\mathcal{T}), \qquad \forall \text{ hierarchies } \mathcal{T}. \tag{2}$$

Applying the above bound with $\mathcal{T} = \mathcal{T}^*$, an optimal hierarchical clustering of $G$ gives us that

$$(1-\epsilon)\mathsf{cost}_G(\mathcal{T}_\mathcal{A}) \overset{\text{Lem 1}}{\leq} \mathsf{cost}_{\widetilde{G}}(\mathcal{T}_\mathcal{A}) \overset{\text{Eq 2}}{\leq} \phi \cdot \mathsf{cost}_{\widetilde{G}}(\mathcal{T}^*) \overset{\text{Lem 1}}{\leq} (1+\epsilon)\phi \cdot \mathsf{cost}_G(\mathcal{T}^*) + \frac{n(n+1)\delta\phi}{2}.$$

Therefore, for $\epsilon \leq 1/2$, we have that

$$\mathsf{cost}_G(\mathcal{T}_\mathcal{A}) \leq \frac{1+\epsilon}{1-\epsilon}\phi \cdot \mathsf{cost}_G(\mathcal{T}^*) + \frac{n(n+1)}{2(1-\epsilon)}\delta\phi \leq (1+4\epsilon)\phi \cdot \mathsf{cost}_G(\mathcal{T}^*) + n(n+1)\delta\phi.$$

$\square$

The above result shows that these weaker cut sparsifiers also approximately preserve the cost of any hierarchical clustering, but only up to an additive $O(\delta n^2)$ factor. Therefore, supposing we could efficiently estimate a lower bound OPT on the cost of an optimal hierarchical clustering in a graph $G$, we could then set the additive error $\delta = \epsilon \mathrm{OPT}/n^2$, giving us that any $\phi$-approximate hierarchical clustering for $\widetilde{G}$ is a $(1+5\epsilon)\phi$-approximate hierarchical clustering for $G$. This implies that hierarchical clustering is effectively equivalent to efficiently computing an $(\epsilon, \delta)$-cut sparsifier with a sufficiently small additive error $\delta$.

The following result fills in the final missing link in our chain of ideas by establishing a general-purpose lower bound on the cost of any hierarchical clustering in an unweighted graph as a function of the number of vertices and edges in the graph.

**Lemma 2.** *Let $G$ be any unweighted graph on $n$ vertices and $m$ edges. Then the cost of any hierarchical clustering in $G$ is at least $4m^2/(3n)$.*

*Proof.* Given any unweighted graph $G = (V, E)$ over $n$ vertices and $m$ edges, fix any hierarchy $\mathcal{T}$ of the vertices $V$. In order to lower bound the cost of $\mathcal{T}$, we shall iteratively modify the "base graph" graph $G$ by moving edges, strictly reducing the cost of $\mathcal{T}$ with each modification such that the final graph has a structure that makes the hierarchical clustering cost of $\mathcal{T}$ easy to analyze. In particular, the final graph would be such that each connected component is either a clique or two cliques connected together by some number of edges.

This is done as follows: given any hierarchy $\mathcal{T}$ of $V$, we perform a level order traversal over the internal nodes of $\mathcal{T}$, and at each node $S$, we modify the graph by pushing edges crossing the split $(S_\ell, S_r)$ down to lower level splits. Formally, let $S^1, \cdots, S^{n-1}$ be a level-order traversal over internal nodes of $\mathcal{T}$. We denote by $G^t = (V, E^t)$ the modified graph after visiting internal node $S^t$, with $G^0 = G$. Given $G^t$, we visit $S^{t+1}$ and modify the graph as follows: if the subgraphs $G^t[S_\ell^{t+1}], G^t[S_\ell^{t+1}]$ induced by vertex sets $S_\ell^{t+1}, S_r^{t+1}$ respectively, are both cliques, then $G^{t+1} = G^t$; else move a maximal number of (arbitrary) edges crossing the split $(S_\ell^{t+1}, S_r^{t+1})$ to any (arbitrary) edge slots that are available in subgraphs $G^t[S_\ell^{t+1}], G^t[S_r^{t+1}]$ until either (a) the split $(S_\ell^{t+1}, S_r^{t+1})$ has no more edges going across in which case the two subgraphs become disconnected, or (b) both of the subgraphs become cliques with the edges remaining going across these cliques. We call the resulting graph $G^{t+1}$. Observe that the cost of $\mathcal{T}$ in $G^{t+1}$ is at most the cost of $\mathcal{T}$ in $G^t$.

Let the final graph obtained after this traversal be $G^{n-1}$. It is easy to see that $G^{n-1}$ is a collection of connected components, with each connected component being either a clique or two cliques with edges going across them, and that $\mathsf{cost}_{G^{n-1}}(\mathcal{T}) \leq \mathsf{cost}_G(\mathcal{T})$. In this graph $G^{n-1}$, (1) let $k_1, \ldots, k_r$ be the cliques, with $k_j$ being the number of vertices in clique $j$, and (2) let $t_1, \ldots t_s$ be the connected components that are two cliques connecting by edges, where each $t_i = \{k_{i,1}, k_{i,2}, c_i\}$ with $k_{i,1}, k_{i,2}$ being the number of vertices in the two cliques of component $t_i$, and $c_i < k_{i,1} \cdot k_{i,2}$ being the number of edges going across the two cliques. Then the cost of $\mathcal{T}$ on $G^{n-1}$ is given by

$$\mathsf{cost}_{G^{n-1}}(\mathcal{T}) = \sum_{j=1}^{r} \frac{k_j^3 - k_j}{3} + \sum_{i=1}^{s} \left( \frac{k_{i,1}^3 - k_{i,1}}{3} + \frac{k_{i,2}^3 - k_{i,2}}{3} + (k_{i,1} + k_{i,2})c_i \right), \tag{3}$$

which follows by construction of $G^{n-1}$ and Fact 2. We also observe that

$$n = \sum_{j=1}^{r} k_j + \sum_{i=1}^{s} (k_{i,1} + k_{i,2}), \text{ and}$$

$$m = \sum_{j=1}^{r} \binom{k_j}{2} + \sum_{i=1}^{s} \left( \binom{k_{i,1}}{2} + \binom{k_{i,2}}{2} + c_i \right). \tag{4}$$

With these three observations, we shall now prove our claimed lower bound. We have that

$$m^2 = \left[ \sum_{j=1}^{r} \binom{k_j}{2} + \sum_{i=1}^{s} \left( \binom{k_{i,1}}{2} + \binom{k_{i,2}}{2} + c_i \right) \right]^2$$

$$= \left[ \sum_{j=1}^{r} k_j^{1/2} \left[ k_j^{-1/2} \binom{k_j}{2} \right] + \sum_{i=1}^{s} k_{i,1}^{1/2} \left[ k_{i,1}^{-1/2} \left( \binom{k_{i,1}}{2} + \frac{c_i}{2} \right) \right] + \sum_{i=1}^{s} k_{i,2}^{1/2} \left[ k_{i,2}^{-1/2} \left( \binom{k_{i,2}}{2} + \frac{c_i}{2} \right) \right] \right]^2$$

$$\overset{(a)}{\leq} \left[ \sum_{j=1}^{r} k_j + \sum_{i=1}^{s} k_{i,1} + \sum_{i=1}^{s} k_{i,2} \right] \left[ \sum_{j=1}^{r} \frac{1}{k_j} \binom{k_j}{2}^2 + \sum_{i=1}^{s} \frac{1}{k_{i,1}} \left( \binom{k_{i,1}}{2} + \frac{c_i}{2} \right)^2 + \sum_{i=1}^{s} \frac{1}{k_{i,2}} \left( \binom{k_{i,2}}{2} + \frac{c_i}{2} \right)^2 \right]$$

$$\overset{(b)}{=} \frac{n}{4} \left[ \sum_{j=1}^{r} k_j(k_j-1)^2 + \sum_{i=1}^{s} \left( k_{i,1}(k_{i,1}-1)^2 + k_{i,2}(k_{i,2}-1)^2 + 2c_i(k_{i,1}+k_{i,2}-2) + c_i^2(k_{i,1}^{-1}+k_{i,2}^{-1}) \right) \right]$$

$$\overset{(c)}{<} \frac{n}{4} \left[ \sum_{j=1}^{r} k_j(k_j-1)^2 + \sum_{i=1}^{s} \left( k_{i,1}(k_{i,1}-1)^2 + k_{i,2}(k_{i,2}-1)^2 + 3c_i(k_{i,1}+k_{i,2}) \right) \right]$$

$$< \frac{3n}{4} \left[ \sum_{j=1}^{r} \frac{k_j(k_j-1)(k_j+1)}{3} + \sum_{i=1}^{s} \left( \frac{k_{i,1}(k_{i,1}-1)(k_{i,1}+1)}{3} + \frac{k_{i,2}(k_{i,2}-1)(k_{i,2}+1)}{3} + c_i(k_{i,1}+k_{i,2}) \right) \right]$$

$$\overset{(d)}{=} \frac{3n}{4} \mathsf{cost}_{G^{n-1}}(\mathcal{T}),$$

where $(a)$ follows by Cauchy-Schwarz inequality, $(b)$ follows by Eq. (4), $(c)$ follows by observing $c_i < k_{i,1} \cdot k_{i,2}$ due to which $c_i^2(k_{i,1}^{-1} + k_{i,2}^{-1}) < c_i(k_{i,1}+k_{i,2})$, and $(d)$ follows from the cost of hierarchical clustering $\mathcal{T}$ in $G^{n-1}$ established in Eq. (3). Therefore, we have that

$$\frac{4m^2}{3n} < \mathsf{cost}_{G^{n-1}}(\mathcal{T}) \leq \mathsf{cost}_{G}(\mathcal{T}),$$

for any hierarchical clustering $\mathcal{T}$ in any graph $G$ on $n$ vertices and $m$ edges. □

We conclude this section with a remark about one particular instantiation of a $\phi$-approximation oracle for hierarchical clustering. Specifically, [19] showed that the recursive sparsest cut algorithm, i.e. recursively splitting the vertices using either the uniform sparsest cut or the balanced cut (sparsest cut that breaks the graph into two roughly equal parts) in the subgraph induced by the vertices, is a $6.75\gamma$-approximation to hierarchical clustering given access to a $\gamma$-approximation algorithm for sparsest cut or balanced cut. The best known polynomial time approximation for either is $O(\sqrt{\log n})$, a celebrated result due to [4]. These results in combination give us the following corollary.

**Corollary 1.** *Given any input weighted graph $G = (V, E)$ on $n$ vertices, and an $(\epsilon, \delta)$-cut sparsifier $\widetilde{G}$ of $G$ for any constant $0 \leq \epsilon \leq 1/2$ and a sufficiently small $0 \leq \delta$, there exists a polynomial time algorithm that given sparsifier $\widetilde{G}$ as the input, finds a hierarchical clustering $\mathcal{T}$ whose cost in $G$ is at most $O(\sqrt{\log n}) \cdot \mathsf{cost}_G(T^*)$, where $T^*$ is the optimal hierarchical clustering in $G$.*

In the following sections, we use this idea of constructing $(\epsilon, \delta)$-cut sparsifiers in three well-studied models for sublinear computation: the streaming model for sublinear space, the query model for sublinear time, and the MPC model for sublinear communication.

## C  Sublinear Space Algorithms in the Streaming Model

We first consider the space bounded setting in the dynamic streaming model, where the input graph is presented as an arbitrary sequence of edge insertions and deletions. The objective is to compute a good hierarchical clustering of the input graph given $O(n \, \mathrm{polylog}(n))$ memory, which is sublinear in the number of edges in the graph (referred to as a *semi-streaming* setting). The following theorem describes the main result of this setting.

**Theorem 10.** *Given any weighted graph $G = (V, E, w)$ with $n$ vertices and the edges of the graph presented in a dynamic stream, a parameter $0 < \epsilon \leq 1/2$, and a $\phi$-approximation oracle for hierarchical clustering, there exists a single-pass semi-streaming algorithm that finds a $(1 + \epsilon)\phi$-approximate hierarchical clustering of $G$ with high probability using $\widetilde{O}(\epsilon^{-2}n)$ space.*

This result is a direct consequence of Lemma 1 and known results [29, 1] for constructing an $(\epsilon, 0)$-cut sparsifiers in single-pass dynamic streams using polynomial time and $\widetilde{O}(\epsilon^{-2}n)$ space. Lastly, Corollary 1 gives us a complete polynomial time single-pass semi-streaming algorithm that finds an $\widetilde{O}(1)$-approximate hierarchical clustering of the input graph in $\widetilde{O}(n)$ space in a dynamic stream.

# D   Sublinear Time Algorithms in the Query Model

We now move our attention to the bounded time setting in the general graph (query) model [31], where the input graph is accessible via the following two[7] queries: (a) Degree queries: given $v \in V$, returns the degree $d_G(v)$, and (b) Neighbour queries: given $v \in V$, $i \leq d_G(v)$, returns the $i^{th}$ neighbour of $v$ (neighbours are ordered arbitrarily). The objective is to compute a good hierarchical clustering of the input graph in time and queries sublinear in the number of edges in the graph. This problem becomes substantially more interesting in this setting, as finding an $(\epsilon, 0)$-cut sparsifier necessarily takes linear $\Omega(n + m)$ queries. Therefore, the key to achieving such a result crucially depends upon being able to efficiently construct these weaker $(\epsilon, \delta)$-cut sparsifiers with a small additive error $\delta$, which is the backbone of our sublinear time result. For simplicity, we begin by presenting our result for unweighted graphs, and then extend it to weighted graphs in subsection D.2.

**Theorem 11.** *Given any unweighted graph $G = (V, E)$ with $n$ vertices and $m = \alpha n^{4/3}$ edges accessible via queries in the general graph model, and any parameter $0 < \epsilon \leq 1/2$, there exists an algorithm that*

(a) *given a $\phi$-approximate hierarchical clustering oracle, finds a $(1 + \epsilon)\phi$-approximate hierarchical clustering of $G$ with high probability using $f(n, \alpha, \epsilon)$ queries, and*

(b) *given any arbitrarily small parameter $0 < \tau < 1/2$, finds an $O(\sqrt{\tau^{-1} \log n})$-approximate hierarchical clustering of $G$ with high probability using $\widetilde{O}(f(n, \alpha, \epsilon) + n^{1+\tau})$ time and queries, where*

$$f(n, \alpha, \epsilon) = \begin{cases} O\left(\alpha n^{4/3}\right) & \alpha < 1 \\ \widetilde{O}\left(\epsilon^{-3}(\alpha^{-2}n^{4/3} + n)\right) & \alpha \geq 1. \end{cases}$$

Note that unlike the sublinear space and communication settings, we cannot directly give a sublinear time $(1 + \epsilon)\phi$-approximation guarantee here; even though the rest of our algorithm (that constructs the $(\epsilon, \delta)$-cut sparsifier) has a sublinear time and query complexity, the running time of the $\phi$-approximate hierarchical clustering oracle to which we are given access can be arbitrarily large[8]. Therefore in this setting, we give a two-part result - the first is a *sublinear query*, $(1 + \epsilon)\phi$-approximation result, and the second is a *sublinear time and query*, $O(\sqrt{\log n})$-approximation result, which follows from a specific sublinear time implementation of a $\phi$-approximate hierarchical clustering oracle with $\phi = O(\sqrt{\log n})$.

The query (and time) complexity in the above result is linear in the number of edges for sparse graphs with fewer than $n\sqrt[3]{n}$ edges, decays as $\widetilde{O}(\alpha^{-2}n^{4/3})$ for moderately dense graphs when the number of edges is in the range $n\sqrt[3]{n}$ and $n\sqrt{n}$, and is $\widetilde{O}(n)$ for dense graphs with more than $n\sqrt{n}$ edges. As we will see in our lower bounds, this complexity is essentially optimal for achieving a $\widetilde{O}(1)$-approximation in each of these three regimes.

*Proof of Theorem 11.* The proof of both parts of Theorem 11 relies on $(\epsilon, \delta)$-cut sparsifiers, which we show in Theorem 12, can be constructed with high probability in $\widetilde{O}(\epsilon^{-2}\delta^{-1}n)$ time and queries. Assuming this construction, the sublinear query, $(1 + \epsilon)\phi$-approximation claim (Theorem 11 (a)) is relatively straightforward to see: we first determine the number of edges $m = \alpha n^{4/3}$ in the input graph by performing $n$ degree queries. If the graph is sufficiently sparse ($m \leq n^{4/3}$), then we simply read the entire graph, which takes $O(m)$ neighbour queries. If not, then the lower bound established in Lemma 2 implies that the cost OPT of any hierarchical

---

[7]As mentioned earlier, this model further allows for a third type of queries: (c) Pair queries: given $u, v \in V$, returns whether $(u, v) \in E$. This is equivalent to assuming the query oracle having internal access to both, an adjacency list representation (for degree and neighbour queries) as well as an adjacency matrix representation (for pair queries) of the input graph. However, our algorithm does not need pair queries, which further strengthens our algorithmic result in this model.

[8]Our sublinear query result more generally implies faster algorithms for hierarchical clustering without much loss in solution quality.

clustering in the input graph is at least $\alpha^2 n^{5/3}$. As a consequence, the additive error $\delta = \epsilon \text{OPT}/n^2 \geq \epsilon\alpha^2 n^{-1/3}$ we can tolerate in our $(\epsilon, \delta)$-sparsifier is also relatively large. Such a sparsifier can then be constructed with high probability in $\widetilde{O}(\epsilon^{-3} \max\{\alpha^{-2}n^{4/3}, n\})$ time and queries. The rest of the proof follows directly by Lemma 1, since the $\phi$-approximate hierarchical clustering oracle uses only the $(\epsilon, \delta)$-cut sparsifier as input, and therefore, makes no additional queries to the input graph.

To prove the sublinear time, $O(\sqrt{\tau^{-1}\log n})$-approximation claim (Theorem 11 (*b*)) where $\tau \in (0, 1/2)$ is any arbitrarily small parameter, we complement the above proof with an instantiation of a sublinear time, $\phi$-approximate hierarchical clustering oracle with $\phi = O(\sqrt{\tau^{-1}\log n})$. As discussed in Corollary 1, this essentially reduces to a sublinear time, $O(\sqrt{\tau^{-1}\log n})$-approximation algorithm for balanced cuts (also called *balanced separators* in the literature). However, the algorithm of [4] cannot be used here due to its quadratic running time. Therefore, we instead refer to another result of [44] that achieves $O(\sqrt{\tau^{-1}\log n})$-approximation for balanced separators by reducing this problem to $\widetilde{O}(n^{\tau})$ single-commodity max-flow computations for any given $\tau \in (0, 1/2)$. While [44] could only achieve this in $\widetilde{O}(m + n^{3/2+\tau})$ time, the bottleneck being the $\widetilde{O}(m^{3/2})$ time flow-computation algorithm of [30] (with the sparsification result of [7] being used to improve this complexity to $\widetilde{O}(n^{3/2})$), we can leverage a very recent breakthrough [14] that gives an $\widetilde{O}(m^{1+o(1)})$ algorithm for exact single-commodity max-flows. This improves the running time of the algorithm of [44] from $\widetilde{O}(m + n^{3/2+\tau})$ to $\widetilde{O}(m + n^{1+\tau})$ without any loss in the approximation factor. Since our $(\epsilon, \delta)$-cut sparsifier $\widetilde{G}$ is very sparse with $f(n, \alpha, \epsilon)$ edges, we can find a $O(\sqrt{\tau^{-1}\log n})$-approximate balanced separator in $\widetilde{G}$ in sublinear $\widetilde{O}(f(n, \alpha, \epsilon) + n^{1+\tau})$-time, for any given $\tau \in (0, 1/2)$. Although we use this subroutine repeatedly (at each split of the graph until we are left with singleton vertices), observe that at any level of the hierarchical clustering tree, the splits at that level together form a *disjoint partition* of $\widetilde{G}$. Now let the set of all internal nodes (splits) of the resultant hierarchical clustering tree at level $i \in [d]$ be $\mathcal{S}_i$, where $d$ is the depth of the tree. Furthermore, for any internal node $S$ in this tree, let $m_S$ be the number of edges in the subgraph $\widetilde{G}[S]$ induced by the set of vertices $S$. Therefore, the running time of the recursive sparsest cut algorithm on $\widetilde{G}$ with the aforementioned $O(\sqrt{\tau^{-1}\log n})$-approximate oracle for balanced separators is given by

$$\widetilde{O}\left(\sum_{i \in [d]} \sum_{S \in \mathcal{S}_i} (m_S + |S|^{1+\tau})\right) \leq \widetilde{O}\left(\sum_{i \in [d]} f(n, \alpha, \epsilon) + n^{1+\tau}\right).$$

Finally, observe that since the splits in the tree are balanced, i.e. a split $S \to (S_\ell, S_r)$ is such that $\min\{|S_\ell|, |S_r|\} \geq |S|/3$, the depth of this hierarchical clustering tree produced $d = O(\log n)$, which gives the total running time of the recursive sparsest cut algorithm on $\widetilde{G}$ as $\widetilde{O}(f(n, \alpha, \epsilon) + n^{1+\tau})$, proving our sublinear time claim. □

We shall now present our sublinear time construction of $(\epsilon, \delta)$-cut sparsifiers for unweighted graphs.

### D.1 A Sublinear Time $(\epsilon, \delta)$-Cut Sparsification Algorithm for Unweighted Graphs

We note that the computation of graph sparsifiers in sublinear time was also studied previously for hypergraphs [16, 17].

**Theorem 12.** *There exists an algorithm that given a query access to an unweighted graph $G = (V, E)$ and parameters $0 < \delta \leq 1$, $0 < \epsilon \leq 1/2$, can find a $(\epsilon, \delta)$-cut sparsifier of $G$ with high probability in $\widetilde{O}(n\delta^{-1}\epsilon^{-2})$ time and queries.*

Our $(\epsilon, \delta)$-cut sparsifier construction broadly builds upon ideas developed in [36] for probabilistic spectral sparsifiers. At a high level, to achieve an additive error of $\delta$, we embed a constant-degree expander $G_x = (V, E_x)$ with edge weights $\delta \leq 1$ on all $n$ vertices in the input graph $G = (V, E)$. This trick gives a tight (and very friendly) bound on the effective resistance of every edge in the resultant composite graph $H = (V, E \cup E_x, w)$, which is a weighted graph with edge set consisting of the union of edges $E$ in the input graph $G$, each having weight 1, and edges $E_x$ in the constant-degree expander $G_x$, each with weight $\delta$ (edges in $E \cap E_x$ are assumed to be two parallel edges, one with weight 1 and the other with weight $\delta$). This is useful as it allows for efficient sparsification of this composite graph using effective resistance sampling of [45], with the sources of error being the usual multiplicative error due to sparsification itself, and a small additive error due to the few extra edges introduced by the expander. There are several well known $\widetilde{O}(n)$ time constructions of constant degree expanders, for example, a random $d$-regular graph is an expander with high probability [26]. This roughly outlines the sparsification algorithm and proof of the above theorem.

A similar idea was used in [36], with the key difference being that they embed an unweighted constant degree expander in a random $\delta n$ subset of vertices. Since the set of vertices where the expander is embedded is random, it is easy to see why this gives a small additive error in expectation for any fixed cut, but could be

very large for some cuts in the graph. Our construction on the other hand provides a sparsifier with stronger guarantees that hold for *every* cut. As outlined above, we start by showing that the effective resistance of any edge $(u, v) \in E \cup E_x$ is tightly bounded.

**Lemma 3.** *Given parameter $\delta \in (0, 1)$, and a composite graph $H = (V, E \cup E_x, w)$ where $G = (V, E)$ is an arbitrary input graph with edges of weight $1$, and $G_x = (V, E_x)$ is a constant-degree expander graph with edges of weight $\delta$, then for any edge $(u, v) \in E \cup E_x$, we have*

$$\frac{1}{2}\left(\frac{1}{d_G(u) + \delta d_{G_x}(u)} + \frac{1}{d_E(v) + \delta d_{G_x}(v)}\right) \le R_H(u, v) \le O\left(\frac{\log n}{\delta}\left(\frac{1}{d_G(u) + \delta d_{G_x}(u)} + \frac{1}{d_E(v) + \delta d_{G_x}(v)}\right)\right),$$

*where $R_H(u, v)$ is the effective resistance of edge $(u, v)$ in graph $H$, and for any vertex $u \in V$, $d_G(u), d_{G_x}(u)$ are the degrees of vertex $u$ in graphs $G$ and $G_x$, respectively.*

*Proof.* For any edge $(u, v) \in E \cup E_x$, let us assume without loss of generality that $k := d_H(u) \le d_H(v)$. The proof of our upper bound on the effective resistance $R_H(u, v)$ relies on a basic property of expander graphs: there are $\Omega(k)$ edge-disjoint paths, each of length at most $O(\log n)$ connecting $u$ to $v$. Since each edge on these paths has weight at least $\delta$, by Rayleigh's monotonicity principle, the effective resistance between $(u, v)$ can be no more than a graph containing exactly $k$ edge-disjoint paths, each of length $O(\log n)$ with each edge on this path having resistance $1/\delta$, which gives us that

$$R_H(u, v) \le O\left(\frac{\log n}{\delta k}\right) \le O\left(\frac{\log n}{\delta}\left(\frac{1}{d_H(u)} + \frac{1}{d_H(v)}\right)\right)$$
$$\le O\left(\frac{\log n}{\delta}\left(\frac{1}{d_G(u) + \delta d_{G_x}(u)} + \frac{1}{d_G(v) + \delta d_{G_x}(v)}\right)\right)$$

To show this many edge-disjoint, short paths property of expanders, we consider two possibilities: either $k < n/\log n$, in which case let $\{u_i\}_{i=1}^k$ be the neighbours of $u$, and let $\{v_i\}_{i=1}^k$ be a set of $k$ neighbours of $v$, chosen and ordered arbitrarily. Then the well known multicommodity flow result of [27] already guarantees the existence of these short edge-disjoint paths connecting every $u_i$ to $v_i$. In the case that $k \ge n/\log n$, consider the (unweighted) subgraph $H_x = (V, E_x \cup E_u \cup E_v)$ induced by the expander edges $E_x$ and edges $E_u, E_v \subseteq E$ incident on vertices $u, v$ in $G$, respectively. We first claim that the min $u$-$v$ cut in $H_x$ contains at least $k/2$ edges; let $(S, \overline{S})$ be the min $u$-$v$ cut, with $|S| \le n/2$ and $s \in \{u, v\}$ being the vertex in $S$. Furthermore, let $k_s \ge k$ be the number of neighbours of $s$, with $k'_s \le k_s$ of them being contained in $S$. Now there are two possibilities, either (a) $k'_s < k_s/2$ in which case the cut $(S, \overline{S})$ already contains the $k_s - k'_s \ge k_s/2$ edges connecting $s$ to its remaining neighbours in $\overline{S}$, or (b) $k'_s \ge k_s/2$ in which case $(S, \overline{S})$ must necessarily cut at least $|S| \ge k_s/2$ edges in $E_x$ due to expansion. Therefore, by the (integral) min-cut max-flow theorem, there are at least $k/2$ edge-disjoint paths from $u$ to $v$. Moreover, we claim that at least half of these paths must be short, specifically, of length $O(\log n)$. To see this, observe that graph $H_x$ contains just $Cn$ edges for some constant $C$, which follows from that fact that $u, v$ each can have at most $n$ neighbours and $G_x$ is a constant degree expander. Now let the integral flow which is of size $f \ge k/2 \ge n/(2 \log n)$ be routed along arbitrary edge-disjoint paths $P_1, \ldots, P_f$. It is easy to see why at least $f/2$ of these paths must be of length at most $2C \log n$, because otherwise, the number of edges contained in just the long paths alone would exceed $(f/2) \cdot (2C \log n) > Cn$, the total number of edges in $H_x$ which is a contradiction. Therefore, there are at least $k/4$ edge disjoint paths between $u, v$ in $H_x \subseteq H$, each of length $O(\log n)$.

Now to prove the lower bound on the effective resistance of any edge $(u, v) \in E \cup E_x$, we add an extra vertex $w$ and replace the edge $(u, v)$ with two edges $(u, w)$ and $(w, v)$, each with weight/capacitance $2w_{uv}$ (doing this twice if edge $(u, v) \in E \cap E_x$, once for the edge $(u, v)$ with $w_{uv} = 1$ and again for the edge with $w_{uv} = \delta$). We then apply Rayleigh's monotonicity principle by shorting all vertices other than $u, v$ in the graph, which gives us that

$$R_H(u, v) \ge \frac{1}{d_G(u) + \delta d_{G_x}(u) + w_{uv}} + \frac{1}{d_G(v) + \delta d_{G_x}(v) + w_{uv}}$$
$$\ge \frac{1}{2}\left(\frac{1}{d_G(u) + \delta d_{G_x}(u)} + \frac{1}{d_G(v) + \delta d_{G_x}(v)}\right),$$

where the final inequality follows from the fact that $w_{uv} < \min_{s \in \{u,v\}}\{d_G(s) + \delta d_{G_x}(s)\}$, which proves our claimed lower bound. $\square$

This tight bound of the order $(d_G(u) + \delta d_{G_x}(u))^{-1} + (d_G(v) + \delta d_{G_x}(v))^{-1}$ on the effective resistances directly allows us to apply the effective resistance sampling scheme of [45] outlined in Algorithm 1 to construct a spectral sparsifier of $H$, which is even stronger than the simple cut sparsifier we require. The following theorem then establishes the properties of the resulting sparsifier $\widetilde{G}$.

---

**Algorithm 1 Sparsify**

---

**Input**. Weighted graph $G = (V, E, w)$, edge sampling probabilities $p$ such that $\sum_{e \in E} p_e = 1$, repetitions $q$.

**Output**. Sparsifier $\widetilde{G} = (V, \widetilde{E}, \widetilde{w})$.

**for** $t = 1, \ldots, q$ **do**

    Sample a random edge $e \in E$ according to $p$. Add $e$ to $\widetilde{E}$ (if it does not already exist) and increase its weight $\widetilde{w}_e$ by $w_e/(q p_e)$.

**end for**

---

**Theorem 13** (Theorem 1 + Corollary 6 of [45])**.** *Given any weighted graph $G = (V, E, w)$ on $n$ vertices with Laplacian L, let $Z_e$ be numbers satisfying $Z_e \geq R_e/\alpha$ for some $\alpha \geq 1$ and $\sum_{e \in E} w_e Z_e \leq \sum_{e \in E} w_e R_e$. Then given any parameter $0 \leq \epsilon \leq 1$, the subroutine* **Sparsify**$(G, p, q)$ *with sampling probabilities $p_e = w_e Z_e/(\sum_{e \in E} w_e Z_e)$ and $q = Cn \log n/\epsilon^2$ for some sufficiently large constant $C$ returns a graph $\widetilde{G}$ whose Laplacian $\widetilde{L}$, with high probability, satisfies*

$$\forall x \in \mathbb{R}^n \quad (1 - \epsilon\sqrt{\alpha})x^\top L x \leq x^\top \widetilde{L} x \leq (1 + \epsilon\sqrt{\alpha})x^\top L x.$$

From the effective resistance bound established in Lemma 3, it is easy to see that sampling edges with parameter $Z_{uv} = (d_G(u) + \delta d_{G_x}(u))^{-1}/2 + (d_G(v) + \delta d_{G_x}(v))^{-1}/2$ satisfies the condition in Theorem 13 with $\alpha = O(\log n/\delta)$ for the graph $H = (V, E \cup E_x, w)$. Given this choice of parameters $Z_e$, it is easy to see that $\sum_{e \in E \cup E_x} w_e Z_e = n/2$, which gives us that the sampling probability for any edge $e \in E \cup E_x$ is given by

$$p_e = \frac{w_e}{n} \left( \frac{1}{d_G(u) + \delta d_{G_x}(u)} + \frac{1}{d_G(v) + \delta d_{G_x}(v)} \right), \tag{5}$$

for which there is a very simple sublinear time rejection sampling scheme given query access to $G$: sample a uniformly at random vertex $u \in V$, and toss a coin with bias $d_G(u)/(d_G(u) + \delta d_{G_x}(u))$ (degree query). If heads, sample a uniformly at random edge incident on $u$ in $G$ (neighbour query). Otherwise, sample a uniformly at random edge incident on $u$ in $G_x$. The complete algorithm is given below.

---

**Algorithm 2** $(\epsilon, \delta)$**-Sparsify**

---

**Input**. Unweighted graph $G = (V, E)$, parameters $0 < \delta \leq 1, 0 < \epsilon \leq 1$.

**Output**. Sparsifier $\widetilde{G} = (V, \widetilde{E}, \widetilde{w})$.

Construct a constant degree expander $G_x = (V, E_x)$.

Let $H = (V, E \cup E_x, w)$ be the composite weighted graph with edge weights $w_e = 1$ for $e \in E$ and $w_e = \delta$ for $e \in E_x$.

Set $\epsilon' = \epsilon\sqrt{\delta/(C_1 \log n)}$ for a sufficiently large constant $C_1$, repetitions $q = C_2 n \log n/(\epsilon')^2$ for a sufficiently large constant $C_2$.

Set sampling probabilities $p$, where for each edge $e \in E \cup E_x$, $p_e$ is as defined in Eq. (5).

Sparsifier $\widetilde{G} = $ **Sparsify**$(H, p, q)$

---

It is easy to see that the above algorithm produces a graph $\widetilde{G}$ with $O(n \log n/(\epsilon')^2) = O(n \log^2 n/(\delta\epsilon^2))$ edges, and runs in time $O(n \log n/(\epsilon')^2) = O(n \log^2 n/(\delta\epsilon^2))$. We shall now prove that $\widetilde{G}$ is an $(\epsilon, \delta)$-sparsifier of $G$ as claimed in Theorem 12.

*Proof of Theorem 12.* We start by observing that Theorem 13, by restricting to vectors $x \in \{0, 1\}^n$ (corresponding to partitions of $V$) and choice of $\epsilon' = \epsilon\sqrt{\delta/(C_1 \log n)}$ with a sufficiently large constant $C_1$, implies that with high probability, the sparsifier $\widetilde{G}$ produced by Algorithm 2 is such that

$$\forall S \subset V, \quad (1 - \epsilon)w_H(S) \leq w_{\widetilde{G}}(S) \leq (1 + \epsilon)w_H(S). \tag{6}$$

Now observe that for any cut $(S, \overline{S})$ in the composite graph $H$,

$$w_G(S) \leq w_H(S) = w_G(S) + w_{G_x}(S) \leq w_G(S) + \delta\Theta(\min\{|S|, |\overline{S}|\}),$$

where the final inequality follows by observing that the weight of any edge $e \in E_x$ is $\delta$, and since the degree of any vertex in $G_x$ is $\Theta(1)$, the number of edges in $G_x$ that cross any cut $(S, \overline{S})$ is $\Theta(\min\{|S|, |\overline{S}|\})$. Combining the above bounds with Eq. (6) gives us the $(\epsilon, \delta)$-cut sparsification guarantees for $\widetilde{G}$ as

$$\forall S \subset V, \quad (1 - \epsilon)w_G(S) \leq w_{\widetilde{G}}(S) \leq (1 + \epsilon)w_G(S) + \Theta(\delta) \cdot \min\{|S|, |\overline{S}|\}.$$

□

## D.2   Extension to Weighted Graphs

In this section, we extend our sublinear time results to weighted graphs $G = (V, E, w)$, where edges $e \in E$ take weights $1 \leq w_e \leq W$, where $W$ is an upper bound on the maximum edge weight. Since there is no universally accepted query model for weighted graphs, we propose the following *generalization* where the algorithm can make (*a*) Degree queries: given $v \in V$, returns the degree $d_G(v)$, and (*b*) Neighbour queries: given $v \in V$, $i \leq d_G(v)$, returns both the $i^{th}$ neighbour of $v$ and the connecting edge weight, with the additional constraint that the neighbours are ordered by increasing edge weights (neighbours connected by equal weight edges are ordered arbitrarily). Note that this generalization reduces to the general graph model when all edge weights are equal. The following theorem describes our upper bound in this more general setting.

**Theorem 14.** *Let $G = (V, E, w)$ be any weighted graph with $n$ vertices and edge weights taking values in a bounded range $[1, W]$. Given any parameter $0 < \epsilon \leq 1/3$, let $m_i = \alpha_i n^{4/3}$ be the number of edges in $G$ with weights in the interval $[(1 + \epsilon)^{i-1}, (1 + \epsilon)^i)$. Then given query access to $G$, there exists an algorithm that*

（a) *given a $\phi$-approximate hierarchical clustering oracle, finds a $(1 + \epsilon)\phi$-approximate hierarchical clustering of $G$ with high probability using $\widetilde{O}\left((\epsilon^{-1} \log W) \cdot (n + \max_i f(n, \alpha_i, \epsilon))\right)$ queries, and*

(b) *given any arbitrarily small parameter $0 < \tau < 1/2$, finds an $O(\sqrt{\tau^{-1} \log n})$-approximate hierarchical clustering of $G$ with high probability using $\widetilde{O}\left(n^{1+\tau} + (\epsilon^{-1} \log W) \cdot (n + \max_i f(n, \alpha_i, \epsilon))\right)$ time and queries, where*

$$f(n, \alpha, \epsilon) = \begin{cases} O\left(\alpha n^{4/3}\right) & \alpha < 1 \\ \widetilde{O}\left(\epsilon^{-3}(\alpha^{-2} n^{4/3} + n)\right) & \alpha \geq 1. \end{cases}$$

Before discussing this result, one might naturally ask whether this stricter requirement of ordering neighbours by weight is really necessary or whether it is possible to achieve a similar result for arbitrary or even random orderings. Towards the end of this section, we will show that this is unfortunately necessary; without the ordering constraint, no non-trivial approximation for hierarchical clustering is possible unless a constant fraction of the edges in the graph are queried, and this holds even if we were to additionally allow pair queries: given $u, v \in V$, return whether $(u, v) \in E$ (and edge weight $w_{uv}$ if affirmative).

At a high level, our sublinear time upper bound for weighted graphs is morally the same as that achieved in the unweighted case, with a $O(\epsilon^{-1} \log W)$ hit to query and time complexity. Algorithmically, we build upon the ideas developed for the unweighted case. We begin by partitioning the edge set $E$ of the input graph $G = (V, E, w)$ into weight classes, where the $i^{th}$ weight class consists of all edges $E_i$ with weights in the interval $[(1+\epsilon)^{i-1}, (1+\epsilon)^i)$. By construction, there are $\log_{(1+\epsilon)} W \leq 2\epsilon^{-1} \log W$ weight classes in total, with the edge sets $\{E_i\}_{i=1}^{\log_{(1+\epsilon)} W}$ being a disjoint partition of $E$. We then approximately sparsify each *unweighted* subgraph $G'_i = (V, E_i)$ using our sublinear time $(\epsilon, \delta)$-Sparsify routine outlined in the previous section, and scale up all the edge weights of the resultant sparsifier $\widetilde{G}'_i$ by the maximum edge weight $W_i = (1 + \epsilon)^i$ of that class. Since for every weight class $i$, the weights of all the edges $E_i$ in that class are within a $(1 + \epsilon)$ factor of each other, the resultant scaled sparsifier $\widetilde{G}_i$ is a good approximate sparsifier for the weighted subgraph $G_i = (V, E_i, w)$. Finally, since the input graph $G = (V, E, w)$ is partitioned into subgraphs $G_i = (V, E_i, w)$, the sum of the scaled sparsifiers $\widetilde{G}_i$ is a good sparsifier for the input graph. Given this sparsifier, the proof of (both claims of) Theorem 14 then follows identically as that of Theorem 11.

An important point to note here is that we do not need to explicitly construct the subgraphs $G'_i$ corresponding to each of the weight classes $i \in [\log_{1+\epsilon} W]$ (which would naively take $O(m)$ time) as our $(\epsilon, \delta)$-sparsification subroutine only requires query access to $G'_i$. This is easy to do in $\widetilde{O}(n)$ time for any weight class assuming the edges incident on vertices are sorted by weights. For any weight class $i$ and any vertex $v$, the set of edges incident on $v$ in subgraph $G_i$ lie in the range of indices $[x_{i-1}(v), x_i(v) - 1]$ where for any weight class $j \in [\log_{1+\epsilon} W]$, vertex $u \in V$, $x_j(u)$ is the first occurrence of an edge incident on $u$ with weight at least $(1+\epsilon)^j$. Both indices $x_i(v), x_{i-1}(v)$ can be found in $O(\log n)$ time and queries using binary search; the degree $d_{G'_i}(v) = x_i(v) - x_{i-1}(v)$, and the $j^{th}$ neighbour of $v$ in $G'_i$ is simply the $(x_{i-1}(v) + j - 1)^{th}$ neighbour of $v$ in $G$. Therefore, the total time and query complexity of setting up query access to $G'_i$ is $O(n \log n)$. We now present a formal proof of Theorem 14, which is achieved by Algorithm 3.

*Proof of Theorem 14.* As with the analysis for unweighted graphs, we begin by establishing a lower bound on the cost of any hierarchical clustering for weighted graphs. Given any weighted graph $G = (V, E, w)$ and a parameter $0 < \epsilon \leq 1/3$, we begin by partitioning the edge set into weight classes, where the $i^{th}$ weight class consists of all edges $E_i$ with weights in the interval $[(1+\epsilon)^{i-1}, (1+\epsilon)^i)$. Therefore, we have that the clustering

---

**Algorithm 3 Weighted Sparsify**

---

**Input**. Weighted graph $G = (V, E, w)$, parameter $0 < \epsilon \leq 1/3$.

**Output**. Sparsifier $\widetilde{G} = (V, \widetilde{E}, \widetilde{w})$.

For every vertex $v \in V$, $x_0(v) = 1$

**for** $i = 1, \ldots, \log_{(1+\epsilon)} W$ **do**

    For every vertex $v \in V$, binary search for $x_i(v)$, the first occurrence of an edge incident on $v$ with weight at least $(1 + \epsilon)^i$.

    Establish query access to $G_i' \leftarrow (V, E_i)$, where $E_i := \{e \in E : (1 + \epsilon)^{i-1} \leq w_e < (1 + \epsilon)^i\}$ using $\{(x_{i-1}(v), x_i(v))\}_{v \in V}$. Let $|E_i| = m_i = \alpha_i n^{4/3}$.

    **if** $\alpha_i \leq 1$ **then**

        Read $G_i = (V, E_i, w)$ entirely, and let this graph be $\widetilde{G}_i$.

    **else**

        Set additive error $\delta_i \leftarrow \epsilon \cdot \min\{\alpha_i^2/n^{1/3}, 1\}$, and $W_i = (1 + \epsilon)^i$.

        $\widetilde{G}_i' \leftarrow (\epsilon, \delta_i)$-**Sparsify**$(G_i')$, where $\widetilde{G}_i' = (V, \widetilde{E}_i, \widetilde{w}_i')$

        Construct sparsifier $\widetilde{G}_i = (V, \widetilde{E}_i, \widetilde{w}_i = W_i \cdot \widetilde{w}_i')$ with edge weights scaled by $W_i$.

    **end if**

**end for**

$\widetilde{G} = \widetilde{G}_1 + \ldots + \widetilde{G}_{\log_{(1+\epsilon)} W}$

---

cost of any hierarchy $\mathcal{T}$ on the weighted graph $G$ is

$$\text{cost}_G(\mathcal{T}) = \sum_{i=1}^{\log_{(1+\epsilon)} W} \text{cost}_{G_i}(\mathcal{T}) \geq \sum_{i=1}^{\log_{(1+\epsilon)} W} (1+\epsilon)^{i-1} \text{cost}_{G_i'}(\mathcal{T}) \overset{\text{Lem 2}}{\geq} \sum_{i=1}^{\log_{(1+\epsilon)} W} \frac{W_i \cdot |E_i|^2}{n}, \quad (7)$$

where the first inequality follows from the fact that the clustering cost function is monotone in edge weights, and every edge in $G_i = (V, E_i, w)$ has weight at least $(1 + \epsilon)^{i-1}$. We now claim that for every weight class $i$, the scaled sparsifier $\widetilde{G}_i$ is a $(O(\epsilon), O(W_i \delta_i))$-sparsifier of the subgraph $G_i = (V, E_i, w)$. To see the lower bound, observe that for any cut $(S, \overline{S})$

$$w_{\widetilde{G}_i}(S) = W_i \cdot w_{\widetilde{G}_i'}(S) \overset{\text{Thm 12}}{\geq} W_i \cdot (1 - \epsilon) w_{G_i'}(S) \geq (1 - \epsilon) w_{G_i}(S), \quad (8)$$

where the final inequality follows from the fact that every edge in $G_i$ has weight at most $W_i$. To see the upper bound, observe that for any cut $(S, \overline{S})$,

$$\begin{aligned} w_{\widetilde{G}_i}(S) = W_i \cdot w_{\widetilde{G}_i'}(S) &\overset{\text{Thm 12}}{\leq} W_i \cdot (1 + \epsilon) w_{G_i'}(S) + O(W_i \delta_i) \cdot \min\{|S|, |\overline{S}|\} \\ &\leq (1 + \epsilon)^2 w_{G_i}(S) + O(W_i \delta_i) \cdot \min\{|S|, |\overline{S}|\} \\ &\leq (1 + 3\epsilon) w_{G_i}(S) + O(W_i \delta_i) \cdot \min\{|S|, |\overline{S}|\}, \end{aligned} \quad (9)$$

where the second inequality follows from the fact that every edge in $G_i$ has weight at least $W_i/(1 + \epsilon)$. Since we have that the edge set $E = E_1 + \ldots + E_{\log_{(1+\epsilon)} W}$, this directly gives us that the scaled sparsifier returned $\widetilde{G} = \widetilde{G}_1 + \ldots + \widetilde{G}_{\log_{(1+\epsilon)} W}$ is a $\left(O(\epsilon), O(\sum_i W_i \delta_i)\right)$-cut sparsifier of $G$, where for any cut $(S, \overline{S})$,

$$(1 - \epsilon) w_G(S) \overset{\text{Eq. 8}}{\leq} w_{\widetilde{G}}(S) \overset{\text{Eq. 9}}{\leq} (1 + 3\epsilon) w_G(S) + O\left(\sum_{i=1}^{\log_{(1+\epsilon)} W} W_i \delta_i\right) \cdot \min\{|S|, |\overline{S}|\}. \quad (10)$$

By choice of each $\delta_i \leq \epsilon |E_i|^2/n^3$, we further have that

$$\sum_{i=1}^{\log_{(1+\epsilon)} W} W_i \delta_i \leq \frac{\epsilon}{n^2} \sum_{i=1}^{\log_{(1+\epsilon)} W} W_i \cdot \frac{|E_i|^2}{n} \overset{\text{Eqn 7}}{\leq} \frac{\epsilon}{n^2} \cdot \text{cost}_G(\mathcal{T}), \qquad \forall \text{ hierarchies } \mathcal{T}.$$

Given this guarantee, the bound on the hierarchical clustering cost claimed in Theorem 14 $(a)$ follows by a straightforward application of Lemma 1.

To complete this proof, the last thing we need to verify is the time and query complexity of Algorithm 3. We shall break down the complexity of this algorithm across the weight classes $i \in [\log_{1+\epsilon} W]$. As described earlier, for any weight class $i$, establishing query access to the subgraph $G_i' = (V, E_i)$ requires at most $\widetilde{O}(n)$ time.

Let $|E_i| = \alpha_i n^{4/3}$ be the number of edges in this subgraph. In the case $\alpha_i \leq 1$, this subgraph is sufficiently sparse and $G_i$ is read entirely which takes $O(\alpha_i n^{4/3})$ time and queries. Otherwise ($\alpha_i > 1$), in which case it is sparsified in $\widetilde{O}(\epsilon^{-3} \max\{\alpha_i^{-2} n^{4/3}, n\})$ time and queries as established in Theorem 12. Therefore, the total complexity of processing a weight class $i$ is $\widetilde{O}(n + f(n, \alpha_i, \epsilon))$, where $f(n, \alpha, \epsilon) = \widetilde{O}(\alpha n^{4/3})$ if $\alpha \leq 1$, and $\widetilde{O}(\epsilon^{-3} \max\{\alpha^{-2} n^{4/3}, n\})$ otherwise. Since there are $O(\epsilon^{-1} \log W)$ weight classes in total, Algorithm 3 runs in time $\widetilde{O}(\epsilon^{-1} n \log W + \sum_i f(n, \alpha_i, \epsilon)) \leq \widetilde{O}((\epsilon^{-1} \log W) \cdot (n + \max_i f(n, \alpha_i, \epsilon)))$.

Lastly, for any given parameter $\tau \in (0, 1/2)$, the sublinear time, $O(\sqrt{\tau^{-1} \log n})$-approximation claim (Theorem 14 (b)) follows by the same $\phi$-approximate hierarchical clustering oracle construction described in the proof of Theorem 11 combined with the fact that our $(\epsilon, \delta)$-cut sparsifier for the weighted graph $G$ now contains $\widetilde{O}((\epsilon^{-1} \log W) \cdot \max_i f(n, \alpha_i, \epsilon))$ edges.

$\square$

### D.2.1 Necessity of Ordering Neighbours by Weight

We conclude this section by showing that the assumption that the adjacency list of each vertex $u$ orders the neighbours of $u$ by weight, is in fact necessary. Otherwise, no non-trivial approximation for hierarchical clustering is possible even when one is allowed to query a constant fraction of edges in the graph. We shall naturally consider only sufficiently dense graphs with $\Omega(n^{4/3})$ edges. While this isn't strictly necessary for our example, our upper (and lower) bounds allow us to simply read the entire graph otherwise, rendering the sparse regime uninteresting. While this is straightforward to see when the upper limit on edge weights $W = \text{poly}(n)$ is large, we can even show this for a relatively small $W = n^{1+\epsilon}$ for any constant $\epsilon > 0$. The example is as follows: consider an input graph $G = (V, E_1 \cup E_2, w)$ with $n$ vertices, and an edge set of size $m$ consisting of the union of two Erdős-Rényi random graphs, where $E_1 \sim \mathcal{G}_{n,p}$ for any $p > n^{-2/3}$ with all edges having weight $1$ and $E_2 \sim \mathcal{G}_{n, 1/3n}$ with all edges having weight $W = n^{1+\epsilon}$ for some constant $\epsilon > 0$. We can assume that edges in both $E_1$ and $E_2$ are given the larger weight.

We shall first establish an upper bound on the cost of the optimal hierarchical clustering in $G$, which we claim is at most $nm + O(nW \log n)$. To prove this, we shall use the fact that with probability at least $1 - 1/n$, (a) the subgraph $G_2 = (V, E_2)$ is a union of connected components, each either a tree or a unicyclic component, and (b) the degree of every vertex in $G_2$ is at most 3. The former is well known in the random graph literature, [23] and the latter follows from Bernstein's concentration inequality. Therefore, hierarchical clustering that first separates the different connected components of $G_2$, following which each connected component is partitioned recursively using a balanced sparsest cut, i.e. the sparsest cut with a constant fraction of the remaining vertices on either side of the cut, will achieve a cost of at most $O(nW \log n)$. The remaining edges in $E_1$, regardless of how they are arranged can cumulatively add no more than $n|E_1|$ to the cost of this hierarchical clustering.

Now consider any (randomized) algorithm that performs at most $2m/9$ neighbour and pair queries in total, and let $\mathcal{T}$ be the hierarchical clustering returned by this algorithm. Consider a balanced cut $(S, \overline{S})$ in this tree, i.e. an internal node $S$ with $\min\{|S|, |\overline{S}|\} \geq n/3$. Since the number of queries made is bounded by $2m/9$, there necessarily are at least $2n^2/9 - 2m/9 \geq n^2/9$ unqueried edge pairs from the cut $(S, \overline{S})$. Furthermore, there are at least $m - 2m/9 = 7m/9$ unqueried edges in $G$. For every unqueried edge, there is at least a constant $(n^2/9)/\binom{n}{2} \geq 2/9$ probability that it realized into an edge slot from the cut $(S, \overline{S})$, and then at least a $1/(3n)$ marginal probability that it came from $E_2$. Therefore in expectation, there are at least $(7m/9) \cdot (2/9) \cdot (1/3n) \geq m/(18n)$ edges from $G_2$, each having weight $W$ that go across the cut $(S, \overline{S})$. Since $|S| \geq n/3$, the contribution of each heavy edge to the cost of $\mathcal{T}$ is at least $n/3 \cdot W$, and therefore, the expected cost of $\mathcal{T}$ is at least $(m/18n) \cdot (n/3) \cdot W = mW/54$ due to these heavy edges alone. Note that this argument holds even if the neighbours of every vertex are ordered randomly.

Now by comparing the cost of the optimal clustering, which is at most $nm + O(nW \log n)$, to the expected cost of the hierarchical clustering produced by an algorithm that makes at most $2m/9$ queries, which is $\Omega(mW)$, it is easy to see that the approximation ratio in expectation is $\Omega(n^\epsilon)$ when $W = n^{1+\epsilon}$ and $m \geq n^{1+\epsilon} \log n$.

## E  Sublinear Communication Algorithms under MPC Model

Finally, we consider the bounded communication setting in the massively parallel computation (MPC) model of [6], where the edge set of the input graph is partitioned across several machines which are inter-connected via a communication network. This model naturally captures certain distributed computing settings [34, 50, 49]. The communication proceeds in synchronous rounds. During each round of communication, any machine can send any information to an arbitrary subset of other machines. However, the total number of bits a machine is allowed to send or receive is limited by the memory of the machine. Between two successive rounds, each machine is allowed to perform an arbitrary computation over their inputs and any other bits received in the previous rounds. At the end, a machine designated as the coordinator is required to output a solution based on its initial input

and the communication it receives. The objective is to study the trade-off between the number of rounds and communication required by each machine, or as alternatively stated, minimize the number of rounds given a fixed communication budget for each machine. Note that the communication budget of each machine is same as the memory given to the machine.

## E.1    A 2-Round $\widetilde{O}(n)$ Communication Algorithm

We first give a 2 round algorithm that requires $\widetilde{O}(n)$ communication per machine. The following is the main result of this section.

**Theorem 15.** *There exists a randomized MPC algorithm that, given a weighted graph $G = (V, E, w)$ over $n$ vertices where edge weights are $O(\text{poly}(n))$, and a $\phi$-approximate hierarchical clustering oracle, can compute with high probability a $(1 + \epsilon)\phi$-approximate hierarchical of $G$ in 2 rounds using $\widetilde{O}(\epsilon^{-2}n)$ communication per machine and access to public randomness.*

In order to prove this theorem we will utilize a result from [1] for constructing $(\epsilon, 0)$-cut sparsifiers using *linear graph sketches*. Given $L : \mathbb{R}^d \to \mathbb{R}^{d'}$ and $x \in \mathbb{R}^d$, we say that $L(x)$ is a sketch of $x$. In order to sketch a graph, we represent each vertex in the graph using a $\binom{n}{2}$-dimensional vector and then compute a sketch for each vertex. Let the vertices in the graph be indexed as $1, \cdots, n$. For each $i \in [n]$, we will define a vector $x^{(i)} \in \{-1, 0, 1\}^{\binom{n}{2}}$ as follows: we first compute a matrix $M$ of size $n \times n$ with

$$M_{ij} = \begin{cases} -1 & (i, j) \in E \text{ and } i < j \\ +1 & (i, j) \in E \text{ and } i > j \\ 0 & \text{otherwise} \end{cases}.$$

The vector $x^i$ is then obtained by flattening $M$ after removing all the diagonal entries. The following theorem summarizes the result of [1] for computing cut-sparsifiers using linear sketches.

**Theorem 16** ([1]). *For any $\epsilon > 0$, there exists a (random) linear function $L : \mathbb{R}^{\binom{n}{2}} \to \mathbb{R}^{O(\epsilon^{-2}\text{poly}\log n)}$ such that, given any graph $G = (V, E, w)$ over $n$ vertices with edge weights that are $O(\text{poly}(n))$, a $(\epsilon, 0)$-cut sparsifier can be constructed with high probability using the sketches $L(x^{(1)}), \cdots, L(x^{(n)})$ of each vertex. Moreover, each of these sketches can by computed using $O(\epsilon^{-2}\text{poly}\log n)$ space given access to fully independent random hash functions.*

Note that an important property of this sketch is that it is linear, which means that (partial) independently computed sketches $L(x^{(i,1)}), \cdots, L(x^{(i,t)})$ for a vertex $i$ can be added together to get a sketch $L(x^{(i)}) = L(x^{(i,1)}) + \cdots + L(x^{(i,t)})$. We will now use this result for computing a cut-sparsifier using 2 rounds of MPC computation. We will use the same construction of linear sketches for each vertex as in this result.

---

**2-round MPC Algorithm:**

1. **Input:** Parameter $\epsilon \in (0, 1/2]$, graph $G = (V, E, w)$ such that edges are partitioned over $k$ machines.

2. Let each machine be responsible for constructing the sketch for $n/k$ (arbitrarily chosen) vertices.

3. Divide the weights into $O(\log n)$ weight classes similar to [1].

4. Each machine *locally* constructs a (random) linear sketch of size $O(\epsilon^{-2}\text{poly}\log n)$ for each vertex and weight class. Each machine computes the sketches according to the same function $L$ using Theorem 16 by computing the same random hash functions through public randomness.

5. **Round 1:** Each machine sends its local linear sketches of a vertex to the machine that is responsible for this vertex.

6. For each weight class, each machine constructs the linear sketches for each of its responsible vertices by adding the corresponding partial linear sketches.

7. **Round 2:** Each machine sends its $n/k$ linear sketches to the coordinator.

8. The coordinator constructs a $(\epsilon, 0)$-cut sparsifier using the algorithm of [1] and outputs a $\psi$-approximate hierarchical clustering over the cut sparsifier.

---

The above pseudo-code outlines the 2-round algorithm for hierarchical clustering in the MPC model. Here, we arbitrarily partition vertices into $k$ sets of size $(n/k)$ each, and designate each machine to be responsible for constructing the sketch for $n/k$ (arbitrarily chosen) vertices. Since, the edges are partitioned over $k$ machines, a given machine might not have all the edges incident on a vertex. Hence, in the first round each machine will

locally construct a linear sketch for each vertex based on its edges. Note that each machine will construct linear sketches using the same function $L$ by sampling the same random hash functions. Then each machine sends their local linear sketches for each vertex to the responsible machines. Each machine can send $\widetilde{O}(\epsilon^{-2}n)$ bits in total as the sketch of each vertex is of size $O(\epsilon^{-2}\text{poly}\log n)$. Moreover, each machine can receive $\widetilde{O}(\epsilon^{-2}n)$ bits as it can receive at most $\widetilde{O}(\epsilon^{-2}n/k)$ bits from $k-1$ other machines. Each machine then constructs the linear sketches for its vertices by adding the corresponding sketches. Note that these sketches are valid due to linearity and the fact that the random hash functions are shared across all machines. Finally, each machine will send its sketches to the coordinator. The coordinator will then compute a cut sparsifier using these sketches and run a $\phi$-approximate hierarchical clustering algorithm over the sparsified graph. Using Theorem 16 and Lemma 1, we can easily argue that the coordinator's hierarchical clustering will be $(1 + O(\epsilon)) \cdot \phi$-approximate with high probability.

## E.2   A 1-Round $\widetilde{O}(n^{4/3})$ Communication Algorithm

We next consider the possibility of computing a good hierarchical clustering in just a single round in the MPC model. However, as we will show in Section G, computing in one round requires $\Omega(n^{4/3})$ communication (and hence, machine memory) requirement, even for unweighted graphs. In this setting, give a 1-round $\widetilde{O}(n^{4/3})$ communication MPC algorithm for hierarchical clustering of unweighted graphs assuming knowledge of the number of edges in the input graph, and the number of machines being bounded by $m/n^{4/3}$.

**Theorem 17.** *There exists a randomized MPC algorithm that, given an unweighted graph $G = (V, E)$ over $n$ vertices and $m$ edges, a parameter $0 < \epsilon \le 1/2$, and a $\phi$-approximate hierarchical clustering oracle, can compute with high probability a $(1+\epsilon)\phi$-approximate hierarchical clustering of $G$ in 1 round using $\widetilde{O}(\epsilon^{-2}n^{4/3})$ communication per machine and $k \le m/n^{4/3}$ machines with access to public randomness.*

The following pseudo-code outlines the 1-round algorithm.

---

**1-round MPC Algorithm:**

1. **Input:** Parameter $\epsilon \in (0, 1/2]$, graph $G = (V, E)$ with $m$ edges such that edges are partitioned over $k \le m/n^{4/3}$ machines each with memory $\widetilde{\Omega}(\epsilon^{-2}n^{4/3})$.

2. **If** $m = \beta n^{4/3}$ for $\beta \ge n^{1/3}$ then

   (a) Each machine samples its each of its local edges independently with probability $p = C(\epsilon^2\beta)^{-1}\log n$ for some sufficiently large constant $C$ and sends all the sampled edges with weight $1/p$ to the coordinator.

   (b) Let $\delta := m^2/n^3 = \beta^2/n^{1/3}$, and let $H = (V, E_h, w_h)$ be the weighted graph induced by the sampled edges received from all machines. The coordinator constructs a constant degree expander $G_x = (V, E_x, w_x)$ with all edges having weight $\epsilon\delta$, and embeds this weighted expander in $H$. Let $\widetilde{G} = (V, E_h \cup E_x, w_h + w_x)$ be the resultant composite graph.

   (c) The coordinator runs a $\phi$-approximate hierarchical clustering on $\widetilde{G}$ and returns the answer.

3. **Else if** $m = \alpha n$ for $\alpha < n^{2/3}$ then

   (a) Each machine computes a (random) linear sketch of size $O(\epsilon^{-2}\text{poly}\log n)$ for all vertices using the local edges. Each machine computes the sketches according to the same function $L$ using Theorem 16 by computing the same random hash functions through public randomness.

   (b) Each machine sends its local linear sketches to the coordinator.

   (c) The coordinator adds the partial linear sketches corresponding to each vertex to get one linear sketch per vertex. It then runs the algorithm of [1] for computing a $(\epsilon, 0)$-cut sparsifier. Finally, it runs a $\phi$-approximate hierarchical clustering over the cut sparsifier and returns the answer.

---

The execution of the above algorithm is divided into two cases based on the number of edges in the graph. We analyze these two cases separately below.

**Analysis for Case 1:** We first observe that in this case, since the total number of edges in the graph $G$ that is distributed across all machines is $\beta n^{4/3}$, and each machine samples its local edges with probability $p = C(\epsilon^2\beta)^{-1}\log n$, the total number of edges sampled across all machine, and therefore, the total communication to the coordinator is $\widetilde{O}(\epsilon^{-2}n^{4/3})$.

We shall now bound the cost of the hierarchical clustering returned by our scheme. We begin by lower bounding the cost of any hierarchical clustering $\mathcal{T}$ of $G$, which by Lemma 2 is at least $\beta^2 n^{5/3}$, which implies that

$\delta := \beta^2/n^{1/3} \leq \mathsf{cost}_G(\mathcal{T})/n^2$ for any hierarchy $\mathcal{T}$. We shall argue that in the weighted sampled graph $H = (V, E_h, w_h)$ received by the coordinator, with probability at least $1 - 1/\mathrm{poly}(n)$, the weight of any cut $(S, \overline{S})$ is such that

$$(1 - \epsilon)w_G(S) - \epsilon\delta \min\{|S|, |\overline{S}|\} \leq w_H(S) \leq (1 + \epsilon)w_G(S) + \epsilon\delta \min\{|S|, |\overline{S}|\}. \tag{11}$$

Assuming this bound, it is relatively straightforward to prove that running the $\phi$-approximate hierarchical clustering algorithm on the composite graph $\widetilde{G} = (V, E_h \cup E_x, w_h + w_x)$ would produce a $(1 + O(\epsilon))\phi$-approximate clustering. This follows by observing that the composite graph $\widetilde{G}$ is an $(\epsilon, \Theta(\epsilon\delta))$-sparsifier of the input graph $G$, as the weight of any cut $(S, \overline{S})$ in $\widetilde{G}$ is

$$(1 - \epsilon)w_G(S) \leq w_{\widetilde{G}}(S) = w_H(S) + w_{G_x}(S) \leq (1 + \epsilon)w_G(S) + \Theta(\epsilon\delta) \min\{|S|, |\overline{S}|\},$$

where both inequalities follow by substituting the bounds in Eq. (11), and observing that for any cut $(S, \overline{S})$, the weight $w_{G_x}(S) = \epsilon\delta \cdot \Theta(\min\{|S|, |\overline{S}|\})$ due to expansion and choice of edge weights in $G_x$. This guarantee together with Lemma 1 proves our claimed bound on the cost of the hierarchical clustering computed by our algorithm.

We shall now prove the bounds claimed in Eq. (11). Consider any cut $(S, \overline{S})$, and let us assume that $|S| = k \leq n/2$. Let $E_S$ be the edges that cross the cut $(S, \overline{S})$ in graph $G$. For every edge $e \in E_S$, we define a random variable $X_e$ that is Bernoulli with parameter $p = C(\epsilon^2\beta)^{-1}\log n$, taking value 1 if edge $e$ is sampled. Therefore, the weight of this cut in $H$ is the random variable $w_H(S) = p^{-1}\sum_{e \in E_S} X_e$. We shall now bound the probability of the bad event where the value of this cut $w_H(S) > (1 + \epsilon)w_G(S) + \epsilon\delta k$ as

$$\Pr(w_H(S) > (1 + \epsilon)w_G(S) + \epsilon\delta k) = \Pr\left(\sum_{e \in E_S} X_e > (1 + \epsilon)\mathbb{E}[w_G(S)] + \epsilon\delta pk\right)$$

$$= \Pr\left(\sum_{e \in E_S}(X_e - p) > \epsilon\mathbb{E}[w_G(S)] + \epsilon\delta pk\right)$$

$$= \Pr\left(\sum_{e \in E_S} Y_e > \epsilon\mathbb{E}[w_G(S)] + \epsilon\delta pk\right),$$

where for any edge $e \in E_S$, random variable $Y_e = X_e - p$ is such that $\mathbb{E}[Y_e] = 0$, $|Y_e| \leq 1 - p$, and $\mathbb{E}[Y_e^2] = p(1 - p)$. Therefore, by Bernstein's inequality,

$$\Pr\left(\sum_{e \in E_S} Y_e > \epsilon\mathbb{E}(w_G(S)) + \epsilon\delta pk\right) \leq \exp\left(-\frac{3}{2(1 - p)} \cdot \frac{\epsilon^2\left(\mathbb{E}[w_G(S)] + \delta pk\right)^2}{(3 + \epsilon)\mathbb{E}[w_G(S)] + \epsilon\delta pk}\right). \tag{12}$$

Now there are two cases, either the cut $(S, \overline{S})$ is such that (a) $\mathbb{E}[w_G(S)] \geq \delta pk$ or (b) $\mathbb{E}[w_G(S)] < \delta pk$. In the first case, we have the upper bound in Eq. (12) is at most

$$\Pr\left(\sum_{e \in E_S} Y_e > \epsilon\mathbb{E}(w_G(S)) + \epsilon\delta pk\right) \leq \exp\left(-\frac{3\epsilon^2\mathbb{E}[w_G(S)]}{2(1 - p)(4 + \epsilon)}\right)$$

$$\leq \exp\left(-\frac{3\epsilon^2\delta pk}{2(1 - p)(4 + \epsilon)}\right)$$

$$\overset{(a)}{\leq} \exp\left(-\frac{3C\beta k\log n}{2(1 - p)(4 + \epsilon)n^{1/3}}\right) \overset{(b)}{\leq} \exp\left(-C'k\log n\right),$$

where $C'$ is a constant, with (a) following by choice of $p$ and $\delta$, and (b) following by observing that $\beta \geq n^{1/3}$. In the second case, we have the upper bound in Eq. (12) is at most

$$\Pr\left(\sum_{e \in E_S} Y_e > \epsilon\mathbb{E}(w_G(S)) + \epsilon\delta pk\right) \leq \exp\left(-\frac{3\epsilon^2\delta pk}{2(1 - p)(3 + 2\epsilon)}\right) \overset{(a)}{\leq} \exp\left(-C'k\log n\right),$$

where (a) follows by the same calculation as the previous case. Therefore, by taking a union bound over all cuts $(S, \overline{S})$ with $|S| = k \leq n/2$, we have that

$$\Pr(\exists\, S : |S| = k, \text{ and } w_H(S) > (1+\epsilon)w_G(S)+\epsilon\delta k) \leq \binom{n}{k}\exp\left(-C'k\log n\right) \leq \exp\left(-(C' - 1)k\log n\right),$$

and therefore, a union bound over all choices of $1 \leq k \leq n/2$ gives us that for a sufficiently large constant $C$, with probability at least $1 - 1/\text{poly}(n)$, we have for all cuts $(S, \overline{S})$

$$w_H(S) \leq (1 + \epsilon)w_G(S) + \epsilon\delta\min\{|S|, |\overline{S}|\}.$$

Following an identical analysis for $Y_e = p - X_e$ gives us that with probability at least $1 - 1/\text{poly}(n)$, we have for all cuts $(S, \overline{S})$

$$w_H(S) \geq (1 - \epsilon)w_G(S) - \epsilon\delta\min\{|S|, |\overline{S}|\},$$

proving the bound claimed in Eq. (11), completing the analysis for this case where $\beta \geq n^{1/3}$.

**Analysis for Case 2:** In this case the number of edges in the graph is at most $n^{5/3}$. Since the memory of each machine is $\widetilde{\Omega}(\epsilon^{-2}n^{4/3})$, the number of machines can be at most $n^{1/3}$. Each machine constructs linear sketches over its input and sends these to the coordinator similar to the 2-round algorithm in Section E.1. Note that the total communication is at most $n^{1/3} \times \widetilde{O}(\epsilon^{-2}n) = \widetilde{O}(\epsilon^{-2}n^{4/3})$ as each machine can only send $\widetilde{O}(\epsilon^{-2}n)$ bits to the coordinator. The coordinator then adds all the sketches corresponding to each vertex and computes a cut sparsifier using the algorithm of [1]. Using Theorem 16, we can again argue that these linear sketches are such that one can recover a $(\epsilon, 0)$-cut sparsifier with high probability. The claimed bound on the cost of the hierarchical clustering recovered then follows by a direct application of Lemma 1.

# F Tight Query Lower Bounds for $\widetilde{O}(1)$-approximation

We note that, for unweighted graphs, our sublinear time algorithm requires only 2 rounds of adaptive queries, where the first round only needs to query vertex degrees. Thus if one assumes prior knowledge of vertex degrees, our algorithm is in fact *non-adaptive*. For weighted graphs, our algorithm requires at most $O(\log n)$ rounds of adaptive queries due to the binary searches. In any case, our algorithm makes at most $\widetilde{O}(n^{4/3})$ queries, where the worst-case input is an unweighted graph of about $\approx n^{4/3}$ edges.

We now show that, in a sharp contrast, even with unlimited adaptivity, our algorithm's query complexity is essentially the best possible for any randomized algorithm that computes a polylog$(n)$-approximate hierarchical clustering tree with high probability. In particular, we establish below tight query lower bounds when the input is an unweighted graph with $m = \Theta(n^\zeta)$ edges for any constant $\zeta \in [0, 2]$. By plugging in $\zeta = 4/3$ in **Case 4**, we get a matching lower bound for the worst-case input graph.

- **Case 1:** $\zeta = 2$. Any binary hierarchical clustering tree has cost $O(n^3)$ (Fact 2), and by Lemma 2, the optimal cost is at least $\Omega(n^3)$. Thus trivially 0 queries are sufficient for $O(1)$-approximation.

- **Case 2:** $\zeta \in [0, 1]$. It is not hard to show an $\Omega(n)$ query lower bound even for $o(n)$-approximation. Specifically, consider using a random matching of size $\Theta(n^\zeta)$ as a hard distribution, whose optimal hierarchical clustering cost is $\Theta(n^\zeta)$. However, any $o(n)$-query algorithm can only discover an $o(1)$-fraction of the matching edges, and with an $\Omega(1)$ fraction of the matching edges having high entropy, any balanced cut of the graph has nontrivial probability of cutting $\Omega(n^\zeta)$ matching edges, incurring a cost of $\Omega(n^{1+\zeta})$.

  On the algorithmic side, one can simply probe all edges with $O(n)$ queries and then run any hierarchical clustering algorithm on the entire graph. Thus the query complexity for $\widetilde{O}(1)$-approximation is settled at $\Theta(n)$.

- **Case 3:** $\zeta \in [3/2, 2)$. One can show an $\Omega(n)$ query lower bound for $\widetilde{O}(1)$-approximation, by considering an input graph obtained by randomly permuting the vertices of a union of vertex-disjoint cliques. We include a proof of this lower bound in Section F.1.

  On the algorithmic side, our sublinear time algorithm obtains an $O(\sqrt{\log n})$-approximation using $\widetilde{O}(n)$ queries in this case, which is nearly optimal.

- **Case 4:** $\zeta \in (1, 3/2)$. Let $\gamma := \zeta - 1 \in (0, 1/2)$. Our sublinear time algorithm obtains an $O(\sqrt{\log n})$-approximation using $\widetilde{O}(n^{\min\{1+\gamma, 2-2\gamma\}})$ queries. We show in Section F.2 that this is nearly optimal even for $\widetilde{O}(1)$-approximation.

## F.1 Lower bound for $m$ between $n^{3/2}$ and $n^2$

**Theorem 18** (Lower bound for $m$ between $n^{3/2}$ and $n^2$). *Let $\gamma \in [1/2, 1)$ be an arbitrary constant. Let $\mathcal{A}$ be a randomized algorithm that, on an input unweighted graph $G = (V, E)$ with $|V| = n$ and $|E| = \Theta(n^{1+\gamma})$, outputs a* polylog$(n)$-*approximate hierarchical clustering tree with probability $\Omega(1)$. Then $\mathcal{A}$ necessarily uses $\Omega(n)$ queries.*

We will show that there exists a distribution $\mathcal{D}$ over graphs with $n$ vertices and $\Theta(n^{1+\gamma})$ edges, on which no deterministic algorithm using $o(n)$ queries can output a polylog$(n)$-approximate hierarchical clustering tree with probability $\geq .99$. This coupled with Yao's minimax principle [47] will prove Theorem 18.

We define $\mathcal{D}$ such that a graph $G \sim \mathcal{D}$ is generated by first taking a union of $n^{1-\gamma}$ vertex-disjoint cliques of size $n^\gamma$, and then permuting the $n$ vertices uniformly at random. More formally, we first pick a uniformly random permutation $\pi : [n] \to [n]$, and then let $G$ be a union of vertex-disjoint cliques $C_1, \ldots, C_{n^{1-\gamma}}$ each of size $n^\gamma$ such that $C_i$ is supported on vertices

$$S_i := \{\pi((i-1)n^\gamma + 1), \ldots, \pi(in^\gamma)\}.$$

By Fact 2, we know that the optimal hierarchical clustering cost of each clique is $O(n^{3\gamma})$. Therefore, summing this cost over all cliques, we have:

**Proposition 1.** *The optimal hierarchical clustering tree of $G$ has cost $O(n^{1+2\gamma})$.*

We now describe a process that interacts with any given deterministic algorithm $\mathcal{A}$ using $o(n)$ queries while generating a uniformly random permutation $\pi : [n] \to [n]$ along with its inverse function $\pi^{-1} : [n] \to [n]$. Specifically, we will generate $\pi, \pi^{-1}$ by realizing them entry by entry adaptively based on the queries made be the algorithm. Thus, when realizing an entry of $\pi$ or $\pi^{-1}$, we will always do so by conditioning on their already realized entries. Also note that since the degree of each vertex is the same (namely $n^\gamma - 1$), we will give the degree information to $\mathcal{A}$ for free at the start. The process then proceeds by the following two principles:

**Principle 1:** Upon a pair query between $i, j$, realize $\pi^{-1}(i), \pi^{-1}(j)$ and then answer the query accordingly.

**Principle 2:** Upon a neighbor query about the $\ell^{\text{th}}$ neighbor of $i$, first realize $\pi^{-1}(i)$. Let $k$ be such that the $\ell^{\text{th}}$ neighbor of $i$ is $\pi(k)$. Then realize $\pi(k)$ and answer the query accordingly.

Clearly, each query triggers the realization of $O(1)$ entries of $\pi$ and $\pi^{-1}$. Thus, after $\mathcal{A}$ terminates, the number of realized entries of $\pi$ and $\pi^{-1}$ is at most $o(n)$. Let $U \subset [n]$ with $|U| \geq (1 - o(1))n$ be the set of indices whose $\pi$ values are not realized, and similarly let $W \subset [n]$ with $|W| = |U| \geq (1 - o(1))n$ be the set of indices whose $\pi^{-1}$ values are not realized.

Let $\mathcal{T}$ be the hierarchical clustering tree output by $\mathcal{A}$, which we suppose for the sake of contradiction is polylog$(n)$-approximate. We first make $\mathcal{T}$ a full binary tree such that the bi-partition of each internal node is $[1/3, 2/3]$-balanced, during which we increase the cost of the tree by at most an $O(1)$ factor. We next consider the bi-partition of the root, which is a cut $(S, \bar{S})$ with $|S| \in [n/3, 2n/3]$.

Let $S' := S \cap W$ and $T' := \bar{S} \cap W$, and thus $(S', T')$ is a bi-partition of $W$. Since $|W| \geq (1 - o(1))n$, we have $|S'| \in [|W|/6, 5|W|/6]$. Since also $|U| \geq (1 - o(1))n$, we have that for at least $\Omega(1)$ fraction of the cliques $C_i$'s (which are supported on $S_i$'s), we have

$$|\{(i-1)n^\gamma + 1, \ldots, in^\gamma\} \cap U| \geq n^\gamma/2.$$

For each such clique $C_i$, the number of edges within $C_i$ that are across $(S', T')$ is $\Omega(n^{2\gamma})$ with high probability. Therefore, the size of the cut $(S, \bar{S})$ is at least $\Omega(n^{1+\gamma})$ with high probability. This means that the cost of $\mathcal{T}$ is at least $\Omega(n^{2+\gamma})$, which together with $\gamma < 1$ contradicts $\mathcal{T}$ being polylog$(n)$-approximate.

## F.2 Lower bound for $m$ between $n$ and $n^{3/2}$

**Theorem 19** (Lower bound for $m$ between $n$ and $n^{3/2}$). *Let $\gamma \in (0, 1/2)$ be an arbitrary constant. Let $\mathcal{A}$ be a randomized algorithm that, on an input unweighted graph $G = (V, E)$ with $|V| = n$ and $|E| = \Theta(n^{1+\gamma})$, outputs with $\Omega(1)$ probability a polylog$(n)$-approximate hierarchical clustering tree. Then $\mathcal{A}$ necessarily uses at least $n^{\min\{1+\gamma, 2-2\gamma\} - o(1)}$ queries.*

By Yao's minimax principle [47], to prove Theorem 19, it suffices to exhibit a hard input distribution on which every *deterministic* algorithm using a small number of queries fails with nontrivial probability. Specifically, we will show that there exists a distribution $\mathcal{D}$ over graphs with $n$ vertices and $\Theta(n^{1+\gamma})$ edges such that, on an input graph drawn from $\mathcal{D}$, any deterministic algorithm using $n^{\min\{1+\gamma, 2-2\gamma\} - \delta}$ queries for any constant $\delta > 0$ can only output a polylog$(n)$-approximate hierarchical clustering tree with $o(1)$ probability.

**The hard distribution.** We start by defining the hard distribution $\mathcal{D}$ over graphs with $n$ vertices and $\Theta(n^{1+\gamma})$ edges. Roughly speaking, we will generate an input graph $G$ by first taking the union of a certain number of cliques $C_1, \ldots, C_k$ of equal size $n/k$, and then adding some artificially structured edges between them. We will then show that even the edges between the cliques are relatively tiny compared to those within, it is necessary to discover them in order to output a good hierarchical clustering solution.

More specifically, we will decide what edges to add between cliques based on the structure of a randomly generated "meta graph" $H$ on $k$ supernodes, with supernode $i$ in $H$ representing the clique $C_i$. We generate the meta graph $H$ by picking a uniformly random perfect matching between the $k$ supernodes (assuming for simplicity $k$ is even). Then for each matched pair of supernodes $i, j$ in the meta graph $H$, we will add between $C_i$ and $C_j$ a random bipartite matching of certain size (note that this matching is in the actual graph $G$ rather than the meta graph $H$). Moreover, when adding the latter matching edges in $G$, we will also delete some edges inside $C_i, C_j$ to ensure that every vertex has the exact same degree, so that an algorithm cannot tell which vertices participate in the meta graph's perfect matching by only looking at the vertex degrees. We will then show:

1. Any deterministic algorithm using $n^{\min\{1+\gamma, 2-2\gamma\}-\delta}$ queries for any $\delta > 0$ can only discover an $o(1)$ fraction of the matching edges in the meta graph $H$.

2. If $\Omega(1)$ fraction of the matching edges have high entropy, an algorithm cannot output a $\mathrm{polylog}(n)$-approximate hierarchical clustering tree with $\Omega(1)$ probability.

We now formally describe how we generate a graph $G$ from $\mathcal{D}$. Let the vertices of $G$ be numbered 1 through $n$. We divide the vertices into $n^{1-\gamma}$ groups $S_1, \ldots, S_{n^{1-\gamma}}$ each of size $n^\gamma$, where

$$S_i := \{(i-1)n^\gamma + 1, \ldots, in^\gamma\} .$$

We then generate the edges of $G$ by the process in Figure 2.

---

1. Generate a meta graph $H$ on supernodes numbered $1, \ldots, n^{1-\gamma}$ by picking a uniformly random perfect matching (of size $n^{1-\gamma}/2$) between them.
2. Initially, add a clique $C_i$ of size $n^\gamma$ to each vertex group $S_i$, and insert the clique edges into the adjacency list of $G$ in an arbitrary order.
3. Let $t \leftarrow n^{\max\{0, 3\gamma-1\}+\frac{1}{\sqrt{\log n}}}$. In what follows, we will add a matching of size $2t$ between each matched clique pair.
4. For each matched pair of supernodes $i, j$ in the meta graph $H$:
   (a) Add a uniformly random *bipartite* matching $M_{i,j}$ of size $2t$ between $S_i$ and $S_j$, and let $T_{i,j}$ denote the vertices matched by $M_{i,j}$ (thus $|T_{i,j} \cap S_i| = |T_{i,j} \cap S_j| = 2t$).
   (b) Inside $S_i$ (resp. $S_j$), pick a uniformly random perfect matching of size $t$ between vertices $T_{i,j} \cap S_i$ (resp. $T_{i,j} \cap S_j$), and delete its edges from clique $C_i$ (resp. $C_j$).
   (c) Modify the adjacency list of the vertices in $G$ by replacing the edges deleted at Step 4b with the edges added at Step 4a. This modification is valid because the degree of each vertex is preserved.

---

Figure 2: Generation of $G \sim \mathcal{D}$.

**Proposition 2.** *All vertices in $G$ have degree exactly $n^\gamma - 1$.*

**Proposition 3.** *The optimal hierarchical clustering tree of $G$ has cost $O(n^{1+2\gamma})$.*

*Proof.* We will construct a hierarchical clustering tree as follows. At the first level, we divide the entire vertex set into $n^{1-\gamma}/2$ clusters where each cluster is a connected component. This step incurs zero cost. We then construct a binary hierarchical clustering tree of each cluster arbitrarily. Since each cluster has $2n^\gamma$ vertices, the hierarchical clustering tree we construct for it has cost bounded by $O(n^{3\gamma})$ (Fact 2). Summing this upper bound over all $n^{1-\gamma}/2$ clusters finishes the proof. $\qquad\square$

**Analysis of deterministic algorithms on $\mathcal{D}$.** Let $\mathcal{A}$ be a deterministic algorithm that makes $n^{\min\{1+\gamma, 2-2\gamma\}-\delta}$ queries for some constant $\delta > 0$. Since all vertices have the same degree $n^\gamma - 1$ in $G$, we will give the degree information to $\mathcal{A}$ for free at the start. We shall then describe a process that interacts with the algorithm $\mathcal{A}$ while generating a $G \sim \mathcal{D}$. To that end, we first define the notion of revealed vertex groups.

**Definition 2** (Revealed vertex groups). *At any given point of the algorithm, we say a vertex group $S_i$ is revealed by $\mathcal{A}$ if at least one of the following is true:*

**Condition 1:** *At least $\frac{n^{2\gamma}}{10000t}$ pair queries involving vertices in $S_i$ are made by $\mathcal{A}$.*

**Condition 2:** *At least $\frac{n^{2\gamma}}{10000t}$ neighbor queries on vertices in $S_i$ are made by $\mathcal{A}$.*

**Condition 3:** *A pair query by $\mathcal{A}$ finds a pair $u, v \in S_i$ not connected by an edge.*

**Condition 4:** *A pair query or a neighbor query by $\mathcal{A}$ finds a pair $u \in S_i, w \notin S_i$ connected by an edge.*

We now describe a process that answers queries made by $\mathcal{A}$ while *adaptively* realizing the edge slots and the adjacency list of $G$, as well as the perfect matching in the meta graph $H$. Whenever realizing a part, we will always do so following the distribution $\mathcal{D}$ *conditioned on* the already realized parts. This means that if a part is already realized or determined by other realized parts, realizing it again will not change it. The process proceeds according to the following three principles:

**Principle 1:** Upon a pair query, realize the corresponding edge slot and answer accordingly.

**Principle 2:** Upon a neighbor query, realize the corresponding entry of the adjacency list and answer accordingly.

**Principle 3:** As soon as a group $S_i$ becomes revealed after a query, due to either large query count or what we have answered by **Principle 1** and **Principle 2**, right away do:

- Realize the supernode $j$ that is matched to $i$ in the meta graph $H$.
- Realize *all* edge slots incident on (and hence also all neighbors of) vertices in $S_i, S_j$.

At any given point of this process, we say a vertex group $S_i$ is *realized* if all edge slots incident on $S_i$ are realized. That is, the realized vertex groups are *exactly* those revealed by $\mathcal{A}$ and the ones matched to them. This in particular implies that a perfect matching has been realized between the realized vertex groups in the meta graph $H$, while none of the unrealized vertex group is matched. As a result, one can show that the queries made so far that involve unrealized vertex groups must have *deterministic* answers:

**Proposition 4.** *At any point of the algorithm $\mathcal{A}$, for the queries already made, we have:*

- *Every pair query between an unrealized vertex group and a realized one discovered no edge.*

- *Every pair query between two unrealized vertex groups discovered no edge.*

- *Every pair query within a same unrealized vertex group discovered an edge.*

- *Every neighbor query on a vertex in an unrealized vertex group found a neighbor within the same group.*

In what follows, we will consider the conditional distribution of $\mathcal{D}$ on all edge slots incident on realized vertex groups, which we denote by $\mathcal{D}_{\mathrm{rz}}$. Note that $G' \sim \mathcal{D}_{\mathrm{rz}}$ is *not* necessarily consistent with the answers we gave to the queries that involve unrealized vertex groups, though these answers are themselves deterministic by Proposition 4. By definition, a graph $G' \sim \mathcal{D}_{\mathrm{rz}}$ can be generated by the process in Figure 3.

---

1. Add the edges incident on the realized vertex groups to $G'$.
2. Add the perfect matching between the realized vertex groups to the meta graph $H$.
3. Add a clique $C_i$ of size $n^\gamma$ to each unrealized vertex group $S_i$.
4. For each unrealized vertex group $S_i$:
   (a) If supernode $i$ is not matched in the meta graph $H$, then match $i$ to another uniformly random unmatched $j$, and change the edges within $S_i \cup S_j$ using Steps 4a-4c in Figure 2.

---

Figure 3: Generation of $G' \sim \mathcal{D}_{\mathrm{rz}}$.

**Proposition 5.** *Consider generating $G' \sim \mathcal{D}_{\mathrm{rz}}$ conditioned on that an unrealized $S_i$ is matched to another unrealized $S_j$ in the meta graph $H$. Then $G'[S_i \cup S_j]$ is consistent with previous answers with probability at least .998.*

*Proof.* First note that, when changing the edges within $S_i \cup S_j$ at Step 4a in Figure 3, the edges we delete from $C_i$ (resp. $C_j$) distribute as a uniformly random matching of size $t$ in $C_i$ (resp. $C_j$), and the edges we add between $S_i, S_j$ distribute as a uniformly random bipartite matching of size $2t$ between $S_i, S_j$, though these distributions are correlated.

Then note that $G'[S_i \cup S_j]$ is *not* consistent with previous answers only if (i) the slot of an edge we delete within $S_i$ or $S_j$ was queried by $\mathcal{A}$, or (ii) an edge we add between $S_i, S_j$ was queried by $\mathcal{A}$. Since $S_i, S_j$ are

both unrevealed, they do *not* satisfy **Condition 1** or **Condition 2**. As a result, we can bound the probability of $G'[S_i \cup S_j]$ being inconsistent with previous answers via a union bound by

$$2 \cdot \frac{2n^{2\gamma}}{10000t} \cdot \frac{t}{\binom{n^{\gamma}}{2}} + \frac{2n^{2\gamma}}{10000t} \cdot \frac{2t}{n^{2\gamma}} \leq .002,$$

which proves the proposition. □

We show that the number of realized vertex groups can be at most a $o(1)$ fraction of the total.

**Proposition 6.** *Upon termination of the algorithm $\mathcal{A}$, the total number of realized vertex groups $S_i$'s is bounded by $o(n^{1-\gamma})$ with probability at least $1 - 1/n^4$.*

*Proof.* The number of vertex groups that satisfy **Condition 1** or **Condition 2** can be at most

$$
\begin{aligned}
\frac{2 \cdot \#\text{queries}}{\frac{n^{2\gamma}}{10000t}} &= \frac{2n^{\min\{1+\gamma, 2-2\gamma\}-\delta}}{\frac{n^{2\gamma}}{10000t}} \\
&= \frac{20000n^{\max\{0, 3\gamma-1\}+\frac{1}{\sqrt{\log n}}} n^{\min\{1+\gamma, 2-2\gamma\}-\delta}}{n^{2\gamma}} \qquad \text{(plugging in the value of } t\text{)} \\
&= 20000n^{1-\gamma+\frac{1}{\sqrt{\log n}}-\delta} \\
&\leq n^{1-\gamma-\Omega(1)} \leq o(n^{1-\gamma}).
\end{aligned}
$$

This means that the total number of realized vertex groups that satisfy **Condition 1** or **Condition 2** and those matched to them is at most $o(n^{1-\gamma})$.

We then bound the number of realized vertex groups that do not satisfy **Condition 1** or **Condition 2** and are not matched to those who satisfy **Condition 1** or **Condition 2**. Each such vertex group must be (matched to) a revealed one that satisfies **Condition 3** or **Condition 4**. We thus consider the probability that a query makes an unrealized vertex group satisfy **Condition 3** or **Condition 4**.

- **Pair query:** If a pair query involves a vertex in an already realized vertex group, then its answer is already determined and it does not reveal any unrealized groups.

  Otherwise, if a pair query only involves unrealized vertex groups, we show that the probability it reveals any unrealized groups is at most $\frac{8t}{n^{2\gamma}}$. First consider the case that the query is within a single unrealized group $S_i$. For a $G' \sim \mathcal{D}_{\text{rz}}$, the probability that this query discovers a non-edge is at most $\frac{t}{\binom{n^{\gamma}}{2}}$. By Proposition 5, conditioned on $S_i$ being matched to another $S_j$ in the meta graph $H$, the probability that $G'[S_i \cup S_j]$ is consistent with previous answers is $\geq .99$. Therefore, this query discovers a non-edge with probability $\leq \frac{2t}{\binom{n^{\gamma}}{2}}$.

  Then consider the case that the query is between two unrealized groups $S_i, S_j$. If $S_i$ is not matched to $S_j$ in the meta graph $H$, then the pair query does not discover an edge, since there is no edge between $S_i, S_j$. Otherwise, for a $G' \sim \mathcal{D}_{\text{rz}}$, conditioned on $S_i$ being matched to $S_j$, the pair query discovers an edge with probability $\frac{2t}{n^{2\gamma}}$. By Proposition 5, the probability that $G'[S_i \cup S_j]$ is consistent with previous answers with probability $\geq .99$. Therefore, this query discovers an edge with probability $\leq \frac{4t}{n^{2\gamma}}$.

- **Neighbor query:** Consider a neighbor query on a vertex $u$ in an unrealized vertex group $S_i$. For a $G' \sim \mathcal{D}_{\text{rz}}$, the query finds an edge going out of $S_i$ with probability $\frac{t}{\binom{n^{\gamma}}{2}}$. By Proposition 5, conditioned on $S_i$ being matched to another $S_j$ in the meta graph $H$, the probability that $G'[S_i \cup S_j]$ is consistent with previous answers with probability $\geq .99$. Therefore, this query discovers an outgoing edge with probability $\leq \frac{2t}{\binom{n^{\gamma}}{2}}$.

Combining the above, a query makes an unrealized vertex group satisfy **Condition 3** or **Condition 4** with probability at most $\frac{8t}{n^{2\gamma}}$. Also, by doing so, a query can increase the number of realized vertex groups by at most 4. As a result, the expected increase in the number of realized groups that do not satisfy **Condition 1**

or **Condition 2** over all queries made by $\mathcal{A}$ is at most

$$4 \cdot \#\text{queries} \cdot \frac{8t}{n^{2\gamma}} = \frac{32n^{\min\{1+\gamma, 2-2\gamma\}-\delta} \cdot t}{n^{2\gamma}}$$

$$= \frac{32n^{\min\{1+\gamma, 2-2\gamma\}-\delta}n^{\max\{0, 3\gamma-1\}+\frac{1}{\sqrt{\log n}}}}{n^{2\gamma}} \qquad \text{(plugging in the value of } t\text{)}$$

$$= 32n^{1-\gamma+\frac{1}{\sqrt{\log n}}-\delta}$$

$$\leq n^{1-\gamma-\Omega(1)} \leq o(n^{1-\gamma}).$$

Then the proposition follows by an application of Chernoff bounds. $\qquad\square$

Suppose we are now at the end of the algorithm $\mathcal{A}$. Let $\mathcal{D}_{\mathrm{rz}}$ be $\mathcal{D}$ conditioned on all edge slots incident on realized vertex groups, as defined above. Similarly, $G' \sim \mathcal{D}_{\mathrm{rz}}$ is *not* necessarily consistent with the answers we gave to $\mathcal{A}$'s queries that involve unrealized vertex groups, albeit these answers are deterministic by Proposition 4. Also, let $\boldsymbol{a}$ denote the answers we gave to all queries made by $\mathcal{A}$, and let $\mathcal{D}_{\mathrm{rz},\boldsymbol{a}}$ denote the conditional distribution of $\mathcal{D}_{\mathrm{rz}}$ on $\boldsymbol{a}$.

**Lemma 4.** *Let $(S, \bar{S})$ be any fixed cut with $|S| \in [n/3, 2n/3]$. With probability at least $1 - 1/n$, the size of the cut $(S, \bar{S})$ in $G'' \sim \mathcal{D}_{\mathrm{rz},\boldsymbol{a}}$ is at least $n^{2\gamma+\frac{1}{\sqrt{\log n}}}/10^7$.*

*Proof.* Suppose after $\mathcal{A}$ terminates, the number of realized vertex groups is bounded by $o(n^{1-\gamma})$, which by Proposition 6 happens with high probability. Suppose we generate a $G' \sim \mathcal{D}_{\mathrm{rz}}$ using the process in Figure 3. Consider an $S_i$ that is among the first unmatched $n^{1-\gamma}/13$ unrealized vertex groups that we iterate over at Step 4a in Figure 3. We claim that, with probability at least .1 over the choice of $S_j$ matched to $S_i$ and the edges we add between $S_i, S_j$, $t/100$ of the latter edges are across the cut $(S, \bar{S})$.

To prove the claim, note that the number of choices of $S_j$ to be matched to $S_i$ is at least $5n^{1-\gamma}/6$. Let $U$ denote the vertices in these $S_j$'s, and thus we have $|U| \geq 5n/6$. Define $T := S \cap U$ and $T' := \bar{S} \cap U$, which satisfy $|T| + |T'| = |U|$ and $|T| \in [|U|/6, 5|U|/6]$. Then the expected number of edge slots between $S_i, S_j$ that are across the cut $(S, \bar{S})$ is given by

$$\frac{1}{\#j\text{'s}} \sum_j |S_i \cap S||S_j \cap \bar{S}| + |S_i \cap \bar{S}||S_j \cap S|$$

$$= \frac{1}{\#j\text{'s}} \left( |S_i \cap S| \cdot |T'| + |S_i \cap \bar{S}| \cdot |T| \right) \qquad \text{(moving the summation inside)}$$

$$\geq \frac{1}{n^{1-\gamma}} \cdot \frac{|U|}{6} \left( |S_i \cap S| + |S_i \cap \bar{S}| \right) \qquad \text{(as } |T|, |T'| \geq |U|/6\text{)}$$

$$\geq \frac{1}{n^{1-\gamma}} \cdot \frac{5n/6}{6} \cdot n^\gamma \qquad \text{(by } |U| \geq 5n/6\text{)}$$

$$> .13n^{2\gamma}.$$

Then the expected number of edges that we add between $S_i, S_j$ that fall in these slots is at least

$$.13n^{2\gamma} \cdot \frac{2t}{n^{2\gamma}} = .26t.$$

Since the number of edges between $S_i, S_j$ is $2t$, by Markov's inequality, the number such edges across the cut $(S, \bar{S})$ is at least $t/100$ with probability $\geq .1$, as desired.

Thus, for a $G' \in \mathcal{D}_{\mathrm{rz}}$, in expectation, at least $n^{1-\gamma}/130$ of the $S_i$'s satisfy that between $S_i$ and the matched $S_j$, $t/100$ edges are across the cut $(S, \bar{S})$. By a Chernoff bound, with probability at least $1 - e^{-n^{1-\gamma}/500}$, the number of such $S_i$'s is at least $n^{1-\gamma}/1300$, in which case the cut size of $(S, \bar{S})$ in $G'$ is at least

$$\frac{t}{100} \cdot \frac{n^{1-\gamma}}{1300} = 130000^{-1}n^{1-\gamma}n^{\max\{0, 3\gamma-1\}+\frac{1}{\sqrt{\log n}}}$$

$$\geq 10^{-7}n^{2\gamma+\frac{1}{\sqrt{\log n}}}.$$

On the other hand, by Proposition 5, $G'$ is consistent with all answers $\boldsymbol{a}$ that we gave to $\mathcal{A}$ with probability at least

$$.998^{n^{1-\gamma}/2} \geq e^{-.0015n^{1-\gamma}}.$$

As a result, the cut $(S, \bar{S})$ in $G'' \sim \mathcal{D}_{\mathrm{rz},\boldsymbol{a}}$ has size at least $10^{-7}n^{2\gamma+\frac{1}{\sqrt{\log n}}}$ with probability at least $1 - e^{-0.0005n^{1-\gamma}}$, which suffices for proving the lemma. $\qquad\square$

We now conclude this section by proving Theorem 19.

*Proof of Theorem 19.* Let $\mathcal{A}$ be a deterministic algorithm that makes $n^{\min\{1+\gamma, 2-2\gamma\}-\delta}$ queries for some constant $\delta > 0$. Suppose for the sake of contradiction, on an input graph $G \sim \mathcal{D}$, $\mathcal{A}$ outputs with probability $\Omega(1)$ a polylog$(n)$-approximate hierarchical clustering tree. First, we turn this tree into a full binary tree such that the bi-partition of each internal node is $[1/3, 2/3]$-balanced, while increasing the cost by at most an $O(1)$ factor. We then consider the bi-partition of the root, which is a cut $(S, \bar{S})$ with $|S| \in [n/3, 2n/3]$. By Lemma 4, conditioned on the answers $\mathcal{A}$ got, this cut has size at least $n^{2\gamma + \frac{1}{\sqrt{\log n}}}/10^7$ with high probability. However, by Proposition 3, the cost of the optimal hierarchical clustering tree of $G$ is at most $O(n^{1+2\gamma})$. This means that $\mathcal{A}$ only obtains an $n^{o(1)}$-approximation with high probability, a contradiction. $\qquad\square$

# G   A One-Round MPC Lower Bound for $\widetilde{O}(1)$-approximation

**Theorem 20.** *Let $P$ be any one-round protocol in the MPC model where each machine has memory $O(n^{4/3-\varepsilon})$ for any constant $\varepsilon > 0$. Then at the end of the protocol $P$, no machine can output a* polylog$(n)$-*approximate hierarchical clustering tree with probability better than $o(1)$.*

To prove the theorem, we will (i) describe the graph distribution from which we generate an input graph, (ii) specify how we split the input graph across multiple machines, and (iii) analyze the performance of any one-round protocol on such input.

**The hard graph instance.** Let $\varepsilon \in (0, 1/3)$ be an arbitrary constant. We first define a "base" graph $\mathcal{G}$ of $2n$ vertices and $\Theta(n^{5/3-\varepsilon})$ edges as follows. $\mathcal{G}$ consists of two vertex-disjoint parts, each supported on $n$ vertices:

**Part 1:** A union of $n^{1/3+\varepsilon}$ bipartite cliques, each of size $n^{2/3-\varepsilon}$ (with each side having $n^{2/3-\varepsilon}/2$ vertices), supported on vertex set $\mathcal{V}_1$ with $|\mathcal{V}_1| = n$.

**Part 2:** A union of $n^{2/3+\varepsilon}$ bipartite cliques, each of size $n^{1/3-\varepsilon}$ (with each side having $n^{1/3-\varepsilon}/2$ vertices), supported on vertex set $\mathcal{V}_2$ with $|\mathcal{V}_2| = n$ that is disjoint from $\mathcal{V}_1$.

We show that the induced subgraph $\mathcal{G}[\mathcal{V}_1]$ can be tiled using edge-disjoint subgraphs that are isomorphic to $\mathcal{G}[\mathcal{V}_2]$.

**Proposition 7.** *The vertex-induced subgraph $\mathcal{G}[\mathcal{V}_1]$ can be decomposed into $n^{1/3}$ edge-disjoint subgraphs $\mathcal{G}_1, \ldots, \mathcal{G}_{n^{1/3}}$, each supported on $\mathcal{V}_1$ and consisting of $n^{2/3+\varepsilon}$ vertex-disjoint bipartite cliques of size $n^{1/3-\varepsilon}$.*

*Proof.* Consider first arbitrarily partitioning vertices on each side of each bipartite clique in $\mathcal{G}[\mathcal{V}_1]$ into vertex subsets of size $n^{1/3-\varepsilon}/2$, and then collapsing each vertex subset into a supernode. By further treating the parallel edges between a same pair of supernodes as a single edge, we have made $G[\mathcal{V}_1]$ a union of $n^{1/3+\varepsilon}$ bipartite cliques each supported on $2n^{1/3}$ supernodes (thus we have $2n^{2/3+\varepsilon}$ supernodes in total). Note that in this contracted version of $\mathcal{G}[\mathcal{V}_1]$, each perfect matching between the $2n^{2/3+\varepsilon}$ supernodes correspond to an edge-induced subgraph that is isomorphic to $\mathcal{G}[\mathcal{V}_2]$. It is now not hard to show that this contracted version of $\mathcal{G}[\mathcal{V}_1]$ can be decomposed into $n^{1/3}$ edge-disjoint perfect matchings, which proves the proposition. $\qquad\square$

In what follows, we will fix an arbitrary such tiling $\mathcal{G}_1, \ldots, \mathcal{G}_{n^{1/3}}$. We next define a distribution $\mathcal{D}$ such that a graph $G \sim \mathcal{D}$ is generated by permuting the vertices of $\mathcal{G}$ uniformly at random. Let $\pi : [2n] \to [2n]$ be the permutation we use to generate $G$. We will then let $V_1, V_2$ be, respectively, $\mathcal{V}_1, \mathcal{V}_2$ under the vertex permutation $\pi$. We also use $G_1, \ldots, G_{n^{1/3}}$ to denote $\mathcal{G}_1, \ldots, \mathcal{G}_{n^{1/3}}$ under the vertex permutation $\pi$, where the former form an edge-disjoint tiling of $G[V_1]$, and each $G_i$ is isomorphic to $G[V_2]$.

The next proposition bounds the optimal hierarchical clustering cost for any input graph $G$ generated as above.

**Proposition 8.** *The optimal hierarchical clustering tree of $G$ has cost at most $O(n^{7/3-2\varepsilon})$.*

*Proof.* We construct a hierarchical clustering tree by the following steps. At the root of the tree, we divide the entire vertex set into $n^{1/3+\varepsilon} + n^{2/3+\varepsilon}$ clusters with each cluster being a connected component. This incurs zero cost of the tree. We next construct a binary hierarchical clustering tree of each cluster arbitrarily. If a cluster is a bipartite clique of size $n^{2/3-\varepsilon}$, then we incur a cost of at most $O(n^{2-3\varepsilon})$ (Fact 2). If a cluster is a bipartite clique of size $n^{1/3-\varepsilon}$, then we incur a cost of $O(n^{1-3\varepsilon})$. Thus the total cost is $O(n^{2-3\varepsilon}) \cdot n^{1/3+\varepsilon} + O(n^{1-3\varepsilon}) \cdot n^{2/3+\varepsilon} \le O(n^{7/3-2\varepsilon})$. $\qquad\square$

**The MPC input distribution.** Consider that in the MPC model each machine has $\Theta(n^{4/3-\varepsilon}\log n)$ bits of memory, and there are in total $\Theta(n^{1/3})$ machines. We will give $G[V_2]$ to a uniformly random machine. We then give each of $G_1, \ldots, G_{n^{1/3}}$ to a uniformly random remaining machine, while ensuring that each machine gets at most one subgraph $G_i$.

Note that, each machine's input has exactly the same distribution. Namely, each machine has the same probability of having a non-empty graph. Moreover, for each machine, conditioned on that it gets at least one edge, the graph it gets is a union of $n^{2/3+\varepsilon}$ bipartite cliques of size $n^{1/3-\varepsilon}$ plus $n$ isolated vertices, with all vertices permuted uniformly at random. However, the input distributions of different machines are correlated.

**Analysis of one-round protocols on the input distribution.** We show that for any one-round protocol $P$, no machine can output a polylog$(n)$-approximate hierarchical clustering tree of $G$ with probability $\Omega(1)$. We will do so by a reduction from a two-party one-way communication problem, which we define next.

We specifically consider the following one-way communication problem in the two-party model, with players Alice and Bob who have shared randomness. Alice is given as input a graph $H$ on $n$ vertices, which is obtained by first taking a union of $n^{1-\gamma}$ bipartite cliques each of size $n^{\gamma}$ (with each side having $n^{\gamma}/2$ vertices), for some constant $\gamma \in (0,1)$, and then permuting the $n$ vertices uniformly at random. The goal of this communication problem is as follows:

> For Alice to send Bob a single (possibly randomized) message such that, for some constant $\delta > 0$, Bob can then output with probability $\Omega(1)$ a cut $(S, \bar{S})$ in $H$ with $|S| \in [n/3, 2n/3]$ and size at most $O(n^{1+\gamma-\delta})$. $\quad(\star)$

We show that this problem requires $\Omega(n)$ communication. In particular, we will prove the following theorem in Section G.1.

**Theorem 21.** *For any constant $\gamma \in (0,1)$, Alice needs to send a message of size $\Omega(n)$ to achieve goal $(\star)$.*

To reduce this two-party communication problem to our MPC problem, we prove the following lemma.

**Lemma 5.** *Suppose for $\varepsilon > 0$, there exists a one-round protocol $P$ in the MPC model with $\Theta(n^{4/3-\varepsilon}\log n)$ bits of memory per machine such that, at the end of $P$, some machine can output a polylog$(n)$-approximate hierarchical clustering tree with probability $\Omega(1)$. Then there exists a one-way protocol $Q$ with message size $o(n)$ in the two-party communication model that achieves goal $(\star)$ for $\gamma = 1/3 - \varepsilon$.*

*Proof.* Suppose at the end of protocol $P$, some machine $M^*$ can output with probability $\Omega(1)$ a polylog$(n)$-approximate hierarchical clustering tree. We first show that such a protocol $P$ implies another protocol $P'$ in which $M^*$ can output a balanced cut of $G[V_2]$ with small size.

**Claim 1.** *There is another one-round protocol $P'$ in the MPC model in which $M^*$ can find with probability $\geq \Omega(1)$ a vertex set $S \subset V_2$ such that $|S| \in [n/3, 2n/3]$ and the number of edges between $S$ and $V_2 \setminus S$ in $G$ is at most $n^{4/3-2\varepsilon}$polylog$(n)$.*

*Proof.* $P'$ proceeds by first simulating $P$, and then, in parallel, letting each machine $M \neq M^*$ sample its input edges with probability $\frac{100\log n}{n^{2/3-\varepsilon}}$ and send the sampled edges to $M^*$. With high probability, the total number of edges that $M^*$ receives is $O(n\log n)$, and each bipartite clique of size $n^{2/3-\varepsilon}$ in $G[V_1]$ is connected by the sampled edges. Therefore, by looking at the connected components of size $n^{2/3-\varepsilon}$ in the subsampled graph, $M^*$ can recover $V_1, V_2$ exactly.

$M^*$ then uses the protocol $P$ to output a hierarchical clustering tree $\mathcal{T}$. We first make $\mathcal{T}$ a fully binary tree without increasing the cost. We then look at an internal node of $\mathcal{T}$ corresponding to a vertex set $T$ such that $|T \cap V_2| \in [n/3, 2n/3]$. One can show that such an internal node exists by starting at the root of the tree and keeping moving to the child that has a larger intersection with $V_2$ until finding a desired node. Let $S := T \cap V_2$. If we consider the edges between $S$ and $V_2 \setminus S$ in $G$, each of them incurs a cost of at least $|T| \geq n/3$ in $\mathcal{T}$. Since the cost of $\mathcal{T}$ is at most $n^{7/3-2\varepsilon}$polylog$(n)$ with probability $\Omega(1)$, the number of edges between $S$ and $V_2 \setminus S$ must be bounded by $n^{4/3-2\varepsilon}$polylog$(n)$ with probability $\Omega(1)$, as desired. $\qquad\square$

We now show how to use the protocol $P'$ to construct a one-way protocol $Q$ in the two-party communication model that achieves goal $(\star)$ with $\gamma = 1/3-\varepsilon$. First note that, since the memory per machine is $\Theta(n^{4/3-\varepsilon}\log n)$, the total message size received by $M^*$ is at most $\Theta(n^{4/3-\varepsilon}\log n)$. This means that, on average, another machine sends a message of size $\Theta(n^{1-\varepsilon}\log n)$ to $M^*$. For each machine $M_i$, let $p_i$ be the success probability of $P'$ conditioned on that $G[V_2]$ is given to machine $M_i$. Since $P'$ succeeds with probability $\Omega(1)$, the average of $p_i$ must be $\Omega(1)$. Based on the above two observations, by applying Markov's inequality twice and then a union

bound, we have that there exists a machine $M_j \neq M^*$ that sends $M^*$ a message of size $O(n^{1-\varepsilon} \log^2 n)$ and has $p_j \geq \Omega(1)$.

The protocol $Q$ proceeds as follows. Upon receiving the input graph, which is a union of $n^{2/3+\varepsilon}$ bipartite cliques of size $n^{1/3-\varepsilon}$, Alice shall treat its vertices as $V_2$ and add $n$ isolated vertices as $V_1$. Then she uses her shared randomness with Bob to permute the $2n$ vertices uniformly at random. Note that now the new graph has the exact same distribution as the input given to machine $M_j$ in the MPC model conditioned on $G[V_2]$ being given to $M_j$. Alice then uses the same message generation algorithm as $M_j$ to produce a message and sends it to Bob.

Upon receiving the message from Alice, Bob himself then simulates protocol $P'$ for other machines $M_i \neq M_j$ by generating their inputs conditioned on the realization of $V_1, V_2$ and simulating their message generation algorithms. Finally, Bob runs the recovery algorithm of $M^*$ to recover a vertex set $S \subset V_2$, which satisfies with probability $= p_j \geq \Omega(1)$ that $|S| \in [n/3, 2n/3]$ and that the number of edges between $S, V_2 \setminus S$ is at most $n^{4/3-2\varepsilon}\mathrm{polylog}(n) \leq O(n^{4/3-1.5\varepsilon})$. This means that the protocol $Q$ achieves goal $(\star)$ for $\gamma = 1/3 - \varepsilon$ with message size $O(n^{1-\varepsilon} \log^2 n) \leq o(n)$. $\qquad\square$

Lemma 5 and Theorem 21 together rule out any one-round protocol in the MPC model with $O(n^{4/3-\varepsilon})$ memory per machine for any constant $\varepsilon > 0$, and thus prove Theorem 20.

## G.1 A Lower Bound in the Two-Party Communication Model

In this section we prove Theorem 21, which gives a lower bound on the communication needed to achieve goal $(\star)$. We first show that any cut in the input graph $H$ given to Alice that has size $\leq O(n^{1+\gamma-\Omega(1)})$ can be made into a cut of size $0$ by changing the sides of an $o(1)$ fraction of vertices.

**Proposition 9.** *For any cut $(S, \bar{S})$ in $H$ with size at most $O(n^{1+\gamma-\delta})$ for any constant $\delta > 0$, one can obtain from it another cut of size $0$ by switching the sides of at most $O(n^{1-\delta})$ vertices.*

*Proof.* Let us call the bipartite cliques in $H$ $C_1, \ldots, C_{n^{1-\gamma}}$ with $C_i$ supported on vertices $S_i$. For each $i$, let $s_i := |S_i \cap S|$ and $t_i := |S_i \cap \bar{S}|$. Then the number of edges of $C_i$ across the cut $(S, \bar{S})$ is at least $\Omega(1) \cdot \min\{s_i, t_i\} \cdot n^\gamma$. This means that we have

$$\Omega(1) \cdot \sum_i \min\{s_i, t_i\} \cdot n^\gamma \leq O(n^{1+\gamma-\delta})$$

which by rearranging gives

$$\sum_i \min\{s_i, t_i\} \leq O(n^{1-\delta}).$$

Therefore we can obtain a new cut of size $0$ from $(S, \bar{S})$ by switching the sides of the vertices that correspond to the summation on the LHS of the above inequality, whose total number is at most $O(n^{1-\delta})$. $\qquad\square$

Using the above lemma, we then show that if one can achieve $(\star)$ using $o(n)$ communication, then one can also output a balanced cut with size $0$ using $o(n)$ communication.

**Lemma 6.** *If there is a protocol that achieves goal $(\star)$ with message size $o(n)$, then there is another protocol in which Alice sends Bob a message of size $o(n)$, such that Bob can then output with probability $\Omega(1)$ a cut $(S', \bar{S'})$ in $H$ with $|S'| \in [n/6, 5n/6]$ and size $0$.*

*Proof.* We will design the second protocol by simulating the first protocol. To this end, let us fix a protocol $P$ that achieves goal $(\star)$ for some $\delta > 0$, which consists of a message generation algorithm $\mathcal{A}$ for Alice and a cut recovery algorithm $\mathcal{B}$ for Bob. At the start, Alice runs $\mathcal{A}$ to generate a message $M$ of size $o(n)$. Then, Alice and Bob use their shared randomness to generate sufficiently many random bits for running $\mathcal{B}$. Alice first runs $\mathcal{B}$ given the message $M$ on her own and gets a cut $(S, \bar{S})$. If the cut satisfies $|S| \in [n/3, 2n/3]$ and has size at most $O(n^{1+\gamma-\delta})$, then Alice sends Bob the message $M$ along with a subset $U$ of $O(n^{1-\delta})$ vertices whose switching sides makes cut $(S, \bar{S})$ have zero size (existence guaranteed by Proposition 9). Then Bob, upon receiving the message $M$ and the subset $U$ of vertices, runs the recovery algorithm $\mathcal{B}$ using the shared random bits with Alice and gets the same cut $(S, \bar{S})$ with $|S| \in [n/3, 2n/3]$ as Alice. By switching the sides of vertices in $U$, Bob then gets a cut $(S', \bar{S'})$ with $|S'| \in [n/6, 5n/6]$ and size $0$, as desired. $\qquad\square$

In light of the above lemma, we now consider another one-way two-party communication problem, where Alice gets a same input graph $H$ obtained by first taking a union of $n^{1-\gamma}$ bipartite cliques of size $n^\gamma$ and then permuting all $n$ vertices uniformly at random, and the goal is

*For Alice to send Bob a single (possibly randomized) message such that Bob can output with probability $\Omega(1)$ a cut $(S, \bar{S})$ in $H$ that satisfies $|S| \in [n/6, 5n/6]$ and has size 0.*   $(\star\star)$

We show that it requires $\Omega(n)$ communication to achieve $(\star\star)$.

**Lemma 7.** *For any constant $\gamma \in (0, 1)$, Alice needs to send a message of size $\Omega(n)$ to achieve goal $(\star\star)$.*

*Proof.* Let $P$ be a protocol that achieves $(\star\star)$. Then on an input graph $H \sim \mathcal{D}$, at the end of the protocol $P$, Bob outputs with probability $\Omega(1)$ a cut $(S, \bar{S})$ in $H$ with $|S| \in [n/6, 5n/6]$ and size 0. We now analyze the entropy of the distribution $\mathcal{D}$ and that of $\mathcal{D}$ conditioned on the message $M$ Bob receives from Alice. First, note that an input graph $H$ can be determined by first dividing the $n$ vertices into $n^{1-\gamma}$ groups each of size $n^\gamma$, and then picking a balanced bi-partition for each group. Thus the total number of different $H$'s can be calculated by

$$N_1 \stackrel{\text{def}}{=} \left( \frac{1}{n^{1-\gamma}!} \prod_{i=1}^{n^{1-\gamma}} \binom{n - (i-1)n^\gamma}{n^\gamma} \right) \cdot \left( \binom{n^\gamma}{n^\gamma/2}/2 \right)^{n^{1-\gamma}}. \tag{13}$$

Thus the entropy of $\mathcal{D}$ is

$$H(\mathcal{D}) = \log_2 N_1.$$

On the other hand, by Fano's inequality, we have

$$H(\mathcal{D}|M) \le H(\mathcal{D}|(S, \bar{S})). \tag{14}$$

We then do a case analysis to calculate $H(\mathcal{D}|(S, \bar{S}))$:

**Case 1:** The cut $(S, \bar{S})$ satisfies $|S| \in [n/6, 5n/6]$ and has size 0, which happens with probability $\Omega(1)$. Since the cut $(S, \bar{S})$ does not cut through any cliques, it must be that $S$ contains entirely $|S|/n^\gamma$ cliques and $\bar{S}$ contains entirely the remaining $|\bar{S}|/n^\gamma$ cliques. Therefore, the total number of graphs $H$ that is consistent with this profile is

$$N_2 \stackrel{\text{def}}{=} \left( \frac{1}{\frac{|S|}{n^\gamma}! \frac{|\bar{S}|}{n^\gamma}!} \prod_{i=1}^{|S|/n^\gamma} \binom{|S| - (i-1)n^\gamma}{n^\gamma} \prod_{j=1}^{|\bar{S}|/n^\gamma} \binom{|\bar{S}| - (j-1)n^\gamma}{n^\gamma} \right) \cdot \left( \binom{n^\gamma}{n^\gamma/2}/2 \right)^{n^{1-\gamma}}.$$

Thus the entropy of $\mathcal{D}$ conditioned on the cut $(S, \bar{S})$ is

$$H(\mathcal{D}|(S, \bar{S})) = \log_2 N_2.$$

We can then calculate the difference between the entropy $H(\mathcal{D})$ and $H(\mathcal{D}|(S, \bar{S}))$ by

$$H(\mathcal{D}) - H(\mathcal{D}|(S, \bar{S}))$$

$$= \log_2 \frac{N_1}{N_2}$$

$$= \log_2 \frac{n!}{|S|!|\bar{S}|!} \frac{\frac{|S|}{n^\gamma}! \frac{|\bar{S}|}{n^\gamma}!}{n^{1-\gamma}!} \qquad \text{(plugging in the values of } N_1, N_2)$$

$$= \log_2 \binom{n}{|S|} \binom{n^{1-\gamma}}{|S|/n^\gamma}^{-1} \qquad \text{(rewriting as binomials)}$$

$$\ge \log_2 \left( \frac{n}{|S|} \right)^{|S|} \left( \frac{|S|/n^\gamma}{en^{1-\gamma}} \right)^{|S|/n^\gamma} \qquad \left( \text{as } \left( \frac{n}{k} \right)^k \le \binom{n}{k} \le \left( \frac{en}{k} \right)^k \text{ for any } n, k \right)$$

$$= \log_2 \left( \frac{n}{|S|} \right)^{|S|(1-1/n^\gamma)} e^{-|S|/n^\gamma}$$

$$\ge \log_2 1.2^{(n/6)(1-100/n^\gamma)} \qquad \text{(as } |S| \in [n/6, 5n/6])$$

$$\ge \Omega(n).$$

**Case 2:** The cut $(S, \bar{S})$ does not satisfy $|S| \in [n/6, 5n/6]$ or has nonzero size, which happens with probability $1 - \Omega(1)$. Since conditioning can only reduce entropy, we have

$$H(\mathcal{D}) - H(\mathcal{D}|(S, \bar{S})) \ge 0.$$

Combining the above two cases with (14), we have

$$H(\mathcal{D}) - H(\mathcal{D}|M) \ge \Omega(n).$$

On the other hand, by the chain rule, we have

$$H(\mathcal{D}|M) + H(M) = H(\mathcal{D}, M) \ge H(\mathcal{D}).$$

As a result, $H(M) \ge \Omega(n)$, and therefore $M$ must contain $\Omega(n)$ bits. $\qquad \square$

We now prove Theorem 21.

*Proof of Theorem 21.* Suppose for the sake of contradiction there exists a protocol with message size $o(n)$ that achieves goal $(\star)$. Then by Lemma 6, there also exists a protocol with message size $o(n)$ that achieves goal $(\star\star)$, contradicting Lemma 7. Therefore Alice needs to send a message of size $\Omega(n)$ in order to achieve goal $(\star)$, as desired. $\qquad\square$

# H    Experimental Results

We now present our experimental results. The goal of our experiments is to test our data/graph sparsification framework (Algorithm 3) which forms the basis of all our algorithms under the different models of computation. We ask the following question: if we run an HC algorithm on the sparsified graph produced by Algorithm 3, then what will be the loss in the quality of solution compared to the same HC algorithm run on the original graph? We would also like to understand the tradeoff between the sparsification rate and the solution quality of certain HC algorithms.

We test our data sparsification algorithm on two large real-world datasets Boston and Newsgroup, which are both part of the standard Scikit-learn library and were used in previous HC papers [18, 42]. We calculate the similarity between two data points using the Gaussian kernel as in [42, 37] and only insert an edge if its weight (i.e. similarity between its two endpoints) is bigger than $10^{-10}$.

Since we are not aware of any existing implementation of the recursive sparsest cut algorithm, we choose to test two linkage algorithms, namely average linkage and complete linkage, which, as shown in [37], are among the best-performed algorithms on the two datasets we consider. We run these two algorithms both on the entire graph and on the sparsified graphs obtained by our Algorithm 3, and compare the cost of the HC we get in each case, where the cost is measured using Dasgupta's objective function (Eq. (1)). When running our Algorithm 3, we also try different sampling rates to obtain subgraphs of various edge densities. Note that, we always measure the cost of an HC in the original graph as opposed to in the subsampled graph.

We present our experimental results in the tables below. Here we write absolute HC cost to denote the cost of the HC obtained by running the corresponding algorithm on the entire graph, and write relative HC cost to denote the cost of the HC obtained by running the algorithm on the subsampled graph divided by the absolute HC cost.

For the Boston dataset, one can see that even when we only sample <5% of the edges, we already get a very good relative HC cost of 1.15; for the Newsgroup dataset, sampling only <0.5% of the edges already gives us a low relative HC cost of 1.188 for average linkage and 1.05 for complete linkage. As the number of sampled edges grow, the relative HC cost becomes smaller and gets closer to 1.

Table 2: Performance of average linkage on Boston dataset (506 vertices, 95566 edges), with absolute HC cost 186085.271.

| Number (Percentage) of Sampled Edges | Relative HC Cost |
|:---:|:---:|
| 4040 (4.23%) | 1.146 |
| 5861 (6.13%) | 1.068 |
| 7010 (7.34%) | 1.043 |
| 9018 (9.44%) | 1.016 |
| 9756 (10.21%) | 1.012 |
| 11941 (12.50%) | 1.000 |

Table 3: Performance of complete linkage on Boston dataset (506 vertices, 95566 edges), with absolute HC cost 226485.146.

| Number (Percentage) of Sampled Edges | Relative HC Cost |
|:---:|:---:|
| 4032 (4.22%) | 1.158 |
| 5885 (6.16%) | 1.135 |
| 7002 (7.33%) | 1.113 |
| 9033 (9.45%) | 1.066 |
| 9773 (10.23%) | 1.037 |
| 11900 (12.45%) | 1.006 |

Table 4: Performance of average linkage on Newsgroup dataset (3516 vertices, 6178853 edges), with absolute HC cost 3960145237.881.

| Number (Percentage) of Sampled Edges | Relative HC Cost |
| --- | --- |
| 28495 (0.46%) | 1.188 |
| 55116 (0.89%) | 1.169 |
| 103804 (1.68%) | 1.143 |
| 203095 (3.29%) | 1.118 |
| 358440 (5.80%) | 1.111 |
| 1029648 (16.66%) | 1.068 |

Table 5: Performance of complete linkage on Newsgroup dataset (3516 vertices, 6178853 edges), with absolute HC cost 3960674776.482.

| Number (Percentage) of Sampled Edges | Relative HC Cost |
| --- | --- |
| 28476 (0.46%) | 1.050 |
| 55042 (0.89%) | 1.010 |
| 103756 (1.68%) | 1.001 |
| 203323 (3.29%) | 1.000 |
| 358446 (5.80%) | 1.000 |
| 1029274 (16.66%) | 1.000 |