# OpenReview forum: "Sublinear Algorithms for Hierarchical Clustering"
_NeurIPS.cc/2022/Conference — NeurIPS 2022 Accept_

### Official Review · Reviewer_kk3J · 2022-07-11

**Rating:** 7
**Confidence:** 3
**Soundness:** 3 good
**Presentation:** 3 good
**Contribution:** 3 good

**Summary:**

This paper focuses on hierarchical clustering over graphs with the objective function of Dasgupta. Sublinear algorithms w.r.t. the number of edges in the input graph are developed in three models of computation, including the dynamic streaming model, the query model, and the massively parallel computation (MPC) model. Interesting matching lower bounds are provided for the upper bounded resource above. Specifically, a 1-pass semi-streaming algorithm naturally follow from existing algorithm [1] and their observation. In the query model, combining existing methods [14,37] and their observation leads to sublinear time algorithms for graphs of different sparsity. For the MPC model, a 2-round $\tilde{O}(m)$-memory algorithm and a 1-round $\tilde{\Theta}(m^{4/3})$-memory algorithm are developed based on [1]. All above algorithms rely on the crucial observations that, from a graph cut perspective, the objective function can be well approximated in a cut sparsifier of the input graph. Moreover, two hard instances are designed to derive time and communication lower bounds in the query and MPC models, respectively. Finally, the extensions to other related hierarchical clustering objectives such as the dissimilarity [15] are discussed.

**Questions:**

In the alternative view of the objective based on S-T cut, a sparsifier approximately preserving all S-T cut values suffers from the lower bound of $\Omega(m)$. When transforming the S-T cut to graph cut ($S,\bar{S}$)-cut, why the barrier of lower bound can be broken?

**Strengths And Weaknesses:**

Strengths:
1. The idea of using cut sparsifiers in improving the computational resource of graph algorithms has been widely used. But using it in hierarchical clustering and providing theoretical analysis on the required resource and approximation factor is not well studied with only a few related work, e.g., [5]. The problem of hierarchical clustering is reduced to the problem of constructing cut sparsifiers with limited resource based on the observations that cut sparsifiers can approximately preserve the cost of a dendrogram. Comprehensive studies on 3 models of computation are performed and upper bounds are complemented with almost matching lower bounds.
2. The developed lower bounds in the query and MPC models are quite interesting and need to overcome multiple challenges. The authors are able to describe the high-level overview in the main text and then refer to the full details in the Appendix.
3. The organization of this work is good. I like the discussions in Sections 4 and 5, providing context on related works and extensions to other objective functions of hierarchical clustering. I browsed some contents in the Appendix and the quality of writing remains good compared to the main text.

Weaknesses:
1. The computational upper bounds developed in Section 2 are not very challenging and based on existing methods in the respective models. The upper and lower bounds in the dynamic streaming model are straightforward.
2. As a reader of a hierarchical clustering paper in NeurIPS, I would expect to see some experimental results on the developed algorithm unless the theoretical contributions are very novel and significant. It would be much better if experiments on some of the algorithms had been implemented and evaluated in real-world datasets.

---

> ### Author Response · Authors · 2022-08-02
> **Response to Reviewer kk3J**
>
> We thank you for your positive feedback and shall address the concerns you have, which are (a) our streaming results being not very challenging (b) experimental results, and (c) answer the question you raise about the $\Omega(m)$ lower bound.
>
> (a) We agree that our streaming results are not particularly difficult to obtain compared to the other results. However, this is only made possible by our novel and unifying framework, which, building on our structural finding for HC (Thm 1), allows one to perform HC computation on a significantly sparser graph while only losing a negligible factor in the solution quality. The fact that our streaming results are not difficult to obtain in fact further demonstrates the power of our framework.
>
> (b) While the primary focus of our paper is to provide a comprehensive picture of HC in the three well-studied sublinear computation models, in light of your suggestion, we conducted some preliminary experiments to test our framework on real-world datasets. We intend to add a more comprehensive set of experiments in the final version of the paper. In particular, we implemented our data sparsification algorithm and recorded the tradeoff between the sparsification rate and the solution quality of certain HC algorithms. We present our experimental results in a separate response (visible to all) and will include them in the final version. These results indicate that our algorithm preserves the solution quality particularly well, while reducing the graph size by a significant amount.
>
> (c) We now answer your question. We first explain why one necessarily needs to store $\Omega(m)$ edges even to *approximately* preserve the number of edges between any two disjoint vertex subsets $S$ and $T$ to some multiplicative factor. This is because such an approximation guarantee allows one to recover *all* edges in the graph - in particular, since for any pair of vertices $u,v\$, the number of edges between $S = \\{u\\}$ and $T = \\{v\\}$ is preserved multiplicatively, we know whether or not an edge exists between any $u,v$. This means that one has to store all $m$ edges of the graph.
>
> When transforming this guarantee to preserve all $(S,\bar{S})$ cut values approximately, we cannot argue the same thing that one has to store all edges. In fact, it is known that a subgraph of $\tilde{O}(n)$ edges that approximately preserves all cut values exists (a.k.a. a cut sparsifier). This first idea allows us to break the $\Omega(m)$ space lower bound barrier. However, this by itself is not sufficient as previous works also show that to find such an $\tilde{O}(n)$-size cut sparsifier necessarily requires $\Omega(m)$ queries (time). We thus get around this obstacle by relaxing the notion of cut sparsifiers by also allowing *additive* errors (in addition to the multiplicative ones) in cut approximations, which allows us to break the $\Omega(m)$ query/time barrier. These two ideas in combination allow us to obtain both sublinear time and space algorithms for HC.

---

### Official Review · Reviewer_1tqq · 2022-07-11

**Rating:** 3
**Confidence:** 3
**Soundness:** 3 good
**Presentation:** 1 poor
**Contribution:** 1 poor

**Summary:**

Hierarchical clustering over graphs ha been mathematically formalized with a natural objective function introduced by Dasgupta [STOC2016]. Unfortunately, this function is hard to optimize and approximation algorithms have been proposed in the TCS literature. This paper studies this problem in the regime of sublinear computational resources, specifically, for three models of computations:
- streaming model = sublinear space
- query model = sublinear time
- MPC model = sublinear communication
For each model, upper and lower bounds are provided.

**Questions:**

I am not sure I understand the motivation for the authors to submit this paper to Neurips and I would like to hear their arguments explaining why it is more appropriate to publish this work at Neurips instead of a TCS conference like SODA, STOC, FOCS? I did count 12 references in JACM, STOC, SODA and FOCS.
I like cross-disciplinary work but then you need a lot more work to motivate your results. As it is written, I hardly see any possible impact for this paper in the ML community. Even the computational models will only be known from a minority of the audience, so that the main results will not be understood.
Similarly, the lower bounds are obtained by constructing very contrived graphs. I understand that these hard instances are instructive in order to understand the worst-case performance of algorithms. But what is the motivation for these bounds in a machine learning setting?

**Strengths And Weaknesses:**

The paper presents informal statements for the main results proven in Appendix. The results appear to be new. There is a recent COLT paper [5] with similar results in the streaming setting.
This paper is well-written and deals with theoretical results.
The motivation for these results is very far from any practical application.

---

> ### Author Response · Authors · 2022-08-02
> **Response to Reviewer 1tqq**
>
> We thank you for your review, and shall address all three of your concerns regarding our work, which to our understanding are (a) choice of venue for this work, (b) its impact in practice, and (c) our lower bound construction.
>
> (a) Regarding the relevance of our work at NeurIPS, we begin by emphasizing the importance of hierarchical clustering (HC) in ML and Data Science, where it is used as an essential tool for data analysis and visualization. Moreover, despite its long history and practical importance, we are only just beginning to understand its theoretical underpinnings, and there is significant interest in the ML community to do so as demonstrated by more recent works on this problem at ML venues. In our paper, we in fact cite 12 other works directly on HC appearing across JMLR, COLT, NeurIPS, ICML and AISTATS, all of which are predominantly theoretical takes on HC, including [32], a completely theoretical follow up to the original paper of [16] that proposes the maximization dual objective for HC, which was an oral presentation at NeurIPS 17. A substantial justification behind your assessment seems to be some of our citations, 12 of which are from the TCS community. However, we would like to point out that just 4 of them are actually relevant to the hierarchical clustering (HC) problem itself [9,10,15,16]. The rest are predominantly technical graph-theoretic tools that we leverage in the design of our algorithms for this problem, which is natural considering the graphical representation of the input data to the HC problem. While we understand your viewpoint of our work being well-suited for a TCS audience, one can make perhaps an even stronger argument for its placement at NeurIPS, and more generally, the ML community, especially given both our fundamental structural findings for HC (Thm 1), as well as the practical ramifications of our work which we shall discuss next.
>
> (b) Our work does in fact have a very significant impact in the real world. In particular, our sublinear time result in Appendix D by itself implies substantially faster HC algorithms in practice in the following sense - give us your preferred HC algorithm, then we show how to cleverly subsample data in a way such that you can run your algorithm much faster on this sparsified data, at the cost of a negligible degradation in the quality of the solution returned (which is also easy to implement given data in an adjacency array). Such a result would be of significant value to any practitioner, especially one in this age of big data. Regarding our other two algorithmic contributions in the streaming and MPC models, with due respect, we strongly disagree with the reviewer's opinion that these said computational models are of little relevance to the ML community. In fact, we cite several works in the ML space that specifically address the HC problem in exactly these models of computation - see for e.g. [5] (COLT 22), [34] (ICML 21), both of which deal with streaming algorithms for HC and [40] (ICML 18) which deals with MPC algorithms for HC. We can point to even more works in just this ICML (2022) and the previous NeurIPS (2021) for clustering in these computation models ([C4] for Streaming, [C1,C3] for MPC, [C2] for Query model) and about a dozen more that generally address learning problems in precisely these computation models. We would again emphasize that these models are in fact commonly accepted theoretical abstractions for very practical modern frameworks of computation, specifically, streaming model for online arrival of data (dynamic streaming for dynamically changing data), query model for adjacency list/matrix representation of graphs, and MPC model for the MapReduce paradigm. Designing algorithms in these frameworks does have a practical value.

---

> > ### Author Response · Authors · 2022-08-02
> > **Continuation of Response to Reviewer 1tqq**
> >
> > (c) On a more technical front, you suggest that our lower bound construction is contrived. However, we would like to point out that lower bounds in general are highly specific constructions meant to expose the fundamental hardness inherent in the problem of interest, demonstrating the limit of what is achievable if one desires broad performance guarantees. With regards to our problem, we can easily bypass our lower bound and give a $\tilde{O}(n)$ time or 1-round $\tilde{O}(n)$ memory MPC algorithm for all edge density regimes by making a simple edge expansion assumption on the input graph. However, we would argue that this in fact would be an example of a contrived result. While we often make simplifying assumptions to make otherwise hard problems tractable, there is often little evidence to support the validity of said assumptions for the data observed in practice. There is no reason to believe real data comes from a well formulated generative process. We further point out the fact that our (time/query) lower bound only applies up until edge density $O(nˆ(4/3))$, which is still an arguably sparse data regime that doesnt practically require substantial computation time anyway. Beyond this density, we achieve strictly sublinear time guarantees, dropping to just $\tilde{O}(n)$ at all edge densities above $O(nˆ(3/2))$. In other words, the actual data-dense regimes which in the real world would be the most expensive ones in terms of computation time, are in fact the easy case, which is apriori not at all obvious. This is a substantial merit of our work. For our MPC results, our lower bound of $\tilde{O}(nˆ(4/3))$ machine memory completely goes away given just 1 additional MPC round. In a real world scenario, there is practically no difference between a 1-round and 2-round MPC algorithm. The fact that our 2-round MPC algorithm for HC works with no assumptions on the input data is also a substantial merit.
> >
> > In summary, there is significant evidence to support not just the relevance of our work in the ML community, but also its substantial broader impact, both in advancing our understanding of this problem on a theoretical front, as well as providing new and computationally efficient tools for solving this problem on a practical front. We would kindly request the reviewer to reevaluate their assessment of our work in light of this discussion, and we would be happy to answer any additional questions the reviewer might have regarding the technical contents of our work.
> >
> >
> >
> > [C1] Cohen-Addad V, Mirrokni V, Zhong P. Massively Parallel $ k $-Means Clustering for Perturbation Resilient Instances. InInternational Conference on Machine Learning 2022 Jun 28 (pp. 4180-4201). PMLR.
> >
> > [C2] Neumann S, Peng P. Sublinear-Time Clustering Oracle for Signed Graphs. InInternational Conference on Machine Learning 2022 Jun 28 (pp. 16496-16528). PMLR.
> >
> > [C3] Cohen-Addad V, Lattanzi S, Norouzi-Fard A, Sohler C, Svensson O. Parallel and Efficient Hierarchical k-Median Clustering. Advances in Neural Information Processing Systems. 2021 Dec 6;34:20333-45.
> >
> > [C4] Wu Y, Tardos J, Bateni M, Linhares A, Goncalves de Almeida FM, Montanari A, Norouzi-Fard A. Streaming Belief Propagation for Community Detection. Advances in Neural Information Processing Systems. 2021 Dec 6;34:26976-88.

---

### Official Review · Reviewer_pUsh · 2022-07-12

**Rating:** 8
**Confidence:** 3
**Soundness:** 3 good
**Presentation:** 4 excellent
**Contribution:** 4 excellent

**Summary:**

This paper considers developing resource efficient algorithms for hierarchical clustering of weighted graphs in several settings (streaming, graph query model, mpc model).  Following recent work which developed an objective function for hierarchical clustering (Dasgupta 2016), the goal is to develop algorithms with sublinear resource usage (in the number of edges) in each model while producing an $O(\phi)$-approximation to the optimal tree for Dasgupta's objective function.  The main results of this paper are to achieve exactly this.  By making a key observation about the structure of the objective function, the authors show that this problem reduces to efficiently constructing a cut sparsifier.  In some settings (e.g. streaming using $\tilde{O}(n)$ space), this can be done immediately via known techniques for constructing cut sparsifiers.  For the other settings (graph query model and the mpc model), a more relaxed notion of cut sparsifier is proposed (which allows for some additive error in addition to the multiplicative error), and the authors give sublinear algorithms for constructing such a relaxed cut sparsifier in the remaining settings.  Lower bounds showing near optimality of the results are also given.

**Questions:**

How would the proposed algorithms compare to practically efficient algorithms? e.g. Sumengen, Baris, et al. "Scaling hierarchical agglomerative clustering to billion-sized datasets." arXiv preprint arXiv:2105.11653 (2021).

**Strengths And Weaknesses:**

There has been significant interest in hierarchical clustering recently and one issue that many algorithms suffer from is scalability.  This paper helps to address this problem in several settings from the perspective of Dasgupta's objective function.  The techniques are natural and are well explained.

One thing that is missing from this paper (but not probably not necessary) is an experimental evaluation to demonstrate the practical effectiveness of the proposed algorithms.

---

> ### Author Response · Authors · 2022-08-02
> **Response to Reviewer pUsh**
>
> We thank you for your positive feedback. We also thank you for bringing to our attention the paper of Sumengen et al., which we were not aware of prior to submission, but will now add a comparison to in the final version of the paper. While the aforementioned paper shares the same broad motivation as ours —  scaling hierarchical clustering to massive datasets — there are several fundamental differences in our works which makes their work largely orthogonal to ours, making a direct comparison difficult.
>
> (a) The foremost point of difference in our works is that the aforementioned paper focuses on parallelizing a very specific type of HC algorithm - hierarchical agglomerative clustering algorithms under a restrictive class of linkage functions.  Our work on the other hand focuses on reducing the computation resource (time/space/communication) requirement of *any given* hierarchical clustering algorithm through data sparsification. While our works share the same motivation, the objective in our works is an important philosophical difference. In the aforementioned work, the objective is to recover (in parallel) the *same cluster tree* that is computed by the usual (sequential) agglomerative HC method. While a natural objective, it has a significant limitation: without restrictive generative assumptions on the input data, parallelization is provably impossible. Our work approaches this problem from an optimization perspective (objective function defined in [15] for dissimilarity data); through sparsification, we aim to recover a cluster tree with *marginal loss in objective function value* as compared to one computed over the entire (dense) input data, achieving a speedup in runtime through sparsity. Moreover, we make no generative assumptions on the input data. We also note that with the exception of average-linkage (for which some weak guarantees are known [10,15]), the agglomerative clustering algorithms considered in the aforementioned work do not have any formal guarantees with regards to the optimization objective of [15] in general.
>
> (b) The aforementioned paper assumes a very specific class of linkage functions - reducible linkage functions. This assumption is crucially important because it allows one to update the dissimilarity of any pair of clusters in $O(1)$ time rather than a time dependent on the cluster sizes and inter-cluster edge density. This substantially reduces the dependence of the runtime on the graph density (the lack of such a restrictive assumption is one of the key sources of difficulty in our work). What is interesting to note however, is that if the result of Sumengen et al were to be generalized to richer classes of linkage functions, then our structural result might potentially be very useful as a preprocessing step (but would again require us to approach the problem from an optimization viewpoint); since sparsification does not distort the objective function value significantly, it might be possible to first sparsify the graph using our sublinear time sparsification result to substantially speed up the update step after every merge. This seems like a very interesting problem for future work.
>
> We hope this answers the reviewer’s question, and would be more than happy to discuss any of the above points in further detail.
>
> We also note that we have conducted some preliminary experiments on real-world datasets, whose results are presented in a separate response (visible to all). We intend to add a more comprehensive set of experiments in the final version of the paper.

---

> > ### Comment · Reviewer_pUsh · 2022-08-08
> > **Response**
> >
> > Thank you for the detailed clarification w.r.t. related work as well as the preliminary experimental results.  They have answered my questions.

---

### Official Review · Reviewer_CYrg · 2022-07-15

**Rating:** 6
**Confidence:** 3
**Soundness:** 2 fair
**Presentation:** 3 good
**Contribution:** 2 fair

**Summary:**

This paper designs sublinear algorithms for hierarchical clustering in dynamic streaming model, query model and MPC model over very large graphs. Minimum cost hierarchical partitioning is the optimization objection (Dasgupta’s cost function). From the theoretical aspect, this paper proved the bounds including lower and upper bound respectively for the three given models. They prove a general structural result that shows that a cut sparsifier can be used to recover a (1 +o(1))-approximation to the underlying HC instance.



**Questions:**

none

**Limitations:**

1.	Some recent works prove a seem better bound for the same problem. The authors should compare to them. For example, Assadi S, Chatziafratis V, Mirrokni V, et al. Hierarchical Clustering in Graph Streams: Single-Pass Algorithms and Space Lower Bounds[C]//Conference on Learning Theory. PMLR, 2022: 4643-4702.
2.	Although proofs on bounds are good evidence for showing the performance, it would be better and attractive to give some practical experimental studies to show the real effectiveness in massive graphs. In some cases, a well theoretical complexity still be not enough for the real huge graphs.

**Strengths And Weaknesses:**

Although this research problem is not novel, sublinear algorithms for hierarchical clustering is one of the fundamental issue in the studies of graphs. Lots of applications need the sublinear algorithms especially for massive graphs. The paper is well written with a good presentation. It supplies enough theoretical proofs for the complexity and (1 +o(1))-approximation is a good general structural result in the three models. However, I have some concerns about the approach and applications.
1.	Some recent works prove a seem better bound for the same problem. The authors should compare to them. For example, Assadi S, Chatziafratis V, Mirrokni V, et al. Hierarchical Clustering in Graph Streams: Single-Pass Algorithms and Space Lower Bounds[C]//Conference on Learning Theory. PMLR, 2022: 4643-4702.
2.	Although proofs on bounds are good evidence for showing the performance, it would be better and attractive to give some practical experimental studies to show the real effectiveness in massive graphs. In some cases, a well theoretical complexity still be not enough for the real huge graphs.

Minor problem:
Although the presentation is relatively well, some necessary examples are needed for the readers to follow clearly.

---

> ### Author Response · Authors · 2022-08-02
> **Response to Reviewer CYrg**
>
> We thank you for your positive feedback and shall address the concerns you have, which are (a) comparison to the recent paper by Assadi S, Chatziafratis V, Mirrokni V, et al. (b) experimental study.
>
> (a) We note that in our submission we already compared our results with the paper by Assadi S, Chatziafratis V, Mirrokni V, et al. (which we cited as [5]) in our discussion of “recent independent work” (see the bottom of page 8), and noted that the only overlap between their results and ours is the single-pass streaming algorithm and the 2-round MPC algorithm. There seems to be a misunderstanding on behalf of the reviewer in their interpretation of our results: for both these settings, the bounds obtained in Assadi et al. are in fact slightly **weaker** than our bounds. We also give a detailed comparison below.
>
> We remark that their work’s primary focus is on estimating the value of the optimal HC while ours is on explicitly computing a good HC tree. However, they do give a single-pass streaming algorithm and a 2-round MPC algorithm for outputting an HC tree. Nonetheless, we note that their bound is in fact slightly weaker than us, by the following comparisons: (i) the space/communication resource they use is the same as us, both $\tilde{O}(n)$; (ii) their approximation guarantee is slightly *worse* than us, in the sense that comparing to an offline HC algorithm, they lose some absolute constant factor $C > 1$ in the approximation ratio, while we only lose a factor of $(1+o(1))$ which is asymptotically strictly smaller than $C$.
>
> (b) We note that the main focus of our paper is to provide a complete picture of HC in the three well-studied sublinear computation models. However, in light of your suggestion, we conducted some preliminary experiments to test our HC framework on real-world datasets – one that allows us to significantly reduce the size of the input data while incurring only a negligible loss in the solution quality. We intend to add a more comprehensive set of experiments in the final version of the paper. In particular, we implemented our data sparsification algorithm (Algorithm 3) and recorded the tradeoff between the sparsification rate and the solution quality of some existing HC algorithms. We present our experimental results in a separate response (visible to all) and will include them in the final version. These results indicate that our algorithm preserves the solution quality particularly well, while reducing the data size by a significant amount.

---

### Author Response · Authors · 2022-08-02
**Preliminary Experimental Results**

In light of the reviewers’ suggestion, we conducted some preliminary experiments whose results are presented here. We intend to add a more comprehensive set of experiments in the final version of the paper. The goal of these experiments is to test our data/graph sparsification framework (Algorithm 3 of our submission) which forms the basis of all our algorithms under the different models of computation. We ask the following question: if we run an HC algorithm on the sparsified graph produced by Algorithm 3, then what will be the loss in the quality of solution compared to the same HC algorithm run on the original graph? We would also like to understand the tradeoff between the sparsification rate and the solution quality of certain HC algorithms.

We test our data sparsification algorithm on two large real-world datasets Boston and Newsgroup, which are both part of the standard Scikit-learn library and were used in previous HC papers [D2,D3]. We calculate the similarity between two data points using the Gaussian kernel as in [D1,D3] and only insert an edge if its weight (i.e. similarity between its two endpoints) is bigger than $10^{-10}$.

Since we are not aware of any existing implementation of the recursive sparsest cut algorithm, we choose to test two linkage algorithms, namely average linkage and complete linkage, which, as shown in [D3], are among the best-performed algorithms on the two datasets we consider. We run these two algorithms both on the entire graph and on the sparsified graphs obtained by our Algorithm 3, and compare the cost of the HC we get in each case, where the cost is measured using Dasgupta’s objective function (Eq.1 in our paper). When running our Algorithm 3, we also try different sampling rates to obtain subgraphs of various edge densities. Note that, we always measure the cost of an HC in the *original* graph as opposed to in the subsampled graph.

We present our experimental results in the tables below. Here we write **absolute HC cost**  to denote the cost of the HC obtained by running the corresponding algorithm on the entire graph, and write **relative HC cost** to denote the cost of the HC obtained by running the algorithm on the subsampled graph *divided by* the absolute HC cost.

For the Boston dataset, one can see that even when we only sample <5% of the edges, we already get a very good relative HC cost of ~1.15; for the Newsgroup dataset, sampling only <0.5% of the edges already gives us a low relative HC cost of 1.188 for average linkage and 1.05 for complete linkage. As the number of sampled edges grow, the relative HC cost becomes smaller and gets closer to 1.

### Boston Dataset (506 Vertices, 95566 Edges)

**Performance of Average Linkage (Absolute HC Cost is 186085.271):**

| &nbsp; Number (Percentage) of Sampled Edges &nbsp;  |   &nbsp; Relative HC Cost |
| :-------------: |:-------------:|
| 4040 (4.23%)      | 1.146     |
| 5861 (6.13%)      | 1.068     |
| 7010 (7.34%)      | 1.043     |
| 9018 (9.44%)      | 1.016     |
| 9756 (10.21%)    | 1.012     |
| 11941 (12.50%)  | 1.000     |

**Performance of Complete Linkage (Absolute Cost is 226485.146):**

| &nbsp; Number (Percentage) of Sampled Edges &nbsp;  |   &nbsp; Relative HC Cost |
| :-------------: |:-------------:|
| 4032 (4.22%)      | 1.158     |
| 5885 (6.16%)      | 1.135     |
| 7002 (7.33%)      | 1.113     |
| 9033 (9.45%)      | 1.066     |
| 9773 (10.23%)    | 1.037     |
| 11900 (12.45%)  | 1.006     |

### Newsgroup Dataset (3516 Vertices, 6178853 Edges)

**Performance of Average Linkage (Absolute Cost is 3960145237.881):**

| &nbsp; Number (Percentage) of Sampled Edges &nbsp;  |   &nbsp; Relative HC Cost |
| :-------------: |:-------------:|
| 28495 (0.46%)       | 1.188    |
| 55116 (0.89%)       | 1.169    |
| 103804 (1.68%)     | 1.143    |
| 203095 (3.29%)     |  1.118   |
| 358440 (5.80%)     | 1.111    |
| 1029648 (16.66%) | 1.068   |

**Performance of Complete Linkage (Absolute Cost is 3960674776.482):**

| &nbsp; Number (Percentage) of Sampled Edges &nbsp;  |   &nbsp; Relative HC Cost |
| :-------------: |:-------------:|
| 28476 (0.46%)       | 1.050  |
| 55042 (0.89%)       | 1.010  |
| 103756 (1.68%)     | 1.001  |
| 203323 (3.29%)     | 1.000  |
| 358446 (5.80%)     | 1.000  |
| 1029274 (16.66%) | 1.000  |

### References

[D1] Roy A, Pokutta S. Hierarchical clustering via spreading metrics. Advances in Neural Information Processing Systems. 2016;29.

[D2] Cohen-Addad V, Kanade V, Mallmann-Trenn F. Hierarchical clustering beyond the worst-case. Advances in Neural Information Processing Systems. 2017;30.

[D3] Manghiuc BA, Sun H. Hierarchical Clustering: $O(1)$-Approximation for Well-Clustered Graphs. Advances in Neural Information Processing Systems. 2021 Dec 6;34:9278-89.

---

### Meta-Review · Area_Chair_D3CP · 2022-08-26

**Recommendation:** Accept
**Confidence:** Certain

**Metareview:**

The paper presents new algorithm for hierarchical clustering in different regimes. In particular they show a new algorithm for a (dynamic) edge streaming model, for a neighbor query model and for the MPC model. The paper contains both nice theory results and in the rebuttal phase the author(s) supported them with interesting experimental results.

Overall, we suggest to accept the paper as poster.

**Award:**

No

---

### Decision · Program_Chairs · 2022-09-14

Accept